# Align Your Flow:
# Scaling Continuous-Time Flow Map Distillation

**Amirmojtaba Sabour**[1,2,3]  **Sanja Fidler**[1,2,3]  **Karsten Kreis**[1]

[1] NVIDIA  [2] University of Toronto  [3] Vector Institute

*Project Page:* https://research.nvidia.com/labs/toronto-ai/AlignYourFlow/

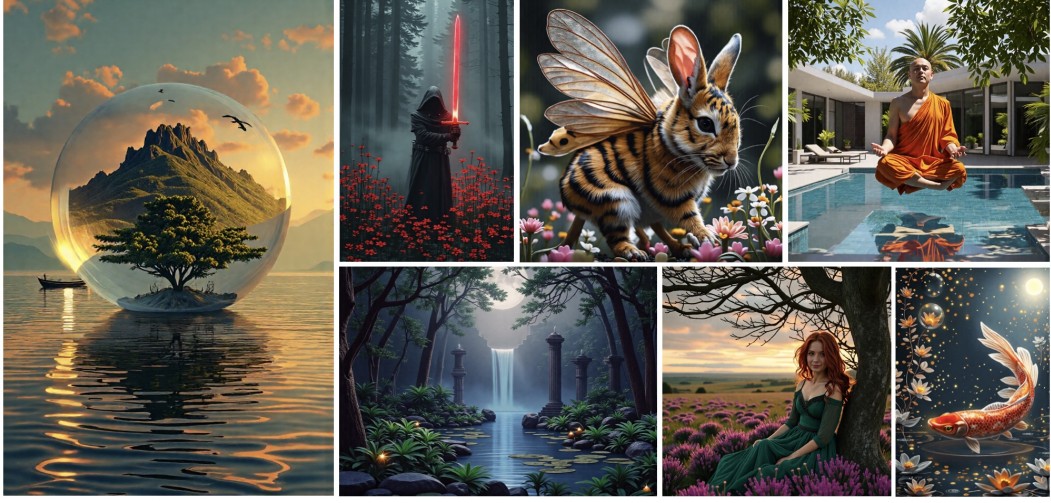

Figure 1: Four-step samples by our distilled text-conditioned flow map model (prompts in App. I).

## Abstract

Diffusion- and flow-based models have emerged as state-of-the-art generative modeling approaches, but they require many sampling steps. Consistency models can distill these models into efficient one-step generators; however, unlike flow- and diffusion-based methods, their performance inevitably degrades when increasing the number of steps, which we show both analytically and empirically. Flow maps generalize these approaches by connecting any two noise levels in a single step and remain effective across all step counts. In this paper, we introduce two new continuous-time objectives for training flow maps, along with additional novel training techniques, generalizing existing consistency and flow matching objectives. We further demonstrate that autoguidance can improve performance, using a low-quality model for guidance during distillation, and an additional boost can be achieved by adversarial finetuning, with minimal loss in sample diversity. We extensively validate our flow map models, called *Align Your Flow*, on challenging image generation benchmarks and achieve state-of-the-art few-step generation performance on both ImageNet 64x64 and 512x512, using small and efficient neural networks. Finally, we show text-to-image flow map models that outperform all existing non-adversarially trained few-step samplers in text-conditioned synthesis.

39th Conference on Neural Information Processing Systems (NeurIPS 2025).

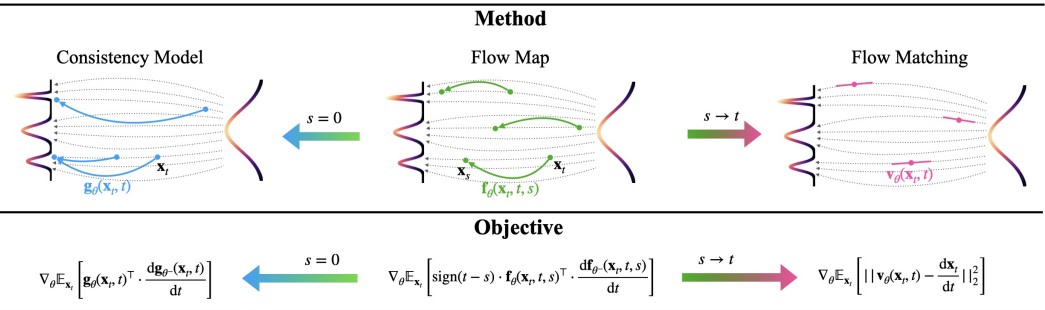

Figure 2: **Overview of Flow Maps**. Flow maps generalize both consistency models and flow matching by connecting any two noise levels $(s, t)$ in a single step. When $s = 0$, flow maps reduce to consistency models; when $s \to t$ they're equivalent to standard flow matching models. Our proposed EMD objective (see Theorem 3.2) similarly generalizes the continuous-time consistency and flow matching losses. For detailed derivations, please see the Appendix.

## 1 Introduction

Diffusion and flow-based generative models have revolutionized generative modeling [25, 70, 45, 62, 14, 8], but they rely on slow iterative sampling. This has led to the development of approaches to accelerate generation. Advanced, higher-order samplers [68, 50, 51, 12, 89, 34, 61] help, but cannot produce high quality outputs with <10 steps. Distillation techniques [63, 67, 87, 72], in contrast, can successfully distill models into few-step generators. In particular, consistency models [71, 69, 49] and a variety of related techniques [38, 79, 80, 42, 94, 17, 22] have gained much attention recently. Consistency models learn to transfer samples that lie on teacher-defined deterministic noise-to-data paths to the same, consistent clean outputs in a single prediction. These approaches excel in few step generation, but have been empirically shown to degrade in performance as the number of steps increases.

In this work, we analytically show that consistency models are inherently incompatible with multi-step sampling. Specifically, we show that their objective of strictly predicting clean outputs inevitably leads to error accumulation over multiple denoising steps. Motivated by this limitation, we turn to the **flow map** formulation as a unifying and more robust alternative. The flow map framework - also known as Consistency Trajectory Models - was introduced in [38, 5] and encompasses diffusion and flow-based models [45], consistency models [71, 69, 49], and other distillation variants [80, 17, 94, 95] within a single coherent formulation. Flow maps allow connecting any two noise levels in a single step, enabling efficient few-step sampling as well as flexible multi-step sampling. As flow maps, figuratively speaking, learn a mapping that "aligns the teacher flow" into a few-step sampler, we call our approach *Align Your Flow (AYF)*. We propose two new continuous-time training objectives, which can be interpreted as AYF's versions of the Eulerian and Lagrangian losses described by Boffi et al. [5]. The new objectives use a consistency condition at either the beginning or the end of a denoising interval. Notably, the first of our objectives generalizes both the continuous-time consistency loss [71, 49] and the flow matching loss [45]. While regular consistency models only perform well for single- or two-step generation and degrade for multi-step sampling, e.g. for 4 steps or more, flow map models such as AYF produce high-quality outputs in this multi-step setting, too.

To scale AYF to high performance, we leverage the recently proposed autoguidance [35], where a low-quality guidance model checkpoint is used together with the regular model to produce a model with enhanced quality. Specifically, we propose to distill an autoguided teacher model into an AYF student and introduce several practical techniques that stabilize flow map training and push performance further. Moreover, unlike prior distillation approaches that rely on adversarial training to boost quality at the expense of sample diversity [67, 66, 87, 86, 38], we show that a short finetuning of a pretrained AYF model with a combination of our proposed flow map objective and an adversarial loss is sufficient to yield significantly sharper images with minimal impact on diversity.

We validate AYF on popular image generation benchmarks and achieve state-of-the-art performance among few-step generators on both ImageNet 64x64 and 512x512, while using only small and efficient neural networks (Fig. 4). For instance, 4-step sampling of AYF's ImageNet models is as fast or faster than previous works' single step generation. Additionally, our adversarially finetuned AYF also achieves significantly higher diversity compared to other adversarial training approaches. We further distill the popular FLUX.1 model [41] and obtain text-to-image AYF flow map models that significantly outperform all existing non-adversarially trained few-step generators in text-conditioned

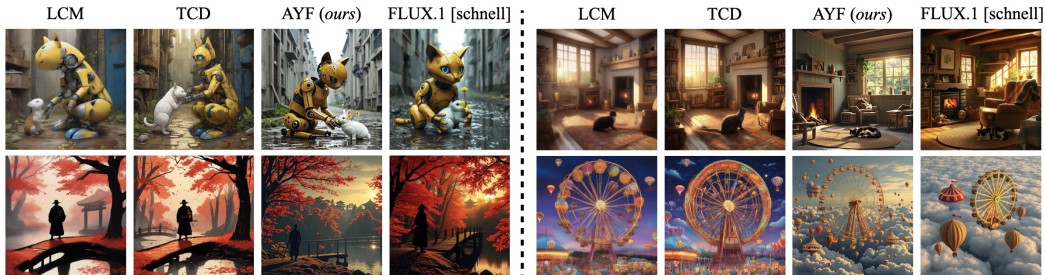

| LCM | TCD | AYF (*ours*) | FLUX.1 [schnell] | LCM | TCD | AYF (*ours*) | FLUX.1 [schnell] |

Figure 3: Samples (4 steps): LCM [54], TCD [94], FLUX.1 [schnell] [41], AYF (view zoomed in).

synthesis (Fig. 1). For these experiments, we use an efficient LoRA [27] framework, avoiding the overhead of many previous text-to-image distillation approaches.

**Contributions.** *(i)* We prove that consistency models inherently suffer from error accumulation in multi-step sampling. *(ii)* We propose *Align Your Flow*, a high-performance few-step flow map model with new theoretical insights. *(iii)* We introduce two new training objectives and stabilization techniques for flow map learning. *(iv)* We apply autoguidance for distillation for the first time and show that adversarial finetuning further boosts performance with minimal loss in diversity. *(v)* We achieve state-of-the-art few-step generation performance on ImageNet, and we also show fast high-resolution text-to-image generation, outperforming all non-adversarial methods in this task.

## 2   Background

**Diffusion Models and Flow Matching.** Diffusion models are probabilistic generative models that inject noise into the data with a forward diffusion process and generate samples by learning and simulating a time-reversed backward diffusion process, initialized by pure Gaussian noise. Flow matching [45, 48, 2, 1, 39] is a generalization of these methods that eliminates the requirement of the noise being Gaussian and allows learning a continuous flow between any two distributions $p_0, p_1$ that converts samples from one to the other.

Denote the data distribution by $\mathbf{x}_0 \sim p_{\text{data}}$ and the noise distribution by $\mathbf{x}_1 \sim p_{\text{noise}}$. Let $\mathbf{x}_t = (1 - t) \cdot \mathbf{x}_0 + t \cdot \mathbf{x}_1$ indicate the noisy samples of the data for time $t \in [0, 1]$, corresponding to the rectified flow [48] or conditional optimal transport [45] formulation. The flow matching training objective is then given by $\mathbb{E}_{\mathbf{x}_0, \mathbf{x}_1, t} \left[ w(t) || \mathbf{v}_\theta(\mathbf{x}_t, t) - (\mathbf{x}_1 - \mathbf{x}_0) ||_2^2 \right]$; $w(t)$ is a weighting function and $\mathbf{v}_\theta$ is a neural network parametrized by $\theta$. The standard sampling procedure starts at $t = 1$ by sampling $\mathbf{x}_1 \sim p_{\text{noise}}$. Then the probability flow ODE (PF-ODE), defined by $\frac{d\mathbf{x}_t}{dt} = \mathbf{v}_\theta(\mathbf{x}_t, t)dt$, is simulated from $t = 1$ to $t = 0$ to obtain the final outputs. We will assume to be in the flow matching framework from this point on of the paper.

**Consistency Models.** Consistency models (CM) [71] train a neural network $\mathbf{f}_\theta(\mathbf{x}_t, t)$ to map noisy inputs $\mathbf{x}_t$ directly to their corresponding clean samples $\mathbf{x}_0$, following the PF-ODE. Consequently, $\mathbf{f}_\theta(\mathbf{x}_t, t)$ must satisfy the *boundary condition* $\mathbf{f}_\theta(\mathbf{x}, 0) = \mathbf{x}$, which is typically enforced by parameterizing $\mathbf{f}_\theta(\mathbf{x}_t, t) = c_{\text{skip}}(t)\mathbf{x}_t + c_{\text{out}}(t)\mathbf{F}_\theta(\mathbf{x}_t, t)$ with $c_{\text{skip}}(0) = 1, c_{\text{out}}(0) = 0$. CMs are trained to have consistent outputs between adjacent timesteps. They can be trained from scratch or distilled from given diffusion or flow models. In this work, we are focusing on distillation. Depending on how time is dealt with, CMs can be split into two categories:

**Discrete-time CMs.** The training objective is defined between adjacent timesteps as

$$\mathbb{E}_{\mathbf{x}_t, t} \left[ w(t) d(\mathbf{f}_\theta(\mathbf{x}_t, t), \mathbf{f}_{\theta^-}(\mathbf{x}_{t-\Delta t}, t - \Delta t)) \right], \tag{1}$$

where $\theta^-$ denotes *stopgrad*($\theta$), $w(t)$ is a weighting function, $\Delta t > 0$ is the distance between adjacent timesteps, and $d(., .)$ is a distance function. Common choices include $\ell_2$ loss $d(\mathbf{x}, \mathbf{y}) = ||\mathbf{x} - \mathbf{y}||_2^2$, Pseudo-Huber loss $d(\mathbf{x}, \mathbf{y}) = \sqrt{||\mathbf{x} - \mathbf{y}||_2^2 + c^2} - c$ [69], and LPIPS loss [90]. Discrete-time CMs are sensitive to the choice of $\Delta t$, and require manually designed annealing schedules [71, 18]. The noisy sample $\mathbf{x}_{t-\Delta t}$ at the preceding timestep $t - \Delta t$ is often obtained from $\mathbf{x}_t$ by numerically solving the PF-ODE, which can cause additional discretization errors.

**Continuous-time CMs.** When using $d(\mathbf{x}, \mathbf{y}) = ||\mathbf{x} - \mathbf{y}||_2^2$ and taking the limit $\Delta t \to 0$, Song et al. [71] show that the gradient of Eq. (1) with respect to $\theta$ converges to

$$\nabla_\theta \mathbb{E}_{\mathbf{x}_t, t} \left[ w(t) \mathbf{f}_\theta^\top (\mathbf{x}_t, t) \frac{\mathrm{d}\mathbf{f}_{\theta^-}(\mathbf{x}_t, t)}{\mathrm{d}t} \right], \tag{2}$$

where $\frac{\mathrm{d}\mathbf{f}_{\theta^-}(\mathbf{x}_t, t)}{\mathrm{d}t} = \nabla_{\mathbf{x}_t} \mathbf{f}_{\theta^-}(\mathbf{x}_t, t) \frac{\mathrm{d}\mathbf{x}_t}{\mathrm{d}t} + \partial_t \mathbf{f}_{\theta^-}(\mathbf{x}_t, t)$ is the tangent of $\mathbf{f}_{\theta^-}$ at $(\mathbf{x}_t, t)$ along the trajectory of the PF-ODE $\frac{\mathrm{d}\mathbf{x}_t}{\mathrm{d}t}$. This means continuous-time CMs do not need to rely on numerical ODE solvers which avoids discretization errors and offers better supervision signals during training. Recently, Lu and Song [49] successfully stabilized and scaled continuous-time CMs and achieved significantly better results compared to the discrete-time approach.

# 3 Continuous-Time Flow Map Distillation

Flow maps generalize diffusion, flow-based and flow and consistency models within a single unified framework by training a neural network $\mathbf{f}_\theta(\mathbf{x}_t, t, s)$ to map noisy inputs $\mathbf{x}_t$ directly to any point $\mathbf{x}_s$ along the PF-ODE in a single step. Unlike consistency models, which only perform well for single- or two-step generation but degrade in multi-step sampling, flow maps remain effective at all step counts.

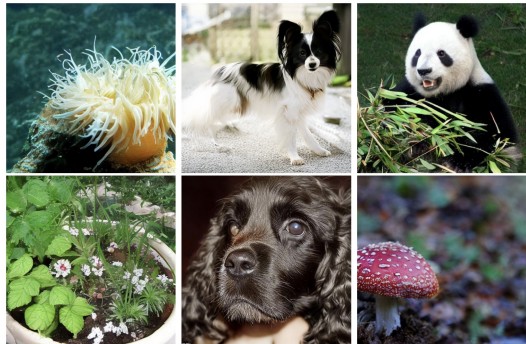

Figure 4: Two-step AYF samples on ImageNet512.

In Sec. 3.1, we first show that standard consistency models are incompatible with multi-step sampling, leading to inevitable performance degradation beyond a certain step count. Next, in Sec. 3.2, we introduce two novel continuous-time objectives for distilling flow maps from a pretrained flow model. Finally, in Sec. 3.3, we explain how we leverage autoguidance to sharpen the flow map. Sec. 3.4 addresses implementation details. The detailed training algorithm for AYF is provided in the Appendix.

## 3.1 Consistency Models are Flawed Multi-Step Generators

CMs are a powerful approach to turn flow-based models into one-step generators. To allow CMs to trade compute for sample quality, a multi-step sampling procedure was introduced by Song et al. [71]. This process sequentially denoises noisy $\mathbf{x}_t$ by first removing all noise to estimate the clean data and then reintroducing smaller amounts of noise. However, in practice, this sampling procedure performs poorly as the number of steps increases and most prior works only demonstrate 1- or 2-step results.

To understand this behavior, we analyze a simple case where the initial distribution is an isotropic Gaussian with standard deviation $c$, i.e. $p_{\text{data}}(\mathbf{x}) = \mathcal{N}(\mathbf{0}, c^2 \mathbf{I})$. The following theorem shows that regardless of how accurate a (non-optimal) CM is, increasing the number of sampling steps beyond a certain point will lead to worse performance due to error accumulation in that setting.

**Theorem 3.1** (Proof in Appendix). *Let $p_{data}(\mathbf{x}) = \mathcal{N}(\mathbf{0}, c^2 \mathbf{I})$ be the data distribution, and let $\mathbf{f}^*(\mathbf{x}_t, t)$ denote the optimal consistency model. For any $\delta > 0$, there exists a suboptimal consistency model $\mathbf{f}(\mathbf{x}_t, t)$ such that*

$$\mathbb{E}_{\mathbf{x}_t \sim p(\mathbf{x}, t)} \big[ \|\mathbf{f}(\mathbf{x}_t, t) - \mathbf{f}^*(\mathbf{x}_t, t)\|_2^2 \big] < \delta \quad \text{for all } t \in [0, 1],$$

*and there is some integer $N$ for which **increasing** the number of sampling steps beyond $N$ **increases** the Wasserstein-2 distance of the generated samples to the ground truth distribution (i.e. a worse approximation of the ground truth).*

This suggests that CMs, by design, are not suited for multi-step generation. Interestingly, when $c = 0.5$—a common choice in diffusion model training, where the data is often normalized to this std. dev. [34]—multi-step CM sampling with a non-optimal CM produces the best samples at two steps (Fig. 5). This is in line with common observations in the literature [49]. This behavior is the opposite of standard diffusion models, which improve as the number of steps increases. Prior works have attempted to address this issue (see Sec. 4), and they all ultimately reduce to special cases of flow maps.

## 3.2 Learning Flow Maps

Flow maps are neural networks $\mathbf{f}_\theta(\mathbf{x}_t, t, s)$ that general- ize CMs by mapping a noisy input $\mathbf{x}_t$ directly to any other point $\mathbf{x}_s$ by following the PF-ODE from time $t$ to $s$. When $s = 0$, they reduce to standard CMs. When performing many small steps, they become equiv- alent to regular flow or diffusion model sampling with the PF-ODE and Euler integration. A valid flow map $\mathbf{f}_\theta(\mathbf{x}_t, t, s)$ must satisfy the *general boundary condition* $\mathbf{f}_\theta(\mathbf{x}_t, t, t) = \mathbf{x}_t$ for all $t$. As is done in prior work, this is enforced in practice by parameterizing the model as $\mathbf{f}_\theta(\mathbf{x}_t, t, s) = c_{\text{skip}}(t, s)\mathbf{x}_t + c_{\text{out}}(t, s)\mathbf{F}_\theta(\mathbf{x}_t, t, s)$ where $c_{\text{skip}}(t, t) = 1$ and $c_{\text{out}}(t, t) = 0$ for all $t$. In this work, we set $c_{\text{skip}}(t, s) = 1$ and $c_{\text{out}}(t, s) = (s - t)$ for simplicity and to align it with an Euler ODE solver.

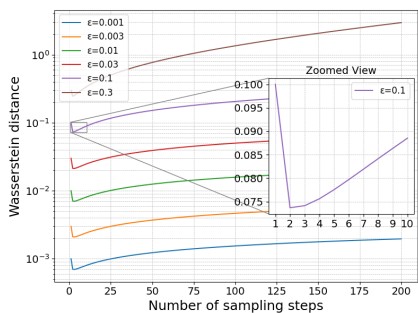

Figure 5: Wasserstein-2 distance be- tween multi-step consistency samples and data distribution ($c$=0.5).

Unlike CMs, which perform poorly in multi-step sampling, flow maps are designed to excel in this scenario. Additionally, their ability to fully traverse the PF-ODE enables them to accelerate tasks such as image inversion and editing by directly mapping images to noise [43].

As we are interested in distilling a diffusion or flow matching model, we assume access to a pretrained velocity model $\mathbf{v}_\phi(\mathbf{x}_t, t)$. The flow map model is trained by aligning its single-step predictions with the trajectories generated by the teacher's PF-ODE, i.e. $\frac{d\mathbf{x}_t}{dt} = \mathbf{v}_\phi(\mathbf{x}_t, t)$. We propose two primary methods for training flow maps. The first training objective aims to ensure that for a fixed $s$, the output of the flow map remains constant as we move $(\mathbf{x}_t, t)$ along the PF-ODE. Let $\theta^- = \text{stopgrad}(\theta)$. The theorem below summarizes the approach. We call this loss *AYF-Eulerian Map Distillation (AYF-EMD)*, as it can also be interpreted as a variant of the Eulerian loss of Boffi et al. [5]. The AYF-EMD loss naturally generalizes the loss used to train continuous-time consistency models [71, 49], as it reduces to the same objective when $s = 0$. Interestingly, it also generalizes the standard flow matching loss, to which it reduces in the limit as $s \to t$. See Appendix for details.

**Theorem 3.2** (Proof in Appendix). *Let $\mathbf{f}_\theta(\mathbf{x}_t, t, s)$ be the flow map. Consider the loss function defined between two adjacent starting timesteps $t$ and $t' = t + \epsilon(s - t)$ for a small $\epsilon > 0$,*

$$\mathbb{E}_{\mathbf{x}_t, t, s}\left[w(t, s)||\mathbf{f}_\theta(\mathbf{x}_t, t, s) - \mathbf{f}_{\theta^-}(\mathbf{x}_{t'}, t', s)||_2^2\right],$$

*where $\mathbf{x}_{t'}$ is obtained by applying a 1-step Euler solver to the PF-ODE from $t$ to $t'$. In the limit as $\epsilon \to 0$, the gradient of this objective with respect to $\theta$ converges to:*

$$\nabla_\theta \mathbb{E}_{\mathbf{x}_t, t, s}\left[w'(t, s)\text{sign}(t - s) \cdot \mathbf{f}_\theta^\top(\mathbf{x}_t, t, s) \cdot \frac{d\mathbf{f}_{\theta^-}(\mathbf{x}_t, t, s)}{dt}\right],$$

*where $w'(t, s) = w(t, s) \times |t - s|$.*

The second approach ensures consistency at timestep $s$ instead. This method tries to ensure that for a fixed $(\mathbf{x}_t, t)$, the trajectory $\mathbf{f}_\theta(\mathbf{x}_t, t, \cdot)$ is aligned with that points' PF-ODE. We call this loss *AYF-Lagrangian Map Distillation (AYF-LMD)*, as it is related to the Lagrangian loss of Boffi et al. [5]. The theorem below formalizes this approach.

**Theorem 3.3** (Proof in Appendix). *Let $\mathbf{f}_\theta(\mathbf{x}_t, t, s)$ be the flow map. Consider the loss function defined between two adjacent ending timesteps $s$ and $s' = s + \epsilon(t - s)$ for a small $\epsilon > 0$,*

$$\mathbb{E}_{\mathbf{x}_t, t, s}\left[w(t, s)||\mathbf{f}_\theta(\mathbf{x}_t, t, s) - ODE_{s' \to s}[\mathbf{f}_{\theta^-}(\mathbf{x}_t, t, s')]||_2^2\right],$$

*where $ODE_{t \to s}(\mathbf{x})$ refers to running a 1-step Euler solver on the PF-ODE starting from $\mathbf{x}$ at timestep $t$ to timestep $s$. In the limit as $\epsilon \to 0$, the gradient of this objective with respect to $\theta$ converges to:*

$$\nabla_\theta \mathbb{E}_{\mathbf{x}_t, t, s}\left[w'(t, s)\,\text{sign}(s - t) \cdot \mathbf{f}_\theta^\top(\mathbf{x}_t, t, s) \cdot \left(\frac{d\mathbf{f}_{\theta^-}(\mathbf{x}_t, t, s)}{ds} - \mathbf{v}_\phi(\mathbf{f}_{\theta^-}(\mathbf{x}_t, t, s), s)\right)\right],$$

*where $w'(t, s) = w(t, s) \times |t - s|$.*

In our 2D toy experiments, comparing the two objectives above, we found the AYF-LMD objective to be more stable. However, when applied to image datasets, it leads to overly smoothened samples that drastically reduce the output quality (see Appendix for detailed ablation studies).

### 3.3 Sharpening the Distribution with Autoguidance

The training objective of diffusion- and flow-based models strongly encourages the model to cover the entire data distribution. Yet it lacks enough data to learn how to generate good samples in the tails of the distribution. The issue is even worse in distilled models which use fewer sampling steps. As a result, many prior distillation methods rely on adversarial objectives to achieve peak performance, often sacrificing diversity and ignoring low-probability regions altogether. The most commonly used technique to partially address this in conditional diffusion and flow-based models is classifier-free guidance (CFG) [24]. CFG trains a flow or diffusion model for both conditional and unconditional generation and steers samples away from the unconditional regions during sampling. Prior works [57, 49] have explored distilling CFG with great success. However, CFG struggles with overshooting the conditional distribution at large guidance scales, which leads to overly simplistic samples [40].

Recently, Karras et al. [35] introduced autoguidance as a better alternative for CFG. Unlike CFG, this technique works for unconditional generation as well. Autoguidance uses a smaller, less trained version of the main model for guidance, essentially steering samples away from low-quality sample regions in the probability distribution, where the weaker guidance model performs particularly poorly. We found that distilling autoguided teacher models can significantly improve performance compared to standard CFG. To the best of our knowledge, we are the first to demonstrate the distillation of autoguided teachers. Specifically, during flow map distillation we define the guidance scale $\lambda$ and use the autoguided teacher velocity

$$\mathbf{v}_\phi^{\text{guided}}(\mathbf{x}_t, t) = \lambda \mathbf{v}_\phi(\mathbf{x}_t, t) + (1 - \lambda)\mathbf{v}_\phi^{\text{weak}}(\mathbf{x}_t, t), \tag{3}$$

where $\mathbf{v}_\phi^{\text{weak}}$ represents the weaker guidance model. In summary, we use autoguidance in the teacher as a mechanism to "sharpen" the distilled flow map model. See Appendix for a visual comparison between autoguidance and CFG on a 2D toy distribution.

### 3.4 Training Tricks

Training continuous-time CMs has historically been unstable [69, 18]. Recently, sCM [49] addressed this issue by introducing techniques focused on parameterization, network architectures, and modifications to the training objective. Following their approach, we stabilize time embeddings and apply tangent normalization, while also introducing a few additional techniques to further improve stability.

Our image models are trained with the AYF-EMD objective in Theorem 3.2, which relies on the tangent function $\frac{d\mathbf{f}_{\theta^-}(\mathbf{x}_t, t, s)}{dt}$. Under our parametrization, this tangent function is computed by

$$\frac{d\mathbf{f}_{\theta^-}(\mathbf{x}_t, t, s)}{dt} = \left( \frac{d\mathbf{x}_t}{dt} - \mathbf{F}_{\theta^-}(\mathbf{x}_t, t, s) \right) + (s - t) \times \frac{d\mathbf{F}_{\theta^-}(\mathbf{x}_t, t, s)}{dt}, \tag{4}$$

where $\frac{d\mathbf{x}_t}{dt} = \mathbf{v}_\phi(\mathbf{x}_t, t)$ represents the direction given by the pretrained diffusion or flow model along the PF-ODE. We find that most terms in this formulation are relatively stable, except for $\frac{d\mathbf{F}_\theta(\mathbf{x}_t, t, s)}{dt} = \nabla_{\mathbf{x}_t} \mathbf{F}_\theta(\mathbf{x}_t, t, s)\frac{d\mathbf{x}_t}{dt} + \partial_t \mathbf{F}_\theta(\mathbf{x}_t, t, s)$. Among these, the instability originates mainly from $\partial_t \mathbf{F}_\theta(\mathbf{x}_t, t, s)$, which can be decomposed into

$$\partial_t \mathbf{F}_\theta(\mathbf{x}_t, t, s) = \frac{\partial c_{\text{noise}}(t)}{\partial t} \cdot \frac{\partial emb(c_{\text{noise}})}{\partial c_{\text{noise}}} \cdot \frac{\partial \mathbf{F}_\theta}{\partial emb},$$

where $emb(\cdot)$ refers to the time embeddings, most commonly in the form of positional embeddings [25, 78] or Fourier embeddings [70, 73]. sCM [49] proposes several techniques to stabilize this term including tangent normalization, adaptive weighting, and tangent warmup.

We use tangent normalization [49], i.e. $\frac{d\mathbf{f}_{\theta^-}}{dt} \rightarrow \frac{d\mathbf{f}_{\theta^-}}{dt}/(||\frac{d\mathbf{f}_{\theta^-}}{dt}|| + c)$ with $c = 0.1$, as we find it to be critical for stable training. However, in our experiments, adaptive weighting had no meaningful impact and can be removed. We make a few tweaks to the time embeddings and tangent warmup to ensure compatibility with flow matching and better training dynamics which we describe below.

**Stabilizing the Time Embeddings** The time embedding layers are one of the causes for the instability of $\partial_t \mathbf{F}_\theta(\mathbf{x}_t, t, s)$. As noted in [49], the $c_{\text{noise}}$ parameterization used in most CMs is based on the EDM [34] framework, where the noise level is defined as $c_{\text{noise}}(\sigma) = \log(\sigma)$. In the flow matching framework, which we use, the noise level for a timestep $t$ is given by $\sigma_t = \frac{t}{1-t}$, which can lead to instabilities when passing through a $\log$ operation as $t \rightarrow 0$ or $t \rightarrow 1$. To address this, we modify the

time parameterization by setting $c_{\text{noise}}(t) = t$, ensuring stable partial derivatives. To utilize pretrained teacher model checkpoints trained with different time parameterizations, we first finetune the student's time embedding module to align with the outputs of the original checkpoints. For example, if we want to adapt EDM2 checkpoints, which use $\sigma_t = \frac{t}{1-t}$, we minimize the following objective:

$$\mathbb{E}_{t \sim p(t)} \left[ \|emb_{\text{new}}(t) - emb_{\text{original}}(\log(\sigma_t))\|_2^2 \right].$$

This approach enables us to re-purpose nearly any checkpoint, making it compatible with our flow matching framework with minimal finetuning, rather than training new models from scratch.

**Regularized Tangent Warmup** We initialize the student model with pretrained flow matching or diffusion model weights, following prior work to speed up training [49, 71]. Lu and Song [49] proposed a gradual warmup procedure for the second term in Eq. (4), i.e., $\frac{\mathrm{d}\mathbf{f}_{\theta^-}(\mathbf{x}_t, t, s)}{\mathrm{d}t}$. Specifically, they introduced a coefficient $r$ that linearly increases from 0 to 1 over the first 10k training iterations, gradually incorporating the term. This warmup has a clear intuitive motivation. When considering only the first term in Eq. (4) (i.e., the $r = 0$ case), the objective simplifies to a regularization term that encourages flow maps to remain close to straight lines (please see the Appendix for the derivation):

$$\nabla_\theta \left[ \text{sign}(t-s) \mathbf{f}_\theta^\top (\mathbf{x}_t, t, s) \times \left( \frac{\mathrm{d}\mathbf{x}_t}{\mathrm{d}t} - \mathbf{F}_{\theta^-}(\mathbf{x}_t, t, s) \right) \right] \propto \nabla_\theta[\|\mathbf{F}_\theta(\mathbf{x}_t, t, s) - \mathbf{v}_\phi(\mathbf{x}_t, t)\|_2^2]. \quad (5)$$

Therefore, for $r < 1$, the warmed-up loss with coefficient $r$ is equivalent to a weighted sum of the actual loss and this regularization term:

$$\nabla_\theta \left[ \text{sign}(t-s) \mathbf{f}_\theta^\top (\mathbf{x}_t, t, s) \left( \frac{\mathrm{d}\mathbf{x}_t}{\mathrm{d}t} - \mathbf{F}_{\theta^-}(\mathbf{x}_t, t, s) + r(s-t) \frac{\mathrm{d}\mathbf{F}_{\theta^-}(\mathbf{x}_t, t, s)}{\mathrm{d}t} \right) \right]$$

$$= r \nabla_\theta \left[ \text{sign}(t-s) \mathbf{f}_\theta^\top (\mathbf{x}_t, t, s) \frac{\mathrm{d}\mathbf{f}_{\theta^-}(\mathbf{x}_t, t, s)}{\mathrm{d}s} \right] + (1-r) \nabla_\theta \left[ |t - s| \cdot \|\mathbf{F}_\theta(\mathbf{x}_t, t, s) - \mathbf{v}_\phi(\mathbf{x}_t, t)\|_2^2 \right].$$

In our experiments, training these models for too long after the warmup phase can cause destabilization. A simple fix is to clamp $r$ to a value smaller than 1, ensuring some regularization remains. We found $r_{\text{max}} = 0.99$ to be effective in all cases.

**Timestep scheduling** As in standard diffusion, flow-based, and consistency models, selecting an effective sampling schedule for $(t, s)$ during training is crucial. Similar to standard consistency models, where information must propagate from $t = 0$ to $t = 1$ over training, flow map models propagate information from small intervals $|s - t| = 0$ to large ones $|s - t| = 1$. For details on our practical implementation of the schedules, as well as a complete training algorithms, please see the Appendix.

## 4 Related Work

**Consistency Models.** Flow Map Models generalize the seminal CMs, introduced by Song et al. [71]. Early CMs were challenging to train and several subsequent works improved their stability and performance, using new objectives [69], weighting functions [18] or variance reduction techniques [79], among other tricks. Truncated CMs [42] proposed a second training stage, focusing exclusively on the noisier time interval, and Lu and Song [49] successfully implemented continuous-time CMs for the first time.

**Flow Map Models.** Consistency Trajectory Models (CTM) [38] can be considered the first flow map-like models. They combine the approach with adversarial training. Trajectory Consistency Distillation [94] extends CTMs to text-to-image generation, and Bidirectional CMs [43] train additionally on timestep pairs with $t < s$, also accelerating inversion and tasks such as inpainting and blind image restoration. Kim et al. [37] trained CTMs connecting arbitrary distributions. Multistep CMs [22] split the denoising interval into sub-intervals and train CMs within each one, enabling impressive generation quality using 2-8 steps. Phased CMs [80] use a similar interval-splitting strategy combined with an adversarial objective. These methods can be seen as learning flow maps by training on $(t, s)$ pairs, where $s$ is the start of the sub-interval containing $t$. Flow Map Matching [5] provides a rigorous analysis of the continuous-time flow map formulation and proposes several continuous-time losses. Shortcut models [17] adopt a similar flow map framework, but these two works struggle to produce high-quality images—in contrast to our novel AYF, the first high-performance continuous-time flow map model.

**Accelerating Diffusion Models.** Early diffusion distillation approaches are knowledge distillation [52] and progressive distillation [63, 57]. Other methods include adversarial distillation [67, 66], variational score distillation (VSD) [87, 86], operator learning [93] and further

Table 1: Sample quality on class-conditional ImageNet 64x64. Recall metric is also included.

| Method | NFE (↓) | FID (↓) | Recall (↑) |
|---|---|---|---|
| **Diffusion Models & Fast Samplers** | | | |
| ADM [11] | 250 | 2.07 | 0.63 |
| RIN [29] | 1000 | 1.23 | - |
| DisCo-Diff [83] | 623 | 1.22 | - |
| DPM-Solver [50] | 20 | 3.42 | - |
| EDM (Heun) [34] | 79 | 2.44 | 0.68 |
| EDM2 [36] | 63 | 1.33 | 0.68 |
| EDM2 + Autoguidance [35] | 63 | **1.01** | 0.69 |
| **Adversarial & Joint Training** | | | |
| BigGAN-deep [7] | 1 | 4.06 | 0.48 |
| StyleGAN-XL [65] | 1 | 1.52 | - |
| Diff-Instruct [55] | 1 | 5.57 | - |
| DMD [87] | 1 | 2.62 | - |
| DMD2 [86] | 1 | 1.28 | - |
| SiD [98] | 1 | 1.52 | 0.63 |
| CTM [38] | 1 | 1.92 | 0.57 |
| | 2 | 1.73 | - |
| Moment Matching [64] | 1 | 3.00 | - |
| | 2 | 3.86 | - |
| GDD-I [92] | 1 | 1.16 | 0.60 |
| SiDA [97] | 1 | 1.11 | 0.62 |
| **AYF + adv. loss (ours)** | 1 | 1.32 | 0.65 |
| | 2 | 1.17 | 0.65 |
| | 4 | 1.10 | 0.65 |
| | 8 | **1.07** | 0.65 |
| **Diffusion Distillation without Adversarial Objectives** | | | |
| DFNO [93] | 1 | 7.83 | 0.61 |
| PID [74] | 1 | 8.51 | - |
| TRACT [4] | 1 | 7.43 | - |
| BOOT [20] | 1 | 16.3 | 0.36 |
| PD [63] | 1 | 10.70 | 0.65 |
| (reimpl. from Heek et al. [22]) | 2 | 4.70 | - |
| EMD [82] | 1 | 2.20 | 0.59 |
| CD [71] | 1 | 6.20 | - |
| MultiStep-CD [22] | 2 | 1.90 | - |
| sCD [49] | 1 | 2.44 | 0.66* |
| | 2 | 1.66 | 0.66* |
| **AYF (ours)** | 1 | 2.98 | 0.65 |
| | 2 | 1.25 | 0.66 |
| | 4 | 1.15 | 0.66 |
| | 8 | **1.12** | 0.66 |
| **Consistency Training** | | | |
| iCT [69] | 1 | 4.02 | 0.63 |
| iCT-deep [69] | 1 | 3.25 | 0.63 |
| ECT [18] | 1 | 2.77 | - |
| sCT [49] | 1 | 2.04 | - |
| | 2 | 1.48 | - |
| TCM-S [42] | 1 | 2.88 | - |
| | 2 | 2.31 | - |
| TCM-XL [42] | 1 | 2.20 | - |
| | 2 | 1.62 | - |

Table 2: Sample quality on class-conditional ImageNet 512x512. For additional baselines, which AYS all outperforms, please see the Appendix.

| Method | NFE (↓) | FID (↓) | #Params | Gflops | Time (s) |
|---|---|---|---|---|---|
| **Teacher Diffusion Model** | | | | | |
| EDM2-S [36] | 63×2 | 2.23 | 280M | 12852 | 8.31 |
| EDM2-XXL [36] | 63×2 | 1.81 | 1.5B | 69552 | 31.50 |
| EDM2-S + Autoguid. [35] | 63×2 | 1.34 | 280M | 12852 | 8.31 |
| EDM2-XXL + Autoguid. [35] | 63×2 | **1.25** | 1.5B | 69552 | 31.50 |
| **Adversarial & Joint Training** | | | | | |
| SiDA-S [97] (best adv. baseline) | 1 | 1.69 | 280M | 102 | 0.06 |
| **AYF-S + adv. loss (ours)** | 1 | 1.92 | 280M | 102 | 0.06 |
| | 2 | 1.81 | 280M | 204 | 0.12 |
| | 4 | **1.64** | 280M | 408 | 0.24 |
| **Consistency Distillation** | | | | | |
| sCD-S [49] | 1 | 3.07 | 280M | 102 | 0.06 |
| | 2 | 2.50 | 280M | 204 | 0.13 |
| sCD-M [49] | 1 | 2.75 | 498M | 181 | 0.10 |
| | 2 | 2.26 | 498M | 362 | 0.20 |
| sCD-L [49] | 1 | 2.55 | 778M | 282 | 0.14 |
| | 2 | 2.04 | 778M | 564 | 0.28 |
| sCD-XL [49] | 1 | 2.40 | 1.1B | 406 | 0.19 |
| | 2 | 1.93 | 1.1B | 812 | 0.38 |
| sCD-XXL [49] | 1 | 2.28 | 1.5B | 552 | 0.25 |
| | 2 | 1.88 | 1.5B | 1104 | 0.50 |
| **AYF-S (ours)** | 1 | 3.32 | 280M | 102 | 0.06 |
| | 2 | 1.87 | 280M | 204 | 0.12 |
| | 4 | **1.70** | 280M | 408 | 0.24 |
| **Consistency Training** | | | | | |
| sCT-S [49] | 1 | 10.13 | 280M | 102 | 0.06 |
| | 2 | 9.86 | 280M | 204 | 0.13 |
| sCT-M [49] | 1 | 5.84 | 498M | 181 | 0.10 |
| | 2 | 5.53 | 498M | 362 | 0.20 |
| sCT-L [49] | 1 | 5.15 | 778M | 282 | 0.14 |
| | 2 | 4.65 | 778M | 564 | 0.28 |
| sCT-XL [49] | 1 | 4.33 | 1.1B | 406 | 0.19 |
| | 2 | 3.73 | 1.1B | 812 | 0.38 |
| sCT-XXL [49] | 1 | 4.29 | 1.5B | 552 | 0.25 |
| | 2 | 3.76 | 1.5B | 1104 | 0.50 |

techniques [55, 20, 4, 74, 85, 48, 82, 46, 98, 96], many of them relying on adversarial losses, too. However, although popular, adversarial methods introduce training complexities due to their GAN-like objectives. VSD exhibits similar properties and does not work well at high guidance levels. Moreover, these methods can produce samples with limited diversity. For these reasons we avoid such objectives and instead rely on autoguidance to achieve crisp high-quality outputs. Finally, many training-free methods efficiently solve diffusion models' generative differential equations [50, 51, 32, 34, 12, 89, 61], but they are unable to perform well when using $<10$ generation steps.

# 5 Experiments

We train AYF flow maps on ImageNet [10] at resolutions $64 \times 64$ and $512 \times 512$, measuring sample quality using Fréchet Inception Distance (FID) [23], as previous works. We also use our AYF framework to distill FLUX.1 [dev] [41], the best text-to-image diffusion model, using an efficient LoRA [27] framework and reduce sampling steps to just 4. Experiment details explained in the Appendix.

**ImageNet Flow Maps.** We adopt the EDM2 [36] framework, using their small "S" models, and modify network parametrization and time embedding layer as detailed in Sec. 3. Pretrained checkpoints available online are used both as teacher network and as flow map initialization. We incorporate autoguidance into the flow map model by introducing an additional input, $\lambda$, corresponding to the guidance scale [49, 57]. During training, $\lambda$ is uniformly sampled from $[1, 3]$ and

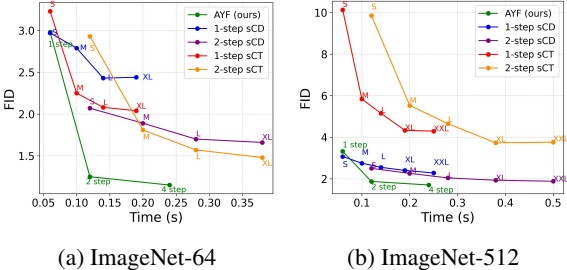

(a) ImageNet-64     (b) ImageNet-512

Figure 6: FID ↓ as function of wall clock time.

applied to the teacher model via autoguidance. At inference, we leverage the $\gamma$-sampling algorithm from [38] for stochastic multistep sampling of flow map models. Results are reported using the optimal $\gamma$ and $\lambda$ values. For ImageNet $512 \times 512$, the teacher and distilled models are in latent space [60].

In Tab. 1 we show ImageNet $64 \times 64$ results, reporting FID and recall scores along with number of neural function evaluations (NFE). Our flow maps achieve the best sample quality among all

non-adversarial few-step methods, given only 2 sampling steps by sacrificing optimal 1-step quality. This is because learning a flow map is a more challenging task compared to only a consistency model. In Tab. 2 we compare AYF against the state-of-the-art consistency model sCD/sCT [49] on ImageNet $512 \times 512$, also reporting total wall sampling clock time, Gflops, and #parameters. We show that although our small-sized model achieves slightly worse one step sample quality, it is on par with the best sCD model at only two steps while using only $18\%$ of the larger models' compute. Increasing the sampling steps to four improves the quality even further while still being over twice as fast as the large 1-step sCM model (wallclock time). We further analyze the performance vs. sampling speed trade-off in Fig. 6, showing that AYF is much more efficient than sCD/sCT (also see Appendix for additional comparison). Autoguidance allows AYF to use a small network and still achieve strong performance and the efficient network results in 2-step or 4-step synthesis still being lightning fast.

**Adversarial finetuning of AYF.** Given a pretrained AYF flow map model, we found that a short finetuning stage using a combination of the EMD objective and an adversarial loss can significantly boost the performance across the board, especially for 1-step generation, with a minimal impact to sample diversity as measured by recall scores. Using this approach, we achieve state-of-the-art performance on few-step generation on ImageNet64 (see Tab. 1). For implementation details, please see Appendix. Additional GAN and diffusion model baselines on ImageNet $512 \times 512$ can be found in the Appendix; AYF outperforms all of them.

**Text-to-Image Flow Maps.** We apply AYF to distill the open-source text-to-image model FLUX.1 [dev] [41] into a few-step generator, finetuning a FLUX.1 base model into a flow map model using LoRA [27] with the objective in Theorem 3.2 for 10,000 steps. This distillation process took approx. four hours on 8 NVIDIA A100 GPUs, which is highly efficient, in contrast to several previous large-scale text-to-image distillation methods.

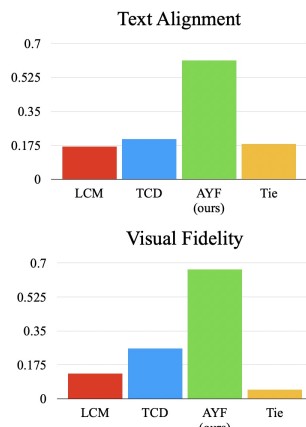

Figure 7: User preferences comparing LoRA-based consistency and flow map models (4-step samples). LCM and TCD use SDXL and AYF uses FLUX.1 [dev] as base model, respectively.

Samples from the model are shown in Fig. 1. We compare to LCM [53, 54] and TCD [94], two consistency-distilled LoRAs trained on top of SDXL [59] without adversarial objectives. To evaluate quality we ran a user study. The results (Fig. 7) show a clear preference for our method. We also provide qualitative comparisons in Fig. 3. Compared to LCM and TCD, our images are more aesthetically pleasing with finer details. We also included FLUX.1 [schnell] [41], a commercially distilled model trained with Latent Adversarial Diffusion Distillation [66]. Our method achieves comparable image quality to the [schnell] model, while requiring only four sampling steps and 32 GPU hours without the use of adversarial losses. In conclusion, AYF achieves state-of-the-art few-step text-to-image generation performance among non-adversarial methods. Detailed ablation studies on different components of AYF (EMD vs. LMD; autoguidance vs. CFG, AYF vs. Shortcut) are presented in the Appendix. Additional qualitative examples of images generated by AYF are shown in the Appendix, too.

## 6    Conclusions

We have presented Align Your Flow (AYF), a novel continuous-time distillation method for training flow maps, which generalizes flow-matching and consistency-based models. Importantly, flow maps remain effective generators across all denoising step counts, unlike standard consistency models; a fact we prove analytically for the first time.

In addition, we use autoguidance to enhance the quality of the teacher model, resulting in an improved distilled student, and an additional boost can be achieved by adversarial finetuning, with minimal impact in sample diversity. We achieve state-of-the-art performance among non-adversarial distillation methods on both ImageNet64 and ImageNet512 generation. Since AYF requires only relatively small neural networks, which further reduces the computational burden and boosts sampling efficiency, even 2-step or 4-step sampling from AYF is as fast or faster than previous single-step generators. We also distill FLUX.1 using an efficient LoRA framework, resulting in state-of-the-art text-to-image generation performance among non-adversarial distillation approaches.

Future work could explore applying AYF to video model distillation or in other domains, for instance in drug discovery for efficient molecule or protein modeling.

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

# Appendix

# A Broader Impact

Generative models have recently seen major advances and one can now routinely synthesize realistic and highly aesthetic images, videos and other modalities. Align Your Flow represents a universal approach to significantly accelerate model sampling from diffusion and flow-based generative models through distillation. Real-time generation can unlock new interactive applications and help artistic and creative expression enabled by generative modeling tools. Moreover, accelerated generation also makes inference workloads more computationally efficient, thereby reducing generative models' energy footprint. Although we evaluated Align Your Flow on image generation benchmark, our proposed methodology is in principle broadly applicable. For instance, one could also imagine future applications in drug discovery, where fast generative models can propose novel molecules rapidly and enable efficient in-silico drug candidate screening. However, generative models like diffusion and flow models, and their distilled versions, can also be used for malicious purposes and, for instance, produce deceptive imagery. Hence, they generally need to be applied with an abundance of caution.

# B Limitations

As shown in our ablation studies (App. G; also see Fig. 8), our AYF models stabilize performance across multi-step sampling. However, this comes at the cost of slightly degraded one-step performance compared to methods focused solely on one-step generation (e.g. sCD [49] or SiDA [97]). Note that adversarial finetuning can mitigate this and improve performance across the board, as shown Fig. 8 and with minimal loss in diversity. We believe that it would be possible to further improve upon that with a more carefully tuned post-training stage, possibly leveraging recent variational score distillation techniques [87, 81]. Additionally, a small gap remains between the AYF model and its multi-step teacher flow-based model, regardless of the number of sampling steps used. This is expected, given that both models have roughly the same number of parameters, yet AYF is trained on a more challenging task and preserves the same noise-to-data mapping. Scaling up model capacity may help bridge this performance gap. Finally, our work focuses on distillation and assumes access to a pretrained flow-based teacher. Prior works [49] suggest that direct consistency training can outperform distillation in certain scenarios. Since our AYF-EMD loss is directly compatible with this paradigm, exploring flow map training without distillation is a promising direction for future work.

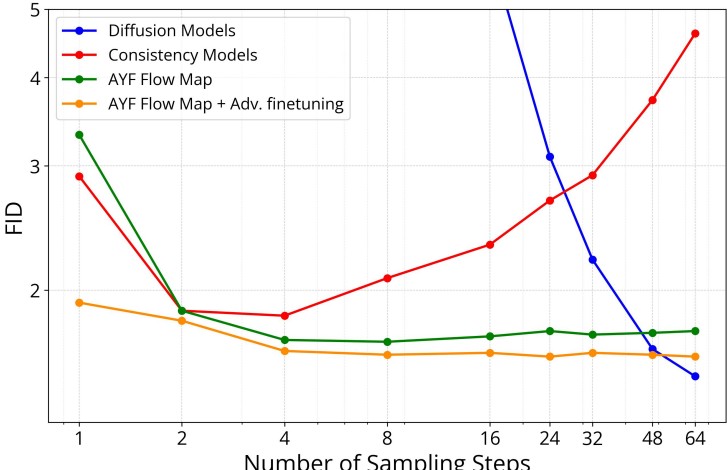

Figure 8: FID versus number of sampling steps on ImageNet 512x512 (lower is better). Diffusion models require dozens of steps to reach good quality, and consistency models deteriorate after only a few, whereas our AYF flow maps maintain low FID across the board. AYF is slightly weaker at a single step generation, but a brief adversarial fine-tuning stage closes this gap and improves quality for all numbers of sampling steps.

# C  Proofs, Derivations and Theoretical Details

## C.1  Theorem: Consistency Models are Flawed Multi-step Generators

**Theorem C.1** (Restated from Theorem 3.1). *Let $p_{data}(\mathbf{x}) = \mathcal{N}(0, c^2\mathbf{I})$ be the data distribution. Then the optimal consistency model $\mathbf{f}^*(\mathbf{x}_t, t)$ is given by*

$$\mathbf{f}^*(\mathbf{x}_t, t) = \frac{c \times \mathbf{x}_t}{\sqrt{t^2 + (1-t)^2 c^2}}.$$

*For any $\epsilon > 0$ consider the following non-optimal consistency model*

$$\mathbf{f}_\epsilon(\mathbf{x}_t, t) = \frac{(c + t \times \epsilon)\mathbf{x}_t}{\sqrt{t^2 + (1-t)^2 c^2}}.$$

*Then, there exists an integer $N$ for which increasing the number of sampling steps during multistep CM sampling beyond $N$ increases the Wasserstein-2 distance of the generated samples to the ground truth distribution.*

*Proof.* In this Gaussian setting, all intermediate noisy distributions $p(\mathbf{x}_t)$, where $\mathbf{x}_t = t \times \mathbf{x}_1 + (1-t) \times \mathbf{x}_0, \mathbf{x}_0 \sim p_{data}, \mathbf{x}_1 \sim \mathcal{N}(0, \mathbf{I})$, and $t \in [0, 1]$, remain Gaussian. As a result, the error of the non-optimal consistency model $\mathbf{f}_\epsilon$ is:

$$\begin{aligned}
\mathbb{E}_{\mathbf{x}_t}[\|\mathbf{f}_\epsilon(\mathbf{x}_t, t) - \mathbf{f}^*(\mathbf{x}_t, t)\|_2^2] &= \mathbb{E}_{\mathbf{x}_t}[t^2\epsilon^2\|\frac{\mathbf{x}_t}{\sqrt{t^2 + (1-t)^2 c^2}}\|_2^2] \\
&= \frac{t^2\epsilon^2}{t^2 + (1-t)^2 c^2}\mathbb{E}_{\mathbf{x}_0, \mathbf{x}_1}[\|t\,\mathbf{x}_1 + (1-t)\,\mathbf{x}_0\|_2^2] = t^2\epsilon^2 \le \epsilon^2,
\end{aligned} \tag{6}$$

and satisfies the error bound in Theorem 3.1 by choosing a small enough $\epsilon$ that satisfies $\epsilon^2 < \delta$.

Due to the Gaussian setting, both the optimal velocity from flow matching, $\mathbf{v}^*(\mathbf{x}_t, t)$, and the optimal denoiser, $\mathbf{D}^*(\mathbf{x}_t, t)$, have closed-form solutions.

We first derive the denoiser. Following the framework of [34], let $p(\mathbf{x}; \sigma)$ denote the distribution obtained by adding independent Gaussian noise with standard deviation $\sigma$ to the data. Then,

$$p(\mathbf{x}; \sigma) = \mathcal{N}(0, (c^2 + \sigma^2)\mathbf{I}) \Rightarrow \nabla_\mathbf{x} \log p(\mathbf{x}; \sigma) = \frac{-\mathbf{x}}{c^2 + \sigma^2} \Rightarrow \mathbf{D}^*(\mathbf{x}, \sigma) = \frac{c^2}{c^2 + \sigma^2}\mathbf{x},$$

where the final step follows from the identity

$$\nabla_\mathbf{x} \log p(\mathbf{x}; \sigma) = (\mathbf{D}(\mathbf{x}; \sigma) - \mathbf{x})/\sigma^2$$

from [34]. Using this, the optimal velocity from flow matching, $\mathbf{v}^*(\mathbf{x}_t, t)$, can be computed as

$$\mathbf{v}^*(\mathbf{x}_t, t) = \mathbb{E}[\mathbf{x}_1 - \mathbf{x}_0 \mid \mathbf{x}_t] = \mathbb{E}\left[\frac{\mathbf{x}_t - \mathbf{x}_0}{t} \mid \mathbf{x}_t\right] = \frac{\mathbf{x}_t}{t} - \frac{1}{t}\mathbb{E}[\mathbf{x}_0 \mid \mathbf{x}_t].$$

Substituting $\mathbf{D}^*\left(\frac{\mathbf{x}_t}{1-t}, \frac{t}{1-t}\right)$ for $\mathbb{E}[\mathbf{x}_0 \mid \mathbf{x}_t]$, we obtain

$$\mathbf{v}^*(\mathbf{x}_t, t) = \frac{\mathbf{x}_t}{t} - \frac{1}{t}\mathbf{D}^*\left(\frac{\mathbf{x}_t}{1-t}, \frac{t}{1-t}\right).$$

Using the closed-form expression for $\mathbf{D}^*(\mathbf{x}, \sigma)$,

$$\mathbf{v}^*(\mathbf{x}_t, t) = \frac{\mathbf{x}_t}{t} - \frac{1}{t} \cdot \frac{c^2}{c^2 + \left(\frac{t}{1-t}\right)^2} \cdot \frac{\mathbf{x}_t}{1-t}.$$

Simplifying further,

$$\mathbf{v}^*(\mathbf{x}_t, t) = \frac{\mathbf{x}_t}{t}\left(1 - \frac{c^2(1-t)}{c^2(1-t)^2 + t^2}\right) = \left(\frac{t - c^2(1-t)}{t^2 + (1-t)^2 c^2}\right)\mathbf{x}_t.$$

Integrating this velocity along the PF-ODE,

$$d\mathbf{x}_t = \mathbf{v}^*(\mathbf{x}_t, t)dt,$$

from $t$ to 0 yields the optimal consistency model:

$$\mathbf{f}^*(\mathbf{x}_t, t) = \frac{c \cdot \mathbf{x}_t}{\sqrt{t^2 + (1-t)^2 c^2}}.$$

Consider the non-optimal consistency model

$$\mathbf{f}_\epsilon(\mathbf{x}_t, t) = \frac{(c + t \times \epsilon)\mathbf{x}_t}{\sqrt{t^2 + (1-t)^2 c^2}}. \tag{7}$$

Note that $\mathbf{f}_\epsilon(\mathbf{x}, 0) = \mathbf{x}$ satisfies the boundary condition and is a valid consistency model.

Let us analyze a single multistep transition from timestep $t$ to timestep $s$. This process consists of two steps: 1. The noise is removed by predicting the clean data using the consistency model, yielding $\mathbf{x}'_0 = \mathbf{f}_\epsilon(\mathbf{x}_t, t)$. 2. The estimated clean data $\mathbf{x}'_0$ is then noised back to timestep $s$ using

$$\mathbf{x}_s = s \times \mathbf{z} + (1-s) \times \mathbf{x}'_0, \quad \mathbf{z} \sim \mathcal{N}(\mathbf{0}, \mathbf{I}).$$

Assuming that $\mathbf{x}_t \sim \mathcal{N}(\mathbf{0}, \sigma_t^2 \mathbf{I})$, $\mathbf{x}_s$ will also follow an isotropic zero-mean Gaussian distribution and is obtained as follows:

$$\mathbf{x}_s = s \times \mathbf{z} + (1-s) \times \frac{(c + t\epsilon)\mathbf{x}_t}{\sqrt{t^2 + (1-t)^2 c^2}}.$$

Therefore, the variance of $x_s$ is given by

$$\mathrm{Var}(\mathbf{x}_s) = s^2 + \frac{(1-s)^2(c+t\epsilon)^2}{t^2 + (1-t)^2 c^2} \mathrm{Var}(\mathbf{x}_t). \tag{8}$$

Using this recurrence relation, we can compute the variance of the distribution obtained by running multistep CM sampling on a uniform $n$-step schedule:

$$[0, \frac{1}{n}, \ldots, \frac{n-1}{n}, 1].$$

Since both the ground truth distribution and the distribution of $x_0$ derived by $n$-step sampling are isotropic zero-mean Gaussians, the Wasserstein-2 distance between them has the following closed form solution:

$$W_2(p(\mathbf{x}_0); p_{data}(\mathbf{x})) = (\sqrt{\mathrm{Var}(\mathbf{x}_0)} - c)^2$$

Let $\mathrm{Var}(s) := \mathrm{Var}(\mathbf{x}_s)$ for convenience. We will show that as $n \to \infty$, the variance $\mathrm{Var}(0)$ when computed via the recurrence defined in Eq. (8) on the uniform $n$ step schedule diverges, i.e. $\mathrm{Var}(0) \to \infty$. This means performing multi-step sampling with the consistency model will result in accumulated errors beyond a certain point.

Define

$$h(s) := \frac{\mathrm{Var}(s)}{s^2 + (1-s)^2 c^2}.$$

Plugging this into Eq. (8) gives:

$$h(s) = \frac{s^2 + (1-s)^2(c+t\epsilon)^2 h(t)}{s^2 + (1-s)^2 c^2}. \tag{9}$$

We know $h(1) = 1$ and $h(0) = \mathrm{Var}(0)/c^2$. So, it's enough to show that $h(0) \to \infty$ as $n \to \infty$.

It's easy to see by induction that $h(t) \geq 1$. Define $g(t) := h(t) - 1$ to measure how much it grows. Then:

$$\begin{aligned} g(s) &= \frac{(1-s)^2(c+t\epsilon)^2(g(t)+1) - (1-s)^2 c^2}{s^2 + (1-s)^2 c^2} \\ &= \frac{(1-s)^2}{s^2 + (1-s)^2 c^2}\left(2ct\epsilon + t^2\epsilon^2 + (c+t\epsilon)^2 g(t)\right) \\ &\geq \frac{(1-s)^2}{s^2 + (1-s)^2 c^2}(2ct\epsilon + c^2 g(t)). \end{aligned} \tag{10}$$

All terms are positive, so let's lower bound $g(t)$ by defining a simpler sequence:

$$g'(s) := \frac{(1-s)^2}{s^2 + (1-s)^2 c^2}(2ct\epsilon + c^2 g'(t)). \tag{11}$$

Clearly $g(s) \geq g'(s)$, so it's enough to show $g'(0) \to \infty$ as $n \to \infty$.

Now define:

$$r(s) := \frac{g'(s)}{2c\epsilon}.$$

Then the recurrence becomes:

$$r(s) = \frac{(1-s)^2}{s^2 + (1-s)^2 c^2}(t + c^2 r(t)), \tag{12}$$

with $r(1) = 0$ and $r(0) = g'(0)/(2c\epsilon)$. So we just need to show $r(0) \to \infty$ as $n \to \infty$.

To analyze this, fix $s$ and $t$, and consider the function:

$$f_{s,t}(x) := \frac{(1-s)^2}{s^2 + (1-s)^2 c^2}(t + c^2 x). \tag{13}$$

This is an affine map with a unique fixed point:

$$o(s,t) = \frac{t(1-s)^2}{s^2}. \tag{14}$$

Subtracting $o(s,t)$ from both sides, we get:

$$f(x) - o(s,t) = \lambda(s)(x - o(s,t)). \tag{15}$$

where:

$$\lambda(s) := \frac{1}{(\frac{s}{(1-s)c})^2 + 1} < 1.$$

So $f_{s,t}(x)$ pulls every point toward its fixed point $o(s,t)$, with a pull factor of $\lambda(s)$. As $s \to 0$, since $o(s,t) \geq \frac{(1-s)^2}{s}$, the fixed point $o(s,t) \to \infty$, and the pull factor $\lambda(s)$ approaches 1, meaning the pull gets weaker.

This means the recurrence in Eq. (12) is applying a sequence of weaker and weaker pulls toward bigger and bigger targets. To prove $r(0) \to \infty$, we now proceed by contradiction. Assume there exists some $\delta > 0$ and an infinite sequence $n_1 < n_2 < \ldots$ such that $r(0) < \delta$ when using the schedule $[0, \frac{1}{n_i}, \ldots, 1]$.

Since $o(s,t) \geq \frac{(1-s)^2}{s}$ and the function $\frac{(1-x)^2}{x} \to \infty$ as $x \to 0$, we can pick $t^* \in [0,1]$ such that for all $s \in [0, t^*]$:

$$\frac{(1-t^*)^2}{t^*} = 2\delta \Rightarrow o(s,t) \geq \frac{(1-s)^2}{s} \geq 2\delta. \tag{16}$$

So every fixed point in $[0, t^*]$ is at least $2\delta$.

Now we split the problem into two cases:

**Case 1**: There exists some $s \in [0, t^*]$ where $r(s) \geq \delta$. Since all future pulls are toward values $> 2\delta$, $r(\cdot)$ will stay above $\delta$, so $r(0) \geq \delta$—a contradiction.

**Case 2**: For all $s \in [0, t^*]$, we have $r(s) < \delta$. Pick any $s^* \in (0, t^*)$. Let $\ell := \lambda(s^*) < 1$, which is the maximum pull factor (corresponding to the weakest pull) on $[s^*, t^*]$. Then for $s^* \leq s < t \leq t^*$:

$$\begin{aligned}
2\delta - r(s) &= (o(s,t) - r(s)) &&+ (2\delta - o(s,t)) \\
&= (o(s,t) - f_{s,t}(r(t))) &&+ (2\delta - o(s,t)) \\
&= \lambda(s)(o(s,t) - r(t)) &&+ (2\delta - o(s,t)) \\
&\leq \lambda(s^*)(o(s,t) - r(t)) &&+ (2\delta - o(s,t)) \\
&= \ell(o(s,t) - r(t)) &&+ (2\delta - o(s,t)) \\
&= \ell(2\delta - r(t)) &&+ (1-\ell)(2\delta - o(s,t)) \\
&\leq \ell(2\delta - r(t)).
\end{aligned}$$

Applying this inequality $M$ times (where $M$ is the number of steps between $s^*$ and $t^*$ on the schedule), we get:

$$\delta \le 2\delta - r(s^*) \le 2\delta\ell^M. \tag{17}$$

If $M$ is large enough so that $2\delta\ell^M < \delta$, we get a contradiction. This happens when $n > M/(t^* - s^*)$.

Since both cases lead to a contradiction, we conclude that $r(0) \to \infty$ as $n \to \infty$, completing the proof.

$\square$

**Intuition:** When doing multi-step sampling with consistency models we first perform denoising by removing all noise from a noisy image to obtain a clean one, and then re-add a smaller amount of noise. But because the model is not perfect, the denoised image drifts slightly off the true data manifold. When noise is added back in, the resulting image is now slightly off the noisy manifold the model was trained on. This mismatch compounds: each denoising step starts from a slightly worse input, pushing the sample further off-manifold over time. As a result, errors accumulate with more sampling steps, leading to degrading image quality beyond a certain point.

## C.2 Theorems: Flow Map Objectives

### C.2.1 AYF-EMD

**Theorem C.2** (Restated from Theorem 3.2). *Let $\mathbf{f}_\theta(\mathbf{x}_t, t, s)$ be the flow map and $\mathbf{v}_\phi(\mathbf{x}_t, t)$ denote the pretrained flow matching model. Define $\theta^- = stopgrad(\theta)$. Let $t$ and $t' = t + \epsilon(s - t)$ denote two adjacent starting timesteps for a small $\epsilon > 0$. Note that $t' \in [t, s]$ is derived by taking a small step from $t$ towards $s$. Consider the following consistency loss function defined*

$$\mathcal{L}_{EMD}^\epsilon(\theta) = \mathbb{E}_{\mathbf{x}_t, t, s}\left[w(t, s)\|\mathbf{f}_\theta(\mathbf{x}_t, t, s) - \mathbf{f}_{\theta^-}(\mathbf{x}_{t'}, t', s)\|_2^2\right], \tag{18}$$

*where $\mathbf{x}_{t'}$ is obtained by using a 1-step euler solver on the PF-ODE, i.e. $d\mathbf{x}_t = \mathbf{v}_\phi(\mathbf{x}_t, t)dt$, from $t$ to $t'$. In the limit as $\epsilon \to 0$, the gradient of this objective with respect to $\theta$ converges to:*

$$\lim_{\epsilon \to 0}(1/\epsilon)\nabla_\theta \mathcal{L}_{EMD}^\epsilon(\theta) = \nabla_\theta \mathbb{E}_{\mathbf{x}_t, t, s}\left[w(t, s)(t - s)\mathbf{f}_\theta^\top(\mathbf{x}_t, t, s)\frac{d\mathbf{f}_{\theta^-}(\mathbf{x}_t, t, s)}{dt}\right]. \tag{19}$$

*Proof.* For this proof, we follow a similar logic as the proof of Theorem 5 in [71]. We start by computing the gradient of Eq. (18) with respect to $\theta$ and simplifying the result to obtain

$$\frac{1}{\epsilon}\nabla_\theta \mathcal{L}_{EMD}^\epsilon(\theta) = \frac{1}{\epsilon}\nabla_\theta \mathbb{E}_{\mathbf{x}_t, t, s}\left[w(t, s)\|\mathbf{f}_\theta(\mathbf{x}_t, t, s) - \mathbf{f}_{\theta^-}(\mathbf{x}_{t'}, t', s)\|_2^2\right]$$

$$= \frac{1}{\epsilon}\mathbb{E}_{\mathbf{x}_t, t, s}\left[w(t, s)\nabla_\theta \mathbf{f}_\theta(\mathbf{x}_t, t, s)^\top\left(\mathbf{f}_\theta(\mathbf{x}_t, t, s) - \mathbf{f}_{\theta^-}(\mathbf{x}_{t'}, t', s)\right)\right]$$

$$= \frac{1}{\epsilon}\mathbb{E}_{\mathbf{x}_t, t, s}\left[w(t, s)\nabla_\theta \mathbf{f}_\theta(\mathbf{x}_t, t, s)^\top\left(\mathbf{f}_\theta(\mathbf{x}_t, t, s) - \left(\mathbf{f}_{\theta^-}(\mathbf{x}_t, t, s) + \frac{\partial \mathbf{f}_{\theta^-}}{\partial \mathbf{x}}(\mathbf{x}_{t'} - \mathbf{x}_t) + \right.\right.\right.$$
$$\left.\left.\left.\frac{\partial \mathbf{f}_{\theta^-}}{\partial t}(t' - t) + O(\epsilon^2)\right)\right)\right]$$

$$= \frac{1}{\epsilon}\mathbb{E}_{\mathbf{x}_t, t, s}\left[w(t, s)\nabla_\theta \mathbf{f}_\theta(\mathbf{x}_t, t, s)^\top\left(-\left(\frac{\partial \mathbf{f}_{\theta^-}}{\partial \mathbf{x}}(\epsilon(s - t) \cdot \mathbf{v}_\phi(\mathbf{x}_t, t)) + \frac{\partial \mathbf{f}_{\theta^-}}{\partial t}(\epsilon(s - t))\right)\right)\right] + O(\epsilon)$$

$$= \mathbb{E}_{\mathbf{x}_t, t, s}\left[w(t, s)(t - s)\nabla_\theta \mathbf{f}_\theta(\mathbf{x}_t, t, s)^\top\left(\frac{\partial \mathbf{f}_{\theta^-}}{\partial \mathbf{x}}\mathbf{v}_\phi(\mathbf{x}_t, t) + \frac{\partial \mathbf{f}_{\theta^-}}{\partial t}\right)\right] + O(\epsilon)$$

$$= \mathbb{E}_{\mathbf{x}_t, t, s}\left[w(t, s)(t - s)\nabla_\theta \mathbf{f}_\theta(\mathbf{x}_t, t, s)^\top\left(\frac{d\mathbf{f}_{\theta^-}(\mathbf{x}_t, t, s)}{dt}\right)\right] + O(\epsilon),$$

Taking the limit of both sides as $\epsilon \to 0$ completes the proof. $\square$

**Corollary C.3.** *Theorem 3.2 assumes that the step size is proportional to the interval length, i.e. $|t - t'| \propto |t - s|$, leading to the introduction of a $(t - s)$ term in the weighting function. This can be eliminated by using $t' = t + sign(s - t) \times \epsilon$ and leads to the following objective:*

$$\nabla_\theta \mathbb{E}_{\mathbf{x}_t, t, s}\left[w(t, s)sign(t - s)\mathbf{f}_\theta^\top(\mathbf{x}_t, t, s)\frac{d\mathbf{f}_{\theta^-}(\mathbf{x}_t, t, s)}{dt}\right]. \tag{20}$$

### C.2.2 AYF-LMD

**Theorem C.4** (Restated from Theorem 3.3). *Let $\mathbf{f}_\theta(\mathbf{x}_t, t, s)$ be the flow map, $\mathbf{v}_\phi(\mathbf{x}_t, t)$ be the pretrained flow matching model, and define $\theta^- = stopgrad(\theta)$. Let $s$ and $s' = s + \epsilon(t - s)$ denote two adjacent ending timesteps for a small $\epsilon > 0$. Note that $s' \in [t, s]$ is obtained by taking a small step from $s$ towards $t$. Consider the following consistency loss function defined*

$$\mathcal{L}_{LMD}^\epsilon(\theta) = \mathbb{E}_{\mathbf{x}_t, t, s}\left[w(t, s)\|\mathbf{f}_\theta(\mathbf{x}_t, t, s) - ODE_{s' \to s}(\mathbf{f}_{\theta^-}(\mathbf{x}_t, t, s'))\|_2^2\right], \quad (21)$$

*where $ODE_{s' \to s}(x)$ refers to running a 1-step euler solver on the PF-ODE, i.e. $d\mathbf{x}_t = \mathbf{v}_\phi(\mathbf{x}_t, t)dt$, starting from $x$ at timestep $s'$ to timestep $s$. In the limit as $\epsilon \to 0$, the gradient of this objective with respect to $\theta$ converges to:*

$$\lim_{\epsilon \to 0}(1/\epsilon)\nabla_\theta \mathcal{L}_{LMD}^\epsilon(\theta) = \nabla_\theta \mathbb{E}_{\mathbf{x}_t, t, s}\left[w(t, s)(s - t)\mathbf{f}_\theta^\top(\mathbf{x}_t, t, s)\left(\frac{d\mathbf{f}_{\theta^-}(\mathbf{x}_t, t, s)}{ds} - \mathbf{v}_\phi(\mathbf{f}_{\theta^-}(\mathbf{x}_t, t, s), s)\right)\right]. \quad (22)$$

*Proof.* We start by computing the gradient of Eq. (21) with respect to $\theta$ and simplifying the results to obtain

$$\frac{1}{\epsilon}\nabla_\theta \mathcal{L}_{LMD}^\epsilon(\theta) = \frac{1}{\epsilon}\nabla_\theta \mathbb{E}_{\mathbf{x}_t, t, s}\left[w(t, s)\|\mathbf{f}_\theta(\mathbf{x}_t, t, s) - ODE_{s' \to s}(\mathbf{f}_{\theta^-}(\mathbf{x}_t, t, s'))\|_2^2\right]$$

$$= \frac{1}{\epsilon}\mathbb{E}_{\mathbf{x}_t, t, s}\left[w(t, s)\nabla_\theta \mathbf{f}_\theta(\mathbf{x}_t, t, s)^\top (\mathbf{f}_\theta(\mathbf{x}_t, t, s) - ODE_{s' \to s}(\mathbf{f}_{\theta^-}(\mathbf{x}_t, t, s')))\right]$$

$$= \frac{1}{\epsilon}\mathbb{E}_{\mathbf{x}_t, t, s}\left[w(t, s)\nabla_\theta \mathbf{f}_\theta(\mathbf{x}_t, t, s)^\top (\mathbf{f}_\theta(\mathbf{x}_t, t, s) - (\mathbf{f}_{\theta^-}(\mathbf{x}_t, t, s') + (s - s') \cdot \mathbf{v}_\phi(\mathbf{f}_{\theta^-}(\mathbf{x}_t, t, s'), s')))\right]$$

$$= \frac{1}{\epsilon}\mathbb{E}_{\mathbf{x}_t, t, s}\left[w(t, s)\nabla_\theta \mathbf{f}_\theta(\mathbf{x}_t, t, s)^\top \left(\mathbf{f}_\theta(\mathbf{x}_t, t, s) - \left(\mathbf{f}_{\theta^-}(\mathbf{x}_t, t, s) + \frac{\partial \mathbf{f}_{\theta^-}(\mathbf{x}_t, t, s)}{\partial s}(s' - s) + O(\epsilon^2)\right.\right.\right.$$
$$\left.\left.\left. + (s - s') \cdot (\mathbf{v}_\phi(\mathbf{f}_{\theta^-}(\mathbf{x}_t, t, s), s) + O(\epsilon)))\right]\right.$$

$$= \frac{1}{\epsilon}\mathbb{E}_{\mathbf{x}_t, t, s}\left[w(t, s)\nabla_\theta \mathbf{f}_\theta(\mathbf{x}_t, t, s)^\top \left(-\left(\frac{\partial \mathbf{f}_{\theta^-}(\mathbf{x}_t, t, s)}{\partial s}(s' - s) + (s - s')\mathbf{v}_\phi(\mathbf{f}_{\theta^-}(\mathbf{x}_t, t, s), s)\right)\right)\right] + O(\epsilon)$$

$$= \mathbb{E}_{\mathbf{x}_t, t, s}\left[w(t, s)(s - t)\nabla_\theta \mathbf{f}_\theta(\mathbf{x}_t, t, s)^\top \left(\frac{\partial \mathbf{f}_{\theta^-}(\mathbf{x}_t, t, s)}{\partial s} - \mathbf{v}_\phi(\mathbf{f}_{\theta^-}(\mathbf{x}_t, t, s), s)\right)\right] + O(\epsilon)$$

Taking the limit of both sides as $\epsilon \to 0$ completes the proof. $\qquad\square$

**Corollary C.5.** *Theorem 3.3 assumes that the step size is proportional to the interval length, i.e. $|s - s'| \propto |t - s|$, leading to the introduction of a $(t - s)$ term in the weighting function. This can be eliminated by using $s' = s + sign(t - s) \times \epsilon$ which leads to the following objective:*

$$\nabla_\theta \mathbb{E}_{\mathbf{x}_t, t, s}\left[w(t, s)sign(s - t)\mathbf{f}_\theta^\top(\mathbf{x}_t, t, s)\left(\frac{d\mathbf{f}_{\theta^-}(\mathbf{x}_t, t, s)}{ds} - \mathbf{v}_\phi(\mathbf{f}_{\theta^-}(\mathbf{x}_t, t, s), s)\right)\right]. \quad (23)$$

### C.3 Derivation: Tangent Warmup as Linearity Regularization

Here, we derive Eq. (5). This equation shows the equivalence between the tangent warmup technique and a regularization term on flow maps that encourages linearity.

$$\nabla_\theta \left[sign(t - s)\mathbf{f}_\theta^\top(\mathbf{x}_t, t, s) \times \left(\frac{d\mathbf{x}_t}{dt} - \mathbf{F}_{\theta^-}(\mathbf{x}_t, t, s)\right)\right]$$

$$= \nabla_\theta \left[sign(t - s)(\mathbf{x}_t + (s - t)\mathbf{F}_\theta(\mathbf{x}_t, t, s)) \times (\mathbf{v}_\phi(\mathbf{x}_t, t) - \mathbf{F}_{\theta^-}(\mathbf{x}_t, t, s))\right]$$

$$= \nabla_\theta \left[-|t - s| \times \mathbf{F}_\theta(\mathbf{x}_t, t, s) \times (\mathbf{v}_\phi(\mathbf{x}_t, t) - \mathbf{F}_{\theta^-}(\mathbf{x}_t, t, s))\right]$$

$$\overset{(i)}{\propto} \nabla_\theta \left[\mathbf{F}_\theta(\mathbf{x}_t, t, s) \times (\mathbf{F}_{\theta^-}(\mathbf{x}_t, t, s) - \mathbf{v}_\phi(\mathbf{x}_t, t))\right] \qquad (24)$$

$$= (\nabla_\theta \mathbf{F}_\theta(\mathbf{x}_t, t, s)) \times (\mathbf{F}_{\theta^-}(\mathbf{x}_t, t, s) - \mathbf{v}_\phi(\mathbf{x}_t, t))$$

$$\overset{(ii)}{=} (\nabla_\theta \mathbf{F}_\theta(\mathbf{x}_t, t, s)) \times (\mathbf{F}_\theta(\mathbf{x}_t, t, s) - \mathbf{v}_\phi(\mathbf{x}_t, t))$$

$$\overset{(iii)}{=} (\nabla_\theta[\mathbf{F}_\theta(\mathbf{x}_t, t, s) - \mathbf{v}_\phi(\mathbf{x}_t, t)]) \times (\mathbf{F}_\theta(\mathbf{x}_t, t, s) - \mathbf{v}_\phi(\mathbf{x}_t, t))$$

$$= 0.5 \times \nabla_\theta \|\mathbf{F}_\theta(\mathbf{x}_t, t, s) - \mathbf{v}_\phi(\mathbf{x}_t, t, s)\|_2^2,$$

where (i) is because we discard $|t - s| \geq 0$, which doesn't change the gradient direction, (ii) is because $\mathbf{F}_\theta = \mathbf{F}_{\theta^-}$, and (iii) is because $\nabla_\theta \mathbf{v}_\phi(\mathbf{x}_t, t) = 0$.

# D    Connections to Existing Methods

In this section, we highlight the connections between AYF and existing methods. We show how AYF generalizes several prior approaches and discuss its relationship to recent concurrent works. See Fig. 9 for a schematic overview of these connections. In the following subsections, we will derive these relationships in detail.

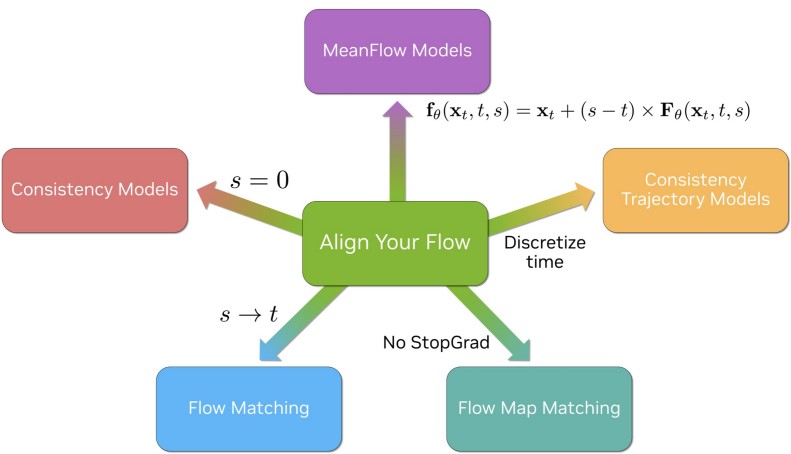

Figure 9: AYF can be seen a a generalization of many prior works such as Flow Matching [45], Continuous-time Consistency Models [49], Flow Map Matching [5], Consistency Trajectory Distillation [94], and the concurrent MeanFlow Models [19].

## D.1    Flow Matching

We show that the AYF-EMD objective introduced in Theorem 3.2 generalizes the standard flow matching objective. In particular, we prove that the gradient of the AYF-EMD objective reduces to the flow matching gradient in the limit as $s \to t$, up to a constant factor.

Recall the flow map parameterization:

$$\mathbf{f}_\theta(\mathbf{x}_t, t, s) = \mathbf{x}_t + (s - t)\,\mathbf{F}_\theta(\mathbf{x}_t, t, s). \tag{25}$$

Substituting into the AYF-EMD objective gives:

$$
\begin{aligned}
&\nabla_\theta \mathbb{E}_{\mathbf{x}_t, t, s} \left[ w'(t, s)\,\mathrm{sign}(t - s)\,\mathbf{f}_\theta^\top(\mathbf{x}_t, t, s)\,\frac{\mathrm{d}\mathbf{f}_{\theta^-}(\mathbf{x}_t, t, s)}{\mathrm{d}t} \right] \\
&= \nabla_\theta \mathbb{E}_{\mathbf{x}_t, t, s} \left[ w'(t, s)\,\mathrm{sign}(t - s)\,(\mathbf{x}_t + (s - t)\mathbf{F}_\theta)^\top \left( \frac{\mathrm{d}\mathbf{x}_t}{\mathrm{d}t} - \mathbf{F}_{\theta^-} + (s - t)\frac{\mathrm{d}\mathbf{F}_{\theta^-}}{\mathrm{d}t} \right) \right] \\
&= \nabla_\theta \mathbb{E}_{\mathbf{x}_t, t, s} \left[ -w'(t, s)\,|t - s|\,\mathbf{F}_\theta^\top \left( \frac{\mathrm{d}\mathbf{x}_t}{\mathrm{d}t} - \mathbf{F}_{\theta^-} + (s - t)\frac{\mathrm{d}\mathbf{F}_{\theta^-}}{\mathrm{d}t} \right) \right] \\
&\propto \nabla_\theta \mathbb{E}_{\mathbf{x}_t, t, s} \left[ \mathbf{F}_\theta^\top \left( \mathbf{F}_{\theta^-} - \frac{\mathrm{d}\mathbf{x}_t}{\mathrm{d}t} + (t - s)\frac{\mathrm{d}\mathbf{F}_{\theta^-}}{\mathrm{d}t} \right) \right].
\end{aligned}
\tag{26}
$$

Taking the limit as $s \to t$ gives:

$$\nabla_\theta \mathbb{E}_{\mathbf{x}_t, t} \left[ \mathbf{F}_\theta^\top(\mathbf{x}_t, t, t) \left( \mathbf{F}_{\theta^-}(\mathbf{x}_t, t, t) - \frac{\mathrm{d}\mathbf{x}_t}{\mathrm{d}t} \right) \right] = \nabla_\theta \mathbb{E}_{\mathbf{x}_t, t} \left[ \left\| \mathbf{F}_\theta(\mathbf{x}_t, t, t) - \frac{\mathrm{d}\mathbf{x}_t}{\mathrm{d}t} \right\|_2^2 \right], \tag{27}$$

which is exactly the standard flow matching loss.

### D.2 Continuous-Time Consistency Models

We show that the AYF-EMD objective introduced in Theorem 3.2 also generalizes the continuous-time consistency model objective. Specifically, we prove that the AYF-EMD objective reduces to the continuous CM objective when $s = 0$, up to a constant factor.

Consider the AYF-EMD objective:

$$\nabla_\theta \mathbb{E}_{\mathbf{x}_t,t,s} \left[ w'(t,s) \operatorname{sign}(t-s) \, \mathbf{f}_\theta^\top(\mathbf{x}_t,t,s) \frac{d\mathbf{f}_{\theta^-}(\mathbf{x}_t,t,s)}{dt} \right]. \tag{28}$$

Setting $s = 0$ gives:

$$\nabla_\theta \mathbb{E}_{\mathbf{x}_t,t} \left[ w'(t,0) \, \mathbf{f}_\theta^\top(\mathbf{x}_t,t,0) \frac{d\mathbf{f}_{\theta^-}(\mathbf{x}_t,t,0)}{dt} \right], \tag{29}$$

since $\operatorname{sign}(t) = 1$ for $t \in [0,1]$. Noting that $\mathbf{f}_\theta(\mathbf{x}_t,t,0)$ corresponds to the CM prediction at noise level $t$, this recovers the standard continuous-time CM objective [49].

### D.3 Consistency Trajectory Models

In this part, we discuss the connection to Consistency Trajectory Models (CTMs) [38, 94]. Recall from the derivation of the AYF-EMD objective in App. C.2.1 that its gradient corresponds to the continuous-time limit of a discrete consistency loss. Interestingly, TCD uses this exact same discrete consistency loss to train its flow map, with a fixed discretization schedule. Therefore, the TCD objective can be seen as a discrete approximation of the AYF-EMD loss.

### D.4 Flow Map Matching

In this part, we highlight the similarities and differences between our AYF objectives and the losses proposed by Boffi et al. [5].

#### D.4.1 EMD

Here, we will show the connection between our AYF-EMD objective and the EMD loss proposed by Boffi et al. [5]. Recall the EMD loss from their work:

$$\nabla_\theta \mathbb{E}_{\mathbf{x}_t,t,s} \left[ w(t,s) \left\| \partial_t \mathbf{f}_\theta(\mathbf{x}_t,t,s) + \nabla_\mathbf{x} \mathbf{f}_\theta(\mathbf{x}_t,t,s) \cdot \frac{d\mathbf{x}_t}{dt} \right\|_2^2 \right] = \nabla_\theta \mathbb{E}_{\mathbf{x}_t,t,s} \left[ w(t,s) \left\| \frac{d\mathbf{f}_\theta(\mathbf{x}_t,t,s)}{dt} \right\|_2^2 \right]. \tag{30}$$

This loss can be derived in a similar fashion as our AYF-EMD loss by introducing a modification to Eq. (18). Specifically, by replacing the second term $\mathbf{f}_{\theta^-}(\mathbf{x}_{t'},t',s)$ with $\mathbf{f}_\theta(\mathbf{x}_{t'},t',s)$ and allowing gradients to flow through both terms, we recover the EMD loss from Boffi et al. [5].

Concretely, we can show that in the limit as $\epsilon \to 0$, the gradient of this objective with respect to $\theta$ converges to:

$$\lim_{\epsilon \to 0} \frac{1}{\epsilon^2} \nabla_\theta \mathbb{E}_{\mathbf{x}_t,t,s} \left[ w(t,s) \| \mathbf{f}_\theta(\mathbf{x}_t,t,s) - \mathbf{f}_\theta(\mathbf{x}_{t'},t',s) \|_2^2 \right] = \nabla_\theta \mathbb{E}_{\mathbf{x}_t,t,s} \left[ w'(t,s) \left\| \frac{d\mathbf{f}_\theta(\mathbf{x}_t,t,s)}{dt} \right\|_2^2 \right], \tag{31}$$

where $w'(t,s) = w(t,s) \times |t-s|^2$.

The proof is as follows:

$$\frac{1}{\epsilon^2} \nabla_\theta \mathbb{E}_{\mathbf{x}_t,t,s} \left[ w(t,s) \| \mathbf{f}_\theta(\mathbf{x}_t,t,s) - \mathbf{f}_\theta(\mathbf{x}_{t'},t',s) \|_2^2 \right]$$

$$= \frac{1}{\epsilon^2} \nabla_\theta \mathbb{E}_{\mathbf{x}_t,t,s} \left[ w(t,s) \left\| \epsilon(t-s) \frac{d\mathbf{f}_\theta(\mathbf{x}_t,t,s)}{dt} + O(\epsilon^2) \right\|_2^2 \right]$$

$$= \nabla_\theta \mathbb{E}_{\mathbf{x}_t,t,s} \left[ w(t,s) \left\| (t-s) \frac{d\mathbf{f}_\theta(\mathbf{x}_t,t,s)}{dt} + O(\epsilon) \right\|_2^2 \right]$$

$$= \nabla_\theta \mathbb{E}_{\mathbf{x}_t,t,s} \left[ w'(t,s) \left\| \frac{d\mathbf{f}_\theta(\mathbf{x}_t,t,s)}{dt} \right\|_2^2 \right] + O(\epsilon),$$

which converges to the gradient of the EMD loss as $\epsilon \to 0$. Note that when deriving AYF-EMD, taking the limit of the discrete consistency loss when $\epsilon \to 0$ only required dividing by $\epsilon$, since the relevant term decays linearly in $\epsilon$. In contrast, the gradient in this case decays quadratically with respect to $\epsilon$, which is why we divide by $\epsilon^2$ before taking the limit.

The small difference in the AYF-EMD derivation, namely applying a stop-gradient operation on the second term in Eq. (18), has a significant effect on training dynamics. Without it, one must backpropagate through the Jacobian-vector product (JVP) used to compute $\frac{d\mathbf{f}_\theta(\mathbf{x}_t, t, s)}{dt}$. This often introduces instability and slows down training. However, with the stop-gradient operation, one must only compute the JVP without needing to backpropagate through it. Fortunately, modern autograd libraries like PyTorch support forward-mode automatic differentiation, which allows computing the JVP efficiently with minimal overhead.

Intuitively, applying the stop-gradient means the output of a large step with the flow map is pushed toward the output of following the PF-ODE trajectory for a bit and then doing a smaller step with the flow map. Without the stop-gradient, the small step is also encouraged to match the outcome of the large step, which is counterintuitive. This is because learning a flow map becomes more difficult as the interval length increases. Smaller steps are typically more reliable and offer better approximations.

Ultimately, the decision to include or omit the stop-gradient operation in Eq. (18) leads to two fundamentally different derivations and objectives. In our experiments, only the AYF-EMD variant, where the stop-gradient is applied, was able to scale effectively to large-scale image datasets and produce high-quality outputs.

### D.4.2 LMD

An analogous connection can also be made between our AYF-LMD objective and the LMD loss from Boffi et al. [5]. Recall the LMD loss from their work:

$$\nabla_\theta \mathbb{E}_{\mathbf{x}_t, t, s} \left[ w(t, s) \left\| \partial_s \mathbf{f}_\theta(\mathbf{x}_t, t, s) - \mathbf{v}_\phi(\mathbf{f}_\theta(\mathbf{x}_t, t, s), s) \right\|_2^2 \right]. \tag{32}$$

Similar to before, we will show that by removing the stop-gradient operation from Eq. (21) and allowing gradients to flow through both terms, we recover the LMD loss from Boffi et al. [5].

Concretely, we can show that as $\epsilon \to 0$, the gradient of this objective with respect to $\theta$ converges to:

$$\lim_{\epsilon \to 0} \frac{1}{\epsilon^2} \mathbb{E}_{\mathbf{x}_t, t, s} \left[ w(t, s) \| \mathbf{f}_\theta(\mathbf{x}_t, t, s) - ODE_{s' \to s}(\mathbf{f}_\theta(\mathbf{x}_t, t, s')) \|_2^2 \right]$$
$$= \nabla_\theta \mathbb{E}_{\mathbf{x}_t, t, s} \left[ w(t, s) \left\| \partial_s \mathbf{f}_\theta(\mathbf{x}_t, t, s) - \mathbf{v}_\phi(\mathbf{f}_\theta(\mathbf{x}_t, t, s), s) \right\|_2^2 \right], \tag{33}$$

where $w'(t, s) = w(t, s) \times |t - s|^2$.

The proof is as follows:

$$\lim_{\epsilon \to 0} \frac{1}{\epsilon^2} \mathbb{E}_{\mathbf{x}_t, t, s} \left[ w(t, s) \| \mathbf{f}_\theta(\mathbf{x}_t, t, s) - ODE_{s' \to s}(\mathbf{f}_\theta(\mathbf{x}_t, t, s')) \|_2^2 \right]$$
$$= \frac{1}{\epsilon^2} \lim_{\epsilon \to 0} \mathbb{E}_{\mathbf{x}_t, t, s} \left[ w(t, s) \| \mathbf{f}_\theta(\mathbf{x}_t, t, s) - (\mathbf{f}_\theta(\mathbf{x}_t, t, s') + (s - s') \cdot \mathbf{v}_\phi(\mathbf{f}_\theta(\mathbf{x}_t, t, s'), s')) \|_2^2 \right]$$
$$= \frac{1}{\epsilon^2} \lim_{\epsilon \to 0} \mathbb{E}_{\mathbf{x}_t, t, s} \left[ w(t, s) \| (\mathbf{f}_\theta(\mathbf{x}_t, t, s) - \mathbf{f}_\theta(\mathbf{x}_t, t, s')) - (\epsilon(s - t) \cdot \mathbf{v}_\phi(\mathbf{f}_\theta(\mathbf{x}_t, t, s'), s')) \|_2^2 \right]$$
$$= \frac{1}{\epsilon^2} \lim_{\epsilon \to 0} \mathbb{E}_{\mathbf{x}_t, t, s} \left[ w(t, s) \| ((s - s') \partial_s \mathbf{f}_\theta(\mathbf{x}_t, t, s) + O(\epsilon^2)) - (\epsilon(t - s) \cdot \mathbf{v}_\phi(\mathbf{f}_\theta(\mathbf{x}_t, t, s), s) + O(\epsilon^2)) \|_2^2 \right]$$
$$= \frac{1}{\epsilon^2} \lim_{\epsilon \to 0} \mathbb{E}_{\mathbf{x}_t, t, s} \left[ w(t, s) \| (\epsilon(t - s) \partial_s \mathbf{f}_\theta(\mathbf{x}_t, t, s) + O(\epsilon^2)) - (\epsilon(t - s) \cdot \mathbf{v}_\phi(\mathbf{f}_\theta(\mathbf{x}_t, t, s), s) + O(\epsilon^2)) \|_2^2 \right]$$
$$= \lim_{\epsilon \to 0} \mathbb{E}_{\mathbf{x}_t, t, s} \left[ w'(t, s) \| (\partial_s \mathbf{f}_\theta(\mathbf{x}_t, t, s) + O(\epsilon)) - (\mathbf{v}_\phi(\mathbf{f}_\theta(\mathbf{x}_t, t, s), s) + O(\epsilon)) \|_2^2 \right]$$
$$= \mathbb{E}_{\mathbf{x}_t, t, s} \left[ w'(t, s) \| \partial_s \mathbf{f}_\theta(\mathbf{x}_t, t, s) - \mathbf{v}_\phi(\mathbf{f}_\theta(\mathbf{x}_t, t, s), s) \|_2^2 \right].$$

As before, applying the stop-gradient operation allows us to avoid backpropagating through the JVP, which speeds up training. Unlike the EMD case though, we found that the LMD loss from Boffi et al.

[5] was already stable in practice and did not introduce training instabilities. Since both versions performed similarly in our experiments, we recommend using AYF-LMD for its improved training efficiency (note, however, that in all image generation experiments, we found AYF-EMD to perform better; see ablation study Tab. 6).

Intuitively, the stop-gradient in this loss plays a similar role as in the other objective: we want the output of a large step of the flow map to match the result of taking a smaller step with the flow map, followed by integrating the PF-ODE.

Another way to understand this is through Eq. (21). This loss fixes the smaller step $\mathbf{f}_\theta(x_t, t, s')$ and optimizes the larger step $\mathbf{f}_\theta(x_t, t, s)$ so that it aligns with the velocity field at the smaller step. As $s' \to s$, we can think of $\mathbf{f}_\theta(x_t, t, s)$ as fixed. The model then adjusts the slope of the flow map with respect to $s$ so that it matches the teacher flow at that point. Without the stop-gradient, the endpoint is no longer fixed and can also move to match the teacher, which changes the optimization behavior.

### D.5  MeanFlow Models

In this section, we will show the connection to MeanFlow Models [19], which are a concurrent work focused on training flow maps from scratch. We will show that the AYF-EMD objective reduces to the MeanFlow loss assuming an Euler parametrization of the flow map $\mathbf{f}_\theta(\mathbf{x}_t, t, s) = \mathbf{x}_t + (s - t)\mathbf{F}_\theta(\mathbf{x}_t, t, s)$. This parametrization is inspired by the first-order DDIM [68] solver of diffusion models, which uses Euler Integration to solve the probability flow ODE.

Recall the MeanFlow objective:

$$\mathcal{L}_{\text{MeanFlow}}(\theta) = \mathbb{E}_{\mathbf{x}_t, t, s}\left[\left\|\mathbf{F}_\theta(\mathbf{x}_t, t, s) - \left(\frac{\mathrm{d}\mathbf{x}_t}{\mathrm{d}t} - (t - s)\frac{\mathrm{d}\mathbf{F}_{\theta^-}(\mathbf{x}_t, t, s)}{\mathrm{d}t}\right)\right\|_2^2\right]. \qquad (34)$$

Taking the gradient with respect to $\theta$:

$$
\begin{aligned}
\nabla_\theta \mathcal{L}_{\text{MeanFlow}}(\theta) &= \nabla_\theta \mathbb{E}_{\mathbf{x}_t, t, s}\left[\left\|\mathbf{F}_\theta(\mathbf{x}_t, t, s) - \left(\frac{\mathrm{d}\mathbf{x}_t}{\mathrm{d}t} - (t - s)\frac{\mathrm{d}\mathbf{F}_{\theta^-}(\mathbf{x}_t, t, s)}{\mathrm{d}t}\right)\right\|_2^2\right] \\
&= \mathbb{E}_{\mathbf{x}_t, t, s}\left[2\nabla_\theta \mathbf{F}_\theta^\top \left(\mathbf{F}_\theta - \frac{\mathrm{d}\mathbf{x}_t}{\mathrm{d}t} + (t - s)\frac{\mathrm{d}\mathbf{F}_{\theta^-}}{\mathrm{d}t}\right)\right] \\
&= \mathbb{E}_{\mathbf{x}_t, t, s}\left[2\nabla_\theta \mathbf{F}_\theta^\top \left(\mathbf{F}_{\theta^-} - \frac{\mathrm{d}\mathbf{x}_t}{\mathrm{d}t} + (t - s)\frac{\mathrm{d}\mathbf{F}_{\theta^-}}{\mathrm{d}t}\right)\right] \\
&= 2\nabla_\theta \mathbb{E}_{\mathbf{x}_t, t, s}\left[\mathbf{F}_\theta^\top \cdot \left(\mathbf{F}_{\theta^-} - \frac{\mathrm{d}\mathbf{x}_t}{\mathrm{d}t} + (t - s)\frac{\mathrm{d}\mathbf{F}_{\theta^-}}{\mathrm{d}t}\right)\right],
\end{aligned}
\qquad (35)
$$

which matches the AYF-EMD objective using an Euler parametrization up to a constant, as shown in Eq. (26).

## E  Flow Maps in Prior and Concurrent Works

Flow maps (or specific instances of them) have appeared in various prior works. In the previous section, we have derived relations to some prior and concurrent works already. In this section, we provide a broader overview. Broadly, prior works on flow maps can be grouped into discrete-time and continuous-time methods.

### E.1  Discrete-time Flow Maps

Flow maps were initially proposed as a natural generalization of consistency models (CMs), and early work primarily focused on discrete-time formulations built on discrete consistency losses.

**Consistency Trajectory Models (CTM)** [38] were the first to explicitly introduce and study flow maps. CTM trains discrete-time flow maps using a combination of discrete consistency loss, flow matching loss, and an adversarial loss. Notably, CTM was also the first to introduce $\gamma$-sampling for stochastic sampling of flow maps. While the models produced high-quality samples, their performance depended heavily on the adversarial component, with a significant FID drop when it was removed.

**Trajectory Consistency Distillation** [94] builds on CTM, extending it to text-to-image generation. They empirically demonstrate that standard CMs degrade in quality as the number of function evaluations (NFEs) increases, a problem which is solved when using flow maps. These models can be viewed as a discrete variant of AYF, as shown in App. D.3.

**Bidirectional CMs** [43] observe that most prior work only trained flow maps in the denoising direction ($s < t$). They propose training on unordered timestep pairs, accelerating application like inversion, inpainting, interpolation, and blind restoration.

Kim et al. [37] trained CTMs connecting arbitrary distributions by operating within the flow matching framework instead of diffusions.

In parallel, **Multistep CMs** [22] attempted to address the poor multi-step behavior of standard CMs. They divide the denoising trajectory into subintervals and train separate CMs within each, achieving strong performance with as few as 2–8 steps. **Phased CMs** [80] adopt a similar idea but add adversarial training within each subinterval. These models effectively learn flow maps by training on $(t, s)$ pairs where $s$ is the start of a fixed subinterval containing $t$. This requires the inference step count to be pre-determined during training, and cannot be changed to trade off compute and quality.

## E.2 Continuous-time Flow Maps

More recently, several methods have tackled training flow maps in continuous time.

**Flow Map Matching** [5] provides a formal and rigorous analysis of continuous-time flow maps and proposes several continuous-time losses. However, their empirical validation is limited to small-scale experiments. In App. D.3, we discuss the connection between AYF and Flow Map Matching.

**Shortcut Models** [17] also operate in the continuous-time flow map setting. They propose an objective combining flow matching and a self-consistency loss. Their full loss (up to constants) is:

$$\mathcal{L}(\theta) = \mathbb{E}_{\mathbf{x}_t,t,s} \left[ \left\| \mathbf{F}_\theta(\mathbf{x}_t, t, t) - \frac{\mathrm{d}\mathbf{x}_t}{\mathrm{d}t} \right\|_2^2 + \left\| \mathbf{f}_\theta(\mathbf{x}_t, t, s) - \mathbf{f}_{\theta^-}\left( \mathbf{f}_{\theta^-}\left( \mathbf{x}_t, t, \frac{t+s}{2} \right), \frac{t+s}{2}, s \right) \right\|_2^2 \right],$$
(36)

where the flow map is parameterized as $\mathbf{f}_\theta(\mathbf{x}_t, t, s) = \mathbf{x}_t + (s - t)\mathbf{F}_\theta(\mathbf{x}_t, t, s)$. Intuitively, the self-consistency term encourages agreement between one large step and two smaller intermediate steps along the PF-ODE denoising path. To train from scratch, they use the empirical estimate $(\mathbf{x}_1 - \mathbf{x}_0)$ in place of $\frac{\mathrm{d}\mathbf{x}_t}{\mathrm{d}t}$. Although their results on ImageNet-256 and CelebAHQ-256 are promising, they use suboptimal architectures and achieve significantly worse FID scores than state-of-the-art methods.

The Shortcut loss can also be used for distillation by replacing $\frac{\mathrm{d}\mathbf{x}_t}{\mathrm{d}t}$ with outputs from a pretrained velocity model. We include this variant in our ablations (see Tab. 6 for details).

**Inductive Moment Matching** [95] is a recent method for training flow maps from scratch. It proposes using an MMD loss to align the *distributions* of $\mathbf{f}_\theta(\mathbf{x}_t, t, s)$ and $\mathbf{f}_{\theta^-}(\mathbf{x}_r, r, s)$ over tuples $(s, r, t)$ satisfying $s < r < t$.

## E.3 Concurrent Works

Two concurrent works have also investigated training continuous-time flow maps.

**MeanFlow Models** [19] study training flow maps from scratch using a loss closely related to our AYF-EMD objective. In fact, their formulation corresponds to a special case of AYF-EMD under an Euler flow map parameterization: $\mathbf{f}_\theta(\mathbf{x}_t, t, s) = \mathbf{x}_t + (s - t) \times \mathbf{F}_\theta(\mathbf{x}_t, t, s)$, as shown in App. D.5. They refer to $\mathbf{F}_\theta(\mathbf{x}_t, t, s)$ as the *Mean Flow* and use their objective to train flow maps from scratch, thereby being complementary to our work, which focuses on distillation. Their method achieves strong one- and two-step results on ImageNet 256x256, showing that the AYF-EMD objective is effective not just for distillation, but also for training from scratch.

**How to Build a Consistency Model** [6] extends the authors' prior work on Flow Map Matching [5], offering a deeper analysis of the EMD and LMD objectives. They also investigate the role of higher-order derivatives, finding them helpful in low-dimensional settings but largely ineffective in high-dimensional ones.

Table 3: Optimal sampling hyperparameters.

| Model | NFE | Stochasticity $\gamma$ | Guidance scale $\lambda$ |
|---|---|---|---|
| ImageNet-64, AYS-S | 1 | 1.0 | 2.0 |
| | 2 | 1.0 | 2.0 |
| | 4 | 0.8 | 2.0 |
| ImageNet-512, AYS-S | 1 | 1.0 | 2.0 |
| | 2 | 1.0 | 2.1 |
| | 4 | 0.9 | 2.0 |

In contrast to these prior and concurrent works, AYF proposes new training objectives, leverages autoguidance for distillation, and shows how brief adversarial fine-tuning can boost performance without reducing diversity. Moreover, we provide new theoretical insights and for the first time analytically prove CMs' deterioration with increased numbers of sampling steps. Finally, we overall scale flow map models to text-to-image generation and state-of-the-art few-step performance on ImageNet benchmarks.

# F   Experiment Details

## F.1   ImageNet Experiments

For our ImageNet experiments, we use publicly available checkpoints from EDM2 [36]. These models are first fine-tuned to align with the flow matching framework (see Sec. 3.4 for details) before being used as teacher models to distill a flow map. We run this finetuning stage for $10,000$ steps using a learning rate of $0.001$.

For the teacher model, we use checkpoints corresponding to the S and XS models, trained on 2147 million and 134 million images, respectively. During training, we randomize the guidance scale by sampling $\lambda$ uniformly from the range $[1,3]$.

In all experiments, we apply tangent normalization and tangent warmup, following the approach introduced in sCM [49], setting $c = 0.1$ and $H = 10000$. To ensure stable training, we define $w(t,s) = \frac{1}{|t-s|^2}$, which removes the $(s-t)^2$ term—one arising from the proportional assumption (see App. C.2.1 for details) and another from the $(s-t)$ coefficient of $\mathbf{F}_\theta(\mathbf{x}_t, t, s)$ in our flow map parameterization. We use a learning rate of $10^{-4}$ and a batch size of 2048 for all experiments for a total of $50,000$ iterations. These experiments were performed using 32 NVIDIA A100 gpus and took approximately 24-48 hours to converge.

A detailed algorithm can be seen in Algorithm 1.

**Timestep scheduling**   We sample the interval distance $|t - s|$ from a normal distribution $\mathcal{N}(P_{\text{mean}}, P_{\text{std}}^2)$, followed by a sigmoid transformation. This prioritizes medium-length intervals and improves overall training stability. Once the interval length is determined, a random interval of that length is uniformly selected, with $t > s$ set as the two endpoints. Since we are only concerned with generation, we do not train on $(t < s)$ pairs.

We find $(P_{\text{mean}}, P_{\text{std}}) = (-0.8, 1.0)$ works well for ImageNet-512, while $(P_{\text{mean}}, P_{\text{std}}) = (-0.6, 1.6)$ works well for ImageNet-64.

**Sampling hyperparameters**   At inference time, we sweep the guidance scale $\lambda$ in the range $[1, 3]$ and the sampling stochasticity $\gamma$ in $[0, 1]$ to determine optimal hyperparameters. Tab. 3 summarizes the selected values. For $n$-step sampling, intermediate timesteps $t_i$ are uniformly distributed over the interval $[0, 1]$.

**Additional ImageNet512 baselines:**   For completeness, we include the additional baselines from Table 2 of sCM [49] in Tab. 4. Our AYF models are able to outperform all methods using only 4 sampling steps.

Table 4: Sample quality on class-conditional ImageNet 512×512. This is an extension of Tab. 2 with further baseline methods.

| Method | NFE (↓) | FID (↓) | #Params |
|---|---|---|---|
| **Diffusion Models** | | | |
| ADM-G [11] | 250 | 7.72 | 559M |
| RIN [29] | 250 | 4.05 | 501M |
| U-ViT-H/4 [3] | 250 | 4.05 | 501M |
| DiT-XL/2 [58] | 250 | 3.04 | 675M |
| SimDiff [26] | 512×2 | 3.02 | 2B |
| VDM++ [39] | 512×2 | 3.15 | 2B |
| DiffIT [21] | 250×2 | 2.67 | 561M |
| DiMR-XL/3R [47] | 250×2 | 2.50 | 725M |
| DiFFUSSM-XL [84] | 250×2 | 3.41 | 673M |
| DiM-H [75] | 250×2 | 3.78 | 860M |
| U-DiT [77] | 250×2 | 3.50 | 604M |
| SiT-XL [56] | 250×2 | 2.62 | 675M |
| MaskDiT [93] | 79×2 | 2.24 | 736M |
| Dis-H/2 [15] | 250×2 | 2.88 | 900M |
| DRWvK-H/2 [16] | 250×2 | 2.95 | 879M |
| EDM2-S [36] | 63×2 | 2.23 | 280M |
| EDM2-M [36] | 63×2 | 2.00 | 498M |
| EDM2-L [36] | 63×2 | 1.87 | 778M |
| EDM2-XL [36] | 63×2 | 1.80 | 1.1B |
| EDM2-XXL [36] | 63×2 | 1.73 | 1.5B |
| **GANs & Masked Models** | | | |
| BigGAN [7] | 1 | 8.31 | 160M |
| StyleGAN-XL [65] | 1×2 | 3.92 | 266M |
| VQGAN [13] | 1024 | 12.57 | 232M |
| MaskGIT [9] | 64×2 | 9.24 | 284M |
| MAGVIT-V2 [88] | 64×2 | 9.11 | 1B |
| MAR [44] | 64×2 | 1.95 | 2.3B |
| VAR-d36-s [76]) | 10×2 | 2.63 | 2.3B |
| **AYF-S (ours)** | 1 | 3.32 | 280M |
| | 2 | 1.87 | 280M |
| | 4 | 1.70 | 280M |
| **AYF-S + adv. loss** *(ours)* | 1 | 1.92 | 280M |
| | 2 | 1.81 | 280M |
| | 4 | **1.64** | 280M |

We also compare our flow map models against several training-free accelerated diffusion samplers [51, 91], as shown in Tab. 5. While these training-free methods enable the base model to approach near-optimal performance with fewer steps, they still require roughly 32 steps or more to achieve good results. In contrast, our AYF models reach strong FID scores using only 1–4 sampling steps.

Table 5: Comparing AYF against training-free accelerated diffusion samplers on class-conditional ImageNet 512x512.

| Inference configurations | FID(↓) | | | | | | |
|---|---|---|---|---|---|---|---|
| | 1-step | 2-step | 4-step | 8-step | 16-step | 32-step | 64-step |
| EDM2 + Heun solver | 566.14 | 453.14 | 388.00 | 142.90 | 9.32 | 1.68 | 1.40 |
| EDM2 + DPM-Solver-2M [51] | 290.28 | 287.78 | 100.36 | 14.20 | 2.19 | 1.49 | 1.39 |
| EDM2 + DPM-Solver-3M [51] | 290.28 | 287.78 | 100.36 | 21.48 | 2.74 | 1.41 | 1.38 |
| EDM2 + UniPC-2M [91] | 290.28 | 287.78 | 101.16 | 12.86 | 2.07 | 1.43 | 1.38 |
| EDM2 + UniPC-3M [91] | 290.28 | 287.78 | 101.16 | 30.02 | 4.40 | 1.40 | 1.37 |
| AYF | 3.32 | 1.87 | 1.70 | 1.69 | 1.72 | 1.73 | 1.75 |

---
**Algorithm 1** Flow Map Distillation with AYF-EMD Loss.
---
1: **Input:** dataset $\mathcal{D}$ with std. $\sigma_d$, autoguided pretrained flow model $\mathbf{v}_\phi^{\text{guided}}(\mathbf{x}_t, t, \lambda)$ with guidance weight $\lambda$, model $\mathbf{F}_\theta(\mathbf{x}_t, t, s, \lambda)$, learning rate $\eta$, distance schedule $(P_{\text{mean}}, P_{\text{std}})$, guidance interval $[\lambda_{\min}, \lambda_{\max}]$, constant $c$, warmup iteration $H$.
2: **Init:** Iters $\leftarrow 0$
3: **repeat**
4:    $\mathbf{x}_0 \sim \mathcal{D}$, $\mathbf{x}_1 \sim \mathcal{N}(\mathbf{0}, \sigma_d^2\mathbf{I})$, $\tau \sim \mathcal{N}(P_{\text{mean}}, P_{\text{std}}^2)$, $\lambda \sim \text{Unif}(\lambda_{\min}, \lambda_{\max})$
5:    $d \leftarrow \sigma(\tau)$, $s \sim \text{Unif}(0, 1-d)$, $t \leftarrow s + d$, $\mathbf{x}_t \leftarrow (1-t)\mathbf{x}_0 + t\mathbf{x}_1$
6:    $\frac{\mathrm{d}\mathbf{x}_t}{\mathrm{d}t} \leftarrow \mathbf{v}_\phi^{\text{guided}}(\mathbf{x}_t, t, \lambda)$
7:    $r \leftarrow \min(0.99, \text{Iters}/H)$
8:    $\mathbf{g} \leftarrow \left(\mathbf{F}_{\theta^-}(\mathbf{x}_t, t, s, \lambda) - \frac{\mathrm{d}\mathbf{x}_t}{\mathrm{d}t}\right) + r(t-s)\frac{\mathrm{d}\mathbf{F}_{\theta^-}(\mathbf{x}_t, t, s, \lambda)}{\mathrm{d}t}$        ▷ Tangent warmup
9:    $\mathbf{g} \leftarrow \mathbf{g}/(\|\mathbf{g}\| + c)$        ▷ Tangent normalization
10:    $\mathcal{L}(\theta) \leftarrow \|\mathbf{F}_\theta(\mathbf{x}_t, t, s, \lambda) - \mathbf{F}_{\theta^-}(\mathbf{x}_t, t, s, \lambda) + \mathbf{g}\|_2^2$
11:    $\theta \leftarrow \theta - \eta\nabla_\theta\mathcal{L}(\theta)$
12:    Iters $\leftarrow$ Iters $+ 1$
13: **until** convergence
---

## F.2 Adversarial Finetuning

For adversarial finetuning, we use the StyleGAN2 discriminator [33] and follow the relativistic pairing GAN (RpGAN) formulation [28, 31]. The complete algorithm is provided in Algorithm 2.

The flow map is optimized by minimizing a combination of the AYF-EMD loss and a weighted RpGAN objective. To compute the adversarial loss, we perform one-step sampling with the flow map (i.e. setting $(t, s) = (1, 0)$) to generate negative samples. Following prior work [13], we apply an adaptive weighting scheme to balance the AYF-EMD and adversarial terms. Additionally, we multiply the adversarial loss by a fixed coefficient $\alpha = 0.1$ to ensure stable training. For the discriminator, we apply $R_1$ and $R_2$ regularization [28] with a regularization weight of $\beta = 0.1$.

We use a learning rate of $2 \times 10^{-5}$ for both networks and a batch size of 1024. Finetuning is run for approximately 3000 iterations using 32 NVIDIA A100 GPUs, taking around 4 hours in total.

## F.3 Training Algorithms

We provide the full AYF algorithm in Algorithm 1, and the variant with adversarial finetuning in Algorithm 2.

## F.4 Text-to-Image Experiments

For our text-to-image experiments, we distill FLUX.1-[dev] [41] into a few-step generator by fine-tuning a LoRA [27] on top of the FLUX base model. We use the AYF-EMD objective and train the LoRA for 10,000 iterations on 8 NVIDIA A100 GPUs. Each GPU fits a single $512 \times 512$ image, resulting in a total batch size of 8. To manage memory, gradient checkpointing and gradient partitioning is used. We also warm up the tangent over the first 2000 iterations. Since FLUX.1-[dev] is guidance-distilled, we cannot use autoguidance and instead rely solely on the distilled guidance.

We train our model using the text-to-image-2M dataset [30] from Hugging Face, which contains over 2 million real and synthetic images. We filter this dataset and train only on the 100K images generated by FLUX.1-[dev] using text prompts from GPT-4o.

## F.5 User Study Details

To evaluate our AYF-LoRA and compare it against TCD-LoRA [94] and LCM-LoRA [53] (both based on SDXL [59]), we conduct a user study with 47 participants. We use a holdout set of 200 prompts generated by GPT-4o from the text-to-image-2M dataset. For each prompt, we generate five images using each of the three methods, all sampled with four steps from the same random seed, resulting in 1000 sets of three generated images. Each participant is given a text prompt and one image from each method, with the image order randomized to prevent bias. Participants are asked

---

**Algorithm 2** Adversarial Flow Map Finetuning with AYF-EMD and Adversarial losses.

---

1: **Input:** dataset $\mathcal{D}$ with std. $\sigma_d$, autoguided pretrained flow model $\mathbf{v}_\phi^{\text{guided}}(\mathbf{x}_t, t, \lambda)$ with guidance weight $\lambda$, model $\mathbf{F}_\theta(\mathbf{x}_t, t, s, \lambda)$, learning rates $\eta_G, \eta_D$, distance schedule $(P_{\text{mean}}, P_{\text{std}})$, guidance interval $[\lambda_{\min}, \lambda_{\max}]$, constant $c$, warmup iteration $H$, discriminator $\mathbf{D}_\psi(\mathbf{x})$, adversarial weights $\alpha, \beta$.
2: **repeat**
3:    **if** Generator step **then**
4:        $\mathbf{x}_0 \sim \mathcal{D}$, $\mathbf{x}_1 \sim \mathcal{N}(\mathbf{0}, \sigma_d^2 \mathbf{I})$, $\tau \sim \mathcal{N}(P_{\text{mean}}, P_{\text{std}}^2), \lambda \sim \text{Unif}(\lambda_{\min}, \lambda_{\max})$
5:        $d \leftarrow \sigma(\tau)$, $s \sim \text{Unif}(0, 1 - d)$, $t \leftarrow s + d$, $\mathbf{x}_t \leftarrow (1 - t)\mathbf{x}_0 + t\mathbf{x}_1$
6:        $\frac{\mathrm{d}\mathbf{x}_t}{\mathrm{d}t} \leftarrow \mathbf{v}_\phi^{\text{guided}}(\mathbf{x}_t, t, \lambda)$
7:        $r \leftarrow 0.99$
8:        $\mathbf{g} \leftarrow \left( \mathbf{F}_{\theta^-}(\mathbf{x}_t, t, s, \lambda) - \frac{\mathrm{d}\mathbf{x}_t}{\mathrm{d}t} \right) + r(t - s)\frac{\mathrm{d}\mathbf{F}_{\theta^-}(\mathbf{x}_t, t, s, \lambda)}{\mathrm{d}t}$                $\triangleright$ Tangent warmup
9:        $\mathbf{g} \leftarrow \mathbf{g}/(\|\mathbf{g}\| + c)$                $\triangleright$ Tangent normalization
10:        $\mathbf{x}_0' \leftarrow \mathbf{f}_\theta(\mathbf{x}_1, 1, 0, \lambda) = \mathbf{x}_1 - \mathbf{F}_\theta(\mathbf{x}_1, 1, 0, \lambda)$
11:        $\mathcal{L}_{EMD}(\theta) \leftarrow \|\mathbf{F}_\theta(\mathbf{x}_t, t, s, \lambda) - \mathbf{F}_{\theta^-}(\mathbf{x}_t, t, s, \lambda) + \mathbf{g}\|_2^2$
12:        $\mathcal{L}_{ADV}(\theta) \leftarrow \text{Softplus}(\mathbf{D}_\psi(\mathbf{x}_0') - \mathbf{D}_\psi(\mathbf{x}_0))$
13:        $w_{adaptive} \leftarrow \text{adaptive\_weight}(\mathcal{L}_{ADV}, \mathcal{L}_{EMD})$
14:        $\mathcal{L}(\theta) \leftarrow \mathcal{L}_{Adv} + \alpha \times w_{adaptive} \times \mathcal{L}_{EMD}$
15:        $\theta \leftarrow \theta - \eta_G \nabla_\theta \mathcal{L}(\theta)$
16:    **else if** Discriminator step **then**
17:        $\mathbf{x}_0 \sim \mathcal{D}$, $\mathbf{x}_1 \sim \mathcal{N}(\mathbf{0}, \sigma_d^2 \mathbf{I})$
18:        $\mathbf{x}_0' \leftarrow \mathbf{f}_\theta(\mathbf{x}_1, 1, 0, \lambda) = \mathbf{x}_1 - \mathbf{F}_\theta(\mathbf{x}_1, 1, 0, \lambda)$
19:        $\mathcal{L}(\psi) \leftarrow \text{Softplus}(\mathbf{D}_\psi(\mathbf{x}_0) - \mathbf{D}_\psi(\mathbf{x}_0')) + \beta \times (\|\nabla_\mathbf{x}\mathbf{D}_\psi(\mathbf{x}_0)\|_2^2 + \|\nabla_\mathbf{x}\mathbf{D}_\psi(\mathbf{x}_0')\|_2^2)$
20:        $\psi \leftarrow \psi - \eta_D \nabla_\psi \mathcal{L}(\psi)$
21: **until** convergence

---

to select the best image based on quality and text alignment or indicate a tie. Fig. 7 summarizes the results. See Fig. 10 for a screenshot of the instruction. The participants were paid 0.10$ per evaluation with an average evaluation time of 15-30s per decision, with an hourly average pay of 18$/hr.

Additionally, we compare our distilled AYF models against the base model (FLUX.1 [dev]) and an adversarially distilled approach (FLUX.1 [schnell]) through three pairwise user studies. We generate 260 images from each model, using 50 sampling steps for FLUX.1 [dev] and 4 steps for both AYF and FLUX.1 [schnell]. Each pairwise study compares two methods at a time, with human raters voting for the preferred image or selecting 'Tie'. A summary of the results can be seen in Fig. 11.

As shown in the results, both distilled models exhibit some quality degradation relative to the base diffusion model, which is expected given FLUX.1 [dev]'s much higher sampling cost (50 steps). When comparing our AYF model to FLUX.1 [schnell], both achieve a similar number of preference votes in terms of image fidelity. However, FLUX.1 [schnell] shows slightly stronger text alignment, which we attribute to the limited number of rendered-text examples in our finetuning dataset. Note that our method's results are achieved by a brief LoRA-based finetuning of the base model, whereas FLUX.1 [schnell] performs full fine-tuning of the model. Future extensions could further enhance quality by adding an adversarial finetuning phase, which we found to significantly improve fidelity without loss in diversity in our ImageNet experiments.

## G    Ablation Studies

In this section, we isolate the effects of AYF's core design choices and compare our method against recent baseline consistency model and flow map approaches.

### G.1    Loss Function and Guidance Ablation

We begin by analyzing two key design decisions behind AYF: (1) using the AYF-EMD loss instead of AYF-LMD, and (2) applying autoguidance on the teacher model instead of classifier-free guidance (CFG), which is used by existing CM methods. To compare AYF-EMD and AYF-LMD, we first

**Compare the given images.**

**Instruction**

In this study, you will be shown 3 images and a text prompt. You must evaluate their quality and how well they match the text.

**Important: Please limit your participation to a maximum of 30 HITs for this study. We are seeking diverse input and wish to involve a large number of people in the process.**

**Images:**

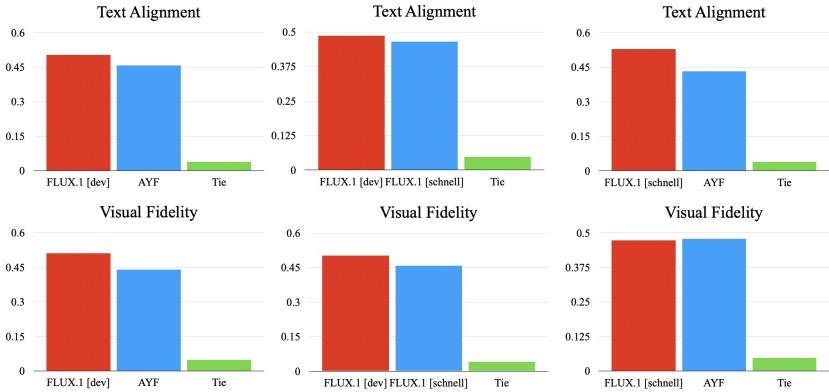

Image 1      Image 2      Image 3

**Text prompt: Neon-blue glowing gecko on finger in cyberpunk city at night**

**Q1: Assess the quality of the three images. Consider aspects like overall clarity, sharpness, and level of detail. Which image is the best?**

○ Image 1 is the best    ○ Image 2 is the best    ○ Image 3 is the best    ○ No major difference

**Q2: Evaluate how accurately each image matches the text. Considering aspects like relevance, accuracy, and completeness of depicted elements. Which image matches the text prompt most closely?**

○ Image 1 aligns best    ○ Image 2 aligns best    ○ Image 3 aligns best    ○ All align similarly

`Submit`

Figure 10: Screenshot of instructions provided to the participants for the human evaluation study.

Figure 11: Pairwise user preferences between FLUX.1 [dev] (50-step samples), FLUX.1 [schnell] (4-step samples, adversarially trained), and our LoRA-based flow map model (4-step samples).

evaluate both qualitatively on a 2D toy dataset (Fig. 12). In this setting, AYF-LMD significantly outperforms AYF-EMD. However, the trend reverses on image datasets, where AYF-EMD consistently yields better results. As shown in Tab. 6, AYF-EMD leads to significantly improved generation quality.

We also evaluate the impact of autoguidance. Replacing it with CFG consistently degrades performance across all sampling steps, highlighting the benefit of autoguidance during distillation. This difference can also be visually seen in the 2D setting (Fig. 13).

Next, we compare AYF against baseline methods. Against **sCD** [49], the current state-of-the-art consistency model, we observe that increasing the number of steps consistently degrades performance beyond 4 steps. In contrast, AYF's performance stabilizes after 8 steps and remains consistently better at 4 steps, narrowing the gap between the few-step student and the multi-step teacher. AYF

achieves this improved multi-step performance by having slightly worse single-step generations. This is expected considering that both models share the same parameter budget but AYF solves a more difficult task. However, performing a short adversarial finetuning on top of a pretrained AYF model significantly boosts performance across all sampling steps (see Fig. 8), overcoming this limitation.

Note that also with autoguidance replaced by CFG, we still outperform sCD for all steps except for single-step generation. This trend remains when re-training sCD with an autoguided teacher and comparing with the full AYF with autoguided teacher. These analyses show that while autoguidance boosts performance, our model is superior to the previous state-of-the-art CM for all sampling step settings except single-step generation also when comparing under the same guidance scheme. This is due to our novel objectives for state-of-the-art continuous-time flow map training.

Compared to **Shortcut models** [17], which we re-implemented, AYF again outperforms them consistently. While Shortcut models improve steadily with increasing NFEs, their few-step performance is significantly worse than both AYF and sCD. Unlike shortcut models, AYF reaches strong performance earlier and plateaus around 8 steps.

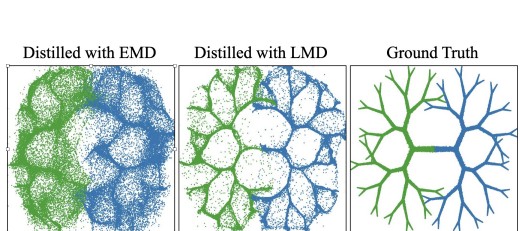

Distilled with EMD     Distilled with LMD     Ground Truth

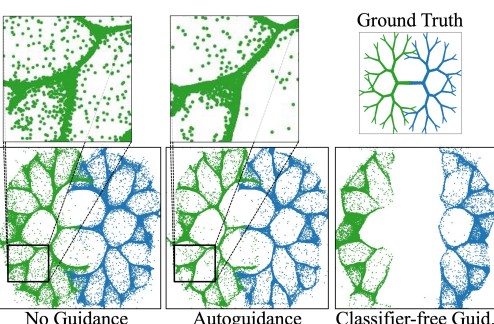

Ground Truth

No Guidance     Autoguidance     Classifier-free Guid.

Figure 12: Four-step samples from distilled AYF flow maps trained using the AYF-EMD and AYF-LMD objectives for a 2D distribution.

Figure 13: Four-step samples from distilled AYF flow maps using no guidance, autoguidance, and CFG (scale 3) for a 2D distribution.

Table 6: Ablation study on ImageNet 512x512. * indicates our reproduction of prior methods. Also see Fig. 8 for a visualization of the key results.

| Training configurations | FID($\downarrow$) | | | | | | | | |
|---|---|---|---|---|---|---|---|---|---|
| | 1-step | 2-step | 4-step | 8-step | 16-step | 24-step | 32-step | 48-step | 64-step |
| AYF-S (with autoguided teacher and AYF-EMD objective) | 3.32 | 1.87 | 1.70 | 1.69 | 1.72 | 1.75 | 1.73 | 1.74 | 1.75 |
| - with AYF-LMD objective instead of AYF-EMD | 12.45 | 8.90 | 6.70 | 6.10 | 5.85 | - | - | - | - |
| - with CFG teacher instead of autoguidance | 4.12 | 2.57 | 2.32 | 2.29 | 2.31 | - | - | - | - |
| sCD-S* (with autoguided teacher) | 2.90 | 1.87 | 1.84 | 2.08 | 2.32 | 2.68 | 2.91 | 3.72 | 4.62 |
| - with CFG teacher instead of autoguidance | 3.26 | 2.68 | 2.72 | 2.81 | 3.33 | - | - | - | - |
| Shortcut model* (with autoguided teacher) | 47.60 | 13.12 | 5.37 | 2.31 | 2.05 | 1.92 | 1.85 | 1.83 | 1.81 |
| EDM-S + autoguidance (teacher) | 566.13 | 298.60 | 89.88 | 26.41 | 6.08 | 3.09 | 2.21 | 1.65 | **1.51** |
| **Adversarial finetuning** | | | | | | | | | |
| AYF-S + adv. loss | **1.92** | **1.81** | **1.64** | **1.62** | **1.63** | **1.61** | **1.63** | **1.62** | 1.61 |

## G.2 Clamped Gradient Warmup and Stable Time Embeddings Ablations

Next, we ablate the two stabilization techniques introduced in Sec. 3.4: (i) stabilizing the teacher model's time embeddings and (ii) applying gradient warmup with clamping. We track the 2-step FID throughout training, as shown in Tab. 7. The results indicate that both techniques are critical for maintaining stable training dynamics and removing either leads to divergence.

## G.3 LoRA Rank Ablations

Finally, we ablate the effect of the LoRA rank in our text-to-image experiments. We trained four models with ranks 16, 32, 64, and 128. Qualitatively, all variants produced nearly identical outputs when given the same noise input. Quantitatively, we conducted a user study with 260 images per rank,

Table 7: 2-step FID values during training on ImageNet 512x512.

| Training configurations | Number of Images Seen | | | | | | | |
|---|---|---|---|---|---|---|---|---|
| | 4M | 8M | 12M | 16M | 20M | 24M | 28M | 32M |
| AYF (stable time embeddings + clamped gradient warmup $r_{max} = 0.99$) | 4.92 | 2.91 | 2.65 | 2.70 | 2.69 | 2.44 | 2.25 | 2.11 |
| - without stable time embeddings | 8.64 | 5.04 | 4.91 | 5.43 | 5.38 | 5.83 | 10.33 | 20.54 |
| - without clamped gradients ($r_{max} = 1.0$) | 4.91 | 2.75 | 2.53 | 2.46 | 2.72 | 4.03 | 51.04 | - |

asking human raters to evaluate outputs based on fidelity and text alignment (voting for the "best" image or selecting "Tie"). As shown in Fig. 14, all LoRA ranks performed comparably, with rank 16 showing slightly lower fidelity. Overall, our results indicate that performance is largely insensitive to the choice of LoRA rank.

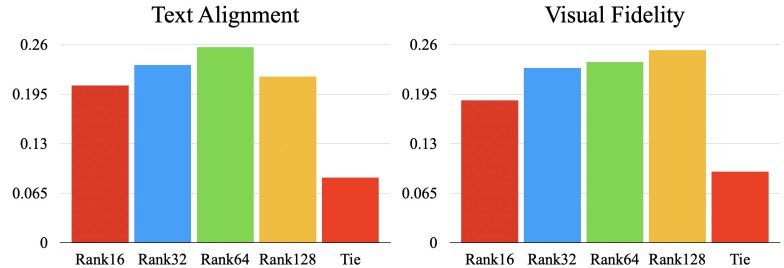

Figure 14: Ablation study on the impact of LoRA ranks in distilling FLUX.1-dev.

# H   Additional Samples

## H.1   Text-to-Image

In Fig. 15, we show additional text-to-image samples generated by our FLUX.1 [dev]-based AYF flow map model using efficient LoRA fine-tuning. We also show the effect of increasing the number of sampling steps of this model in Fig. 17. Additionally, we show some side-by-side comparisons between our model and prior LoRA based consistency models in Fig. 16. We find that our model produces sharper and more detailed images with better prompt adherence.

## H.2   ImageNet-512

In Figs. 19 to 28, we show additional one- and two-step samples generated by our ImageNet-512 AYF model. We also show the effect of increasing the number of sampling steps of this model in Fig. 18. Only very tiny quality differences are visible.

## H.3   ImageNet-64

In Figs. 29 and 30, we show additional one- and two-step samples generated by our ImageNet-64 AYF model.

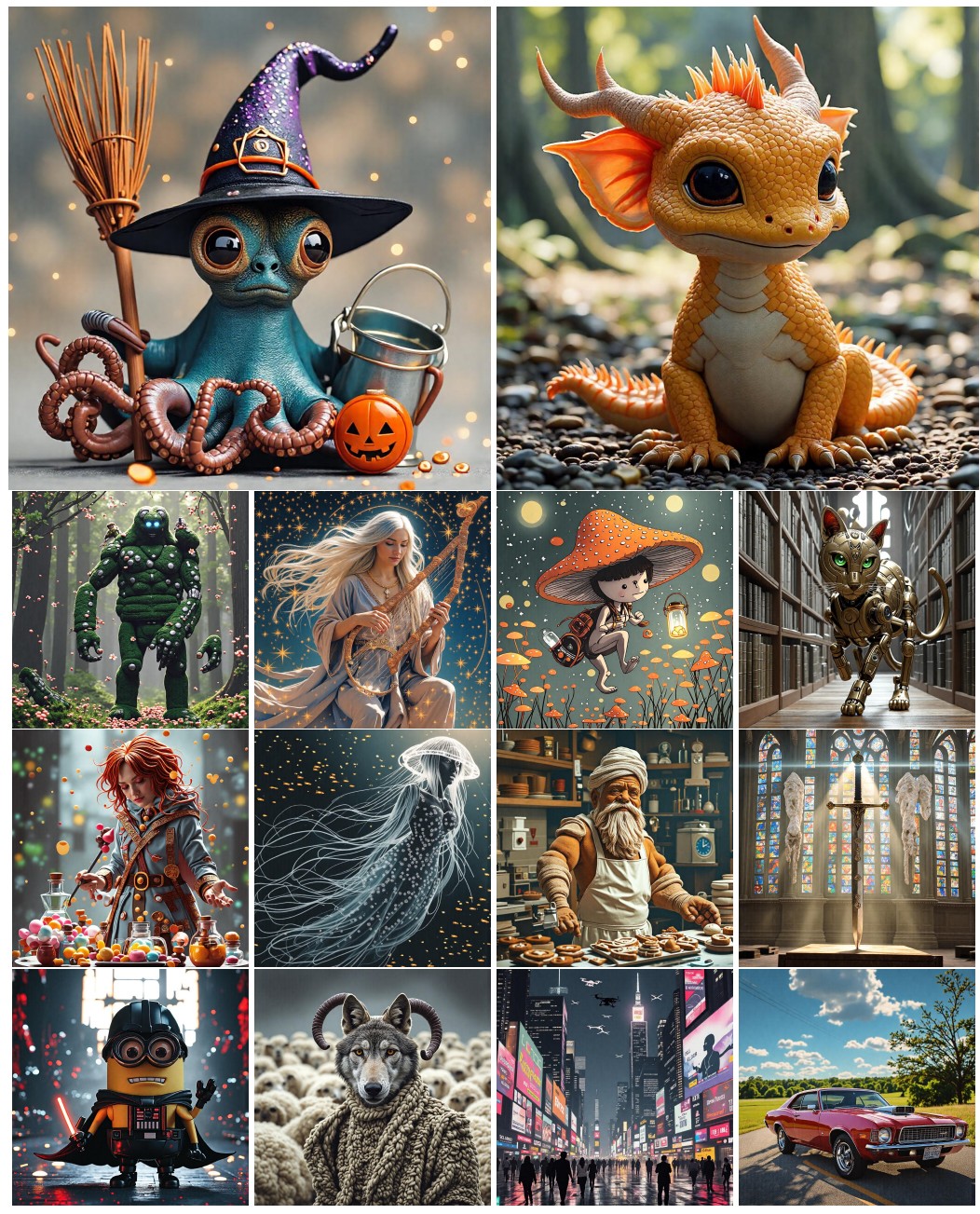

Figure 15: Selected 4-step samples generated by our FLUX.1 [dev]-based AYF flow map model using efficient LoRA fine-tuning.

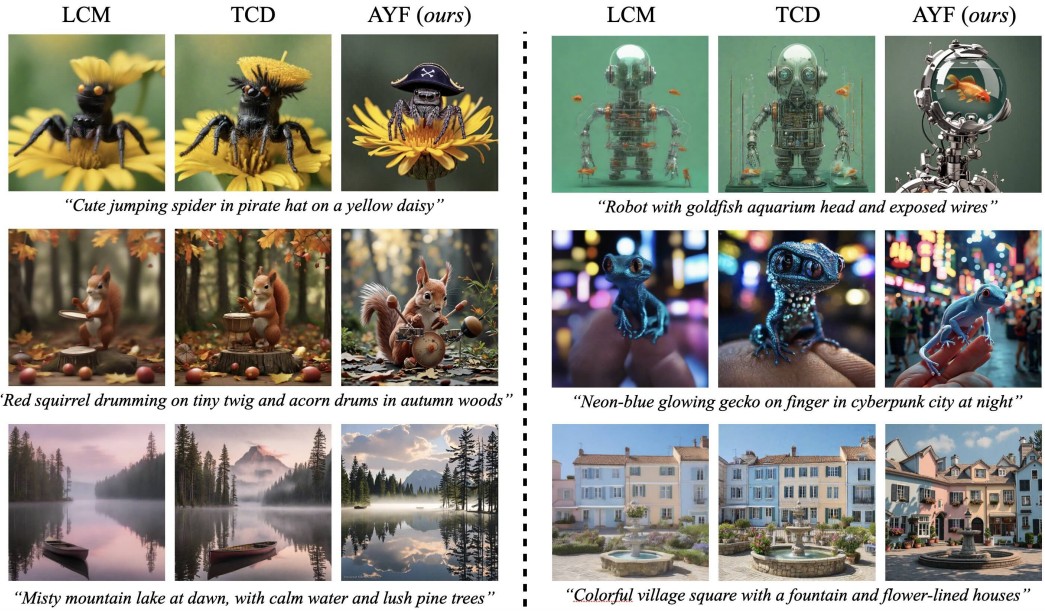

Figure 16: Qualitative comparison between 4-step samples from LCM [54], TCD [94], and AYF (view zoomed in).

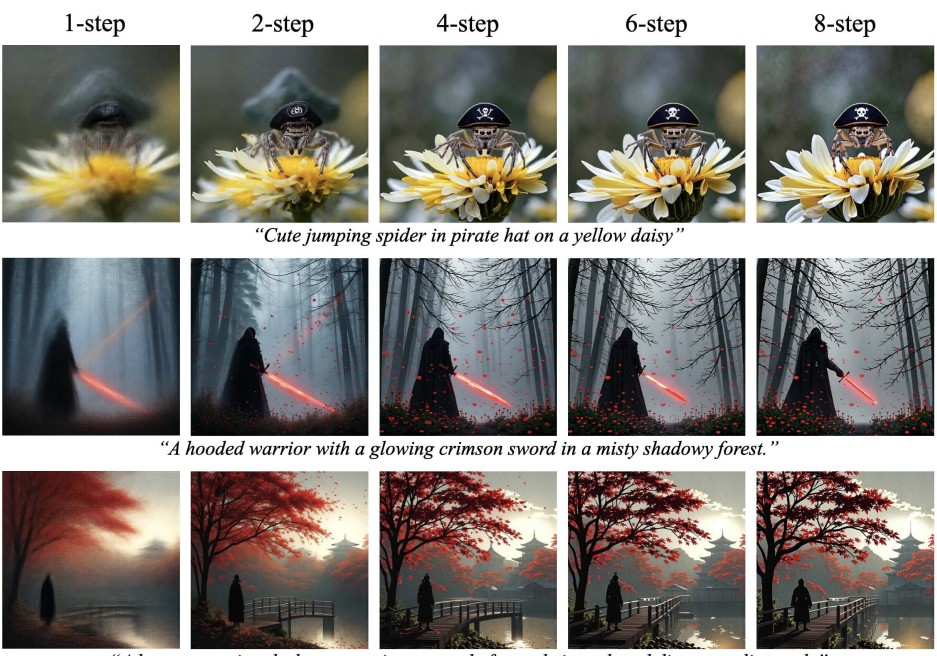

Figure 17: The effect of increasing number of steps when sampling from the text-to-image AYF model.

1-step  2-step  4-step  6-step  8-step

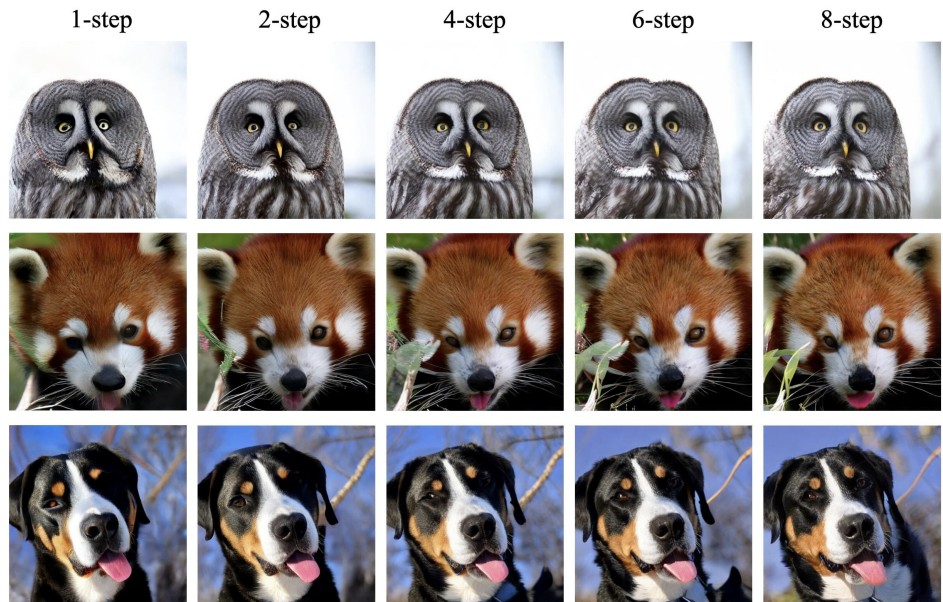

Figure 18: The effect of increasing the number of steps when sampling from the AYF model on ImageNet512 (best viewed zoomed-in).

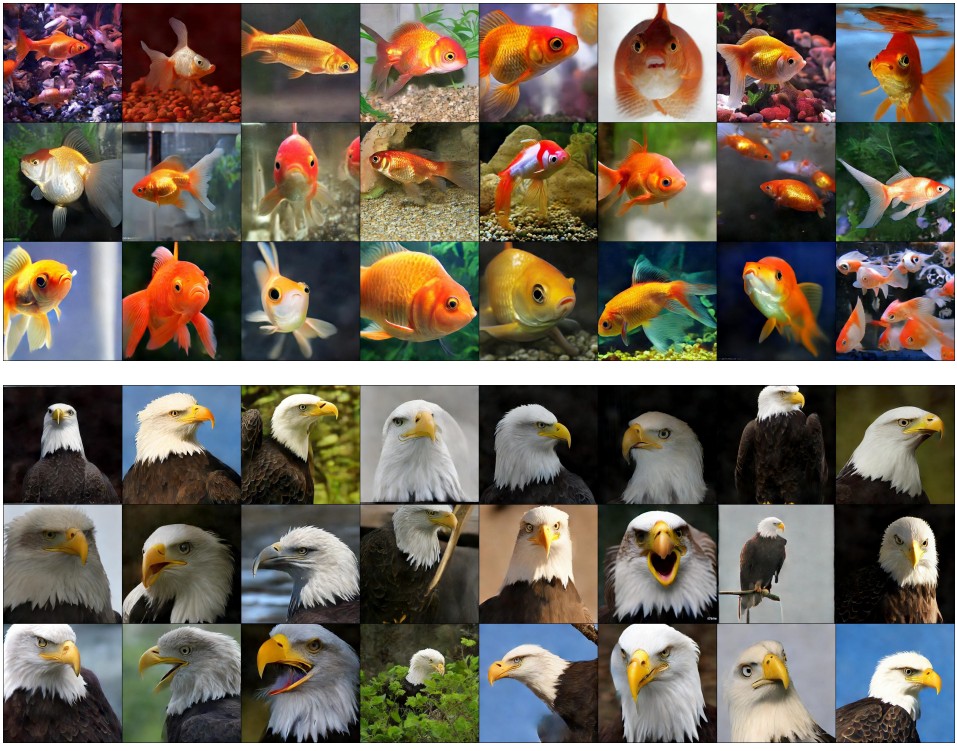

Figure 19: Selected one-step samples generated by our ImageNet512 AYF-S model, shown for classes 1 (goldfish) and 22 (bald eagle).

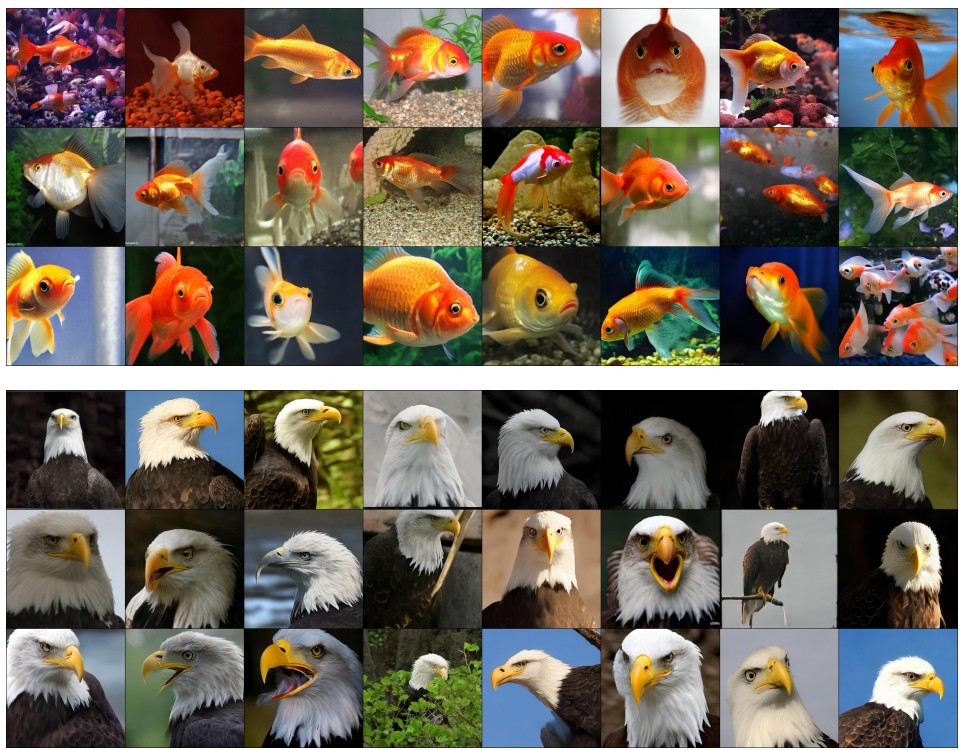

Figure 20: Selected two-step samples generated by our ImageNet512 AYF-S model, shown for classes 1 (goldfish) and 22 (bald eagle).

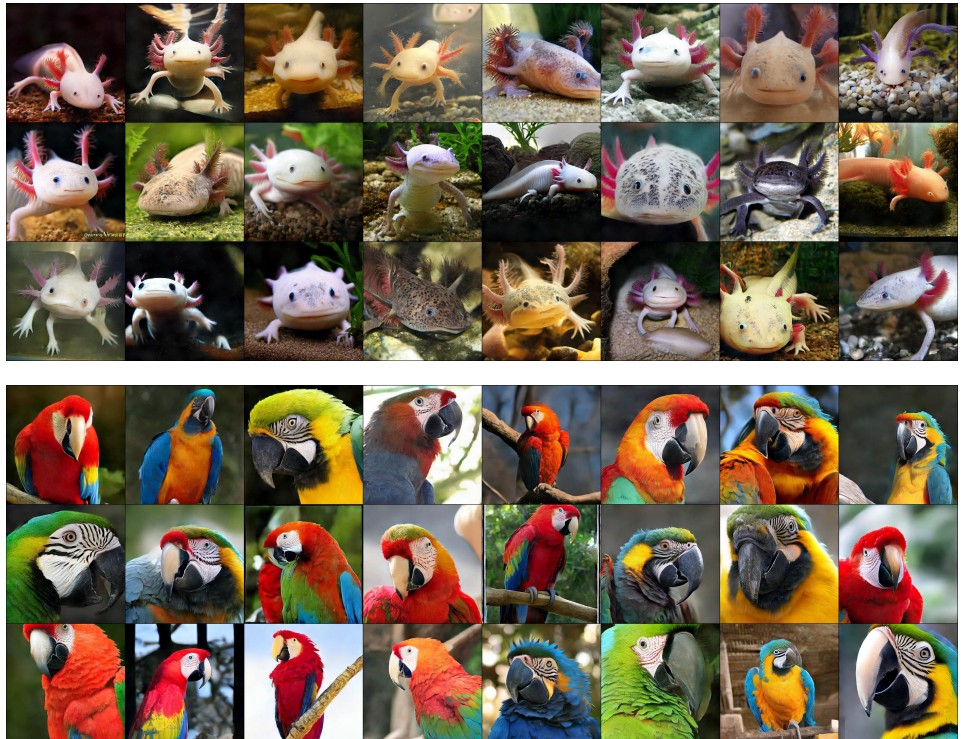

Figure 21: Selected one-step samples generated by our ImageNet512 AYF-S model, shown for classes 29 (axolotl) and 88 (macaw).

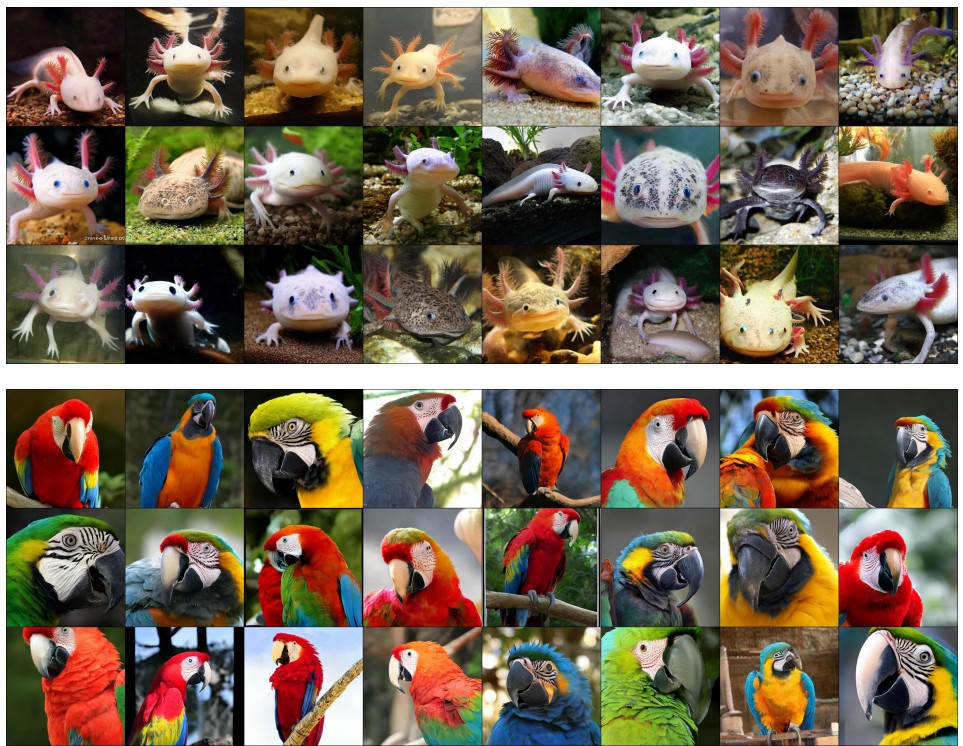

Figure 22: Selected two-step samples generated by our ImageNet512 AYF-S model, shown for classes 29 (axolotl) and 88 (macaw).

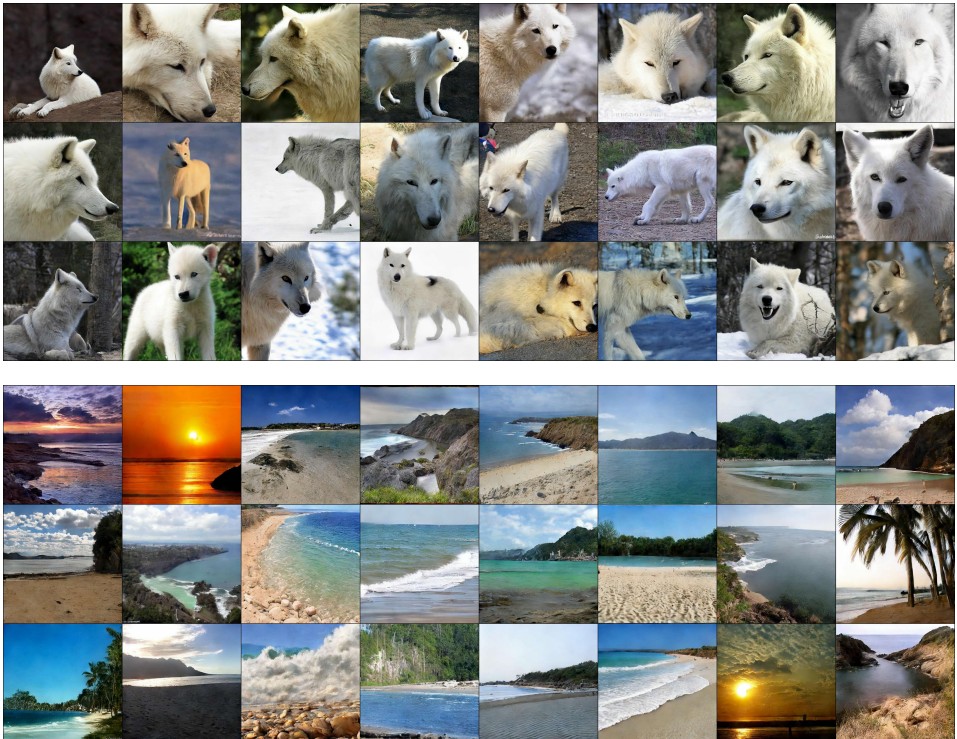

Figure 23: Selected one-step samples generated by our ImageNet512 AYF-S model, shown for classes 270 (white wolf) and 978 (coast).

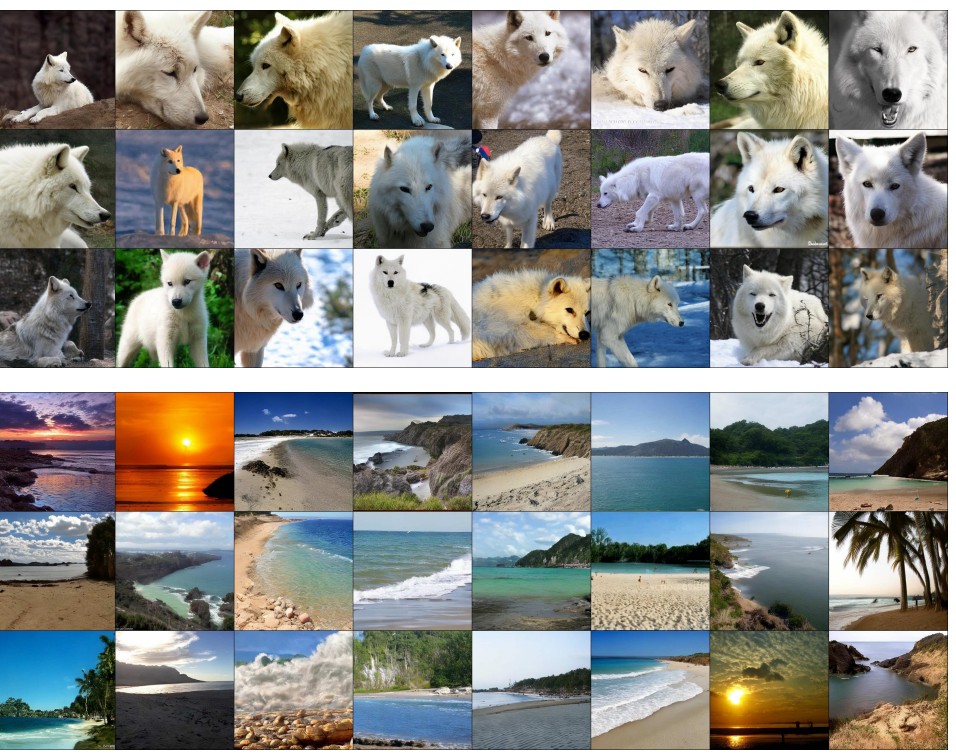

Figure 24: Selected two-step samples generated by our ImageNet512 AYF-S model, shown for classes 270 (white wolf) and 978 (coast).

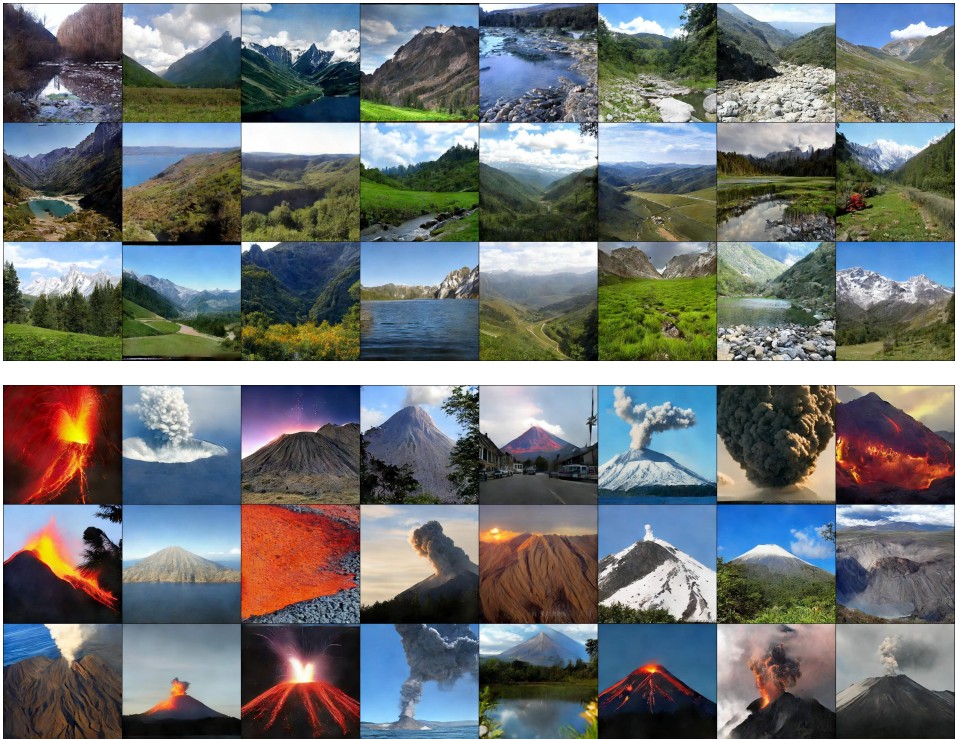

Figure 25: Selected one-step samples generated by our ImageNet512 AYF-S model, shown for classes 979 (valley) and 980 (volcano).

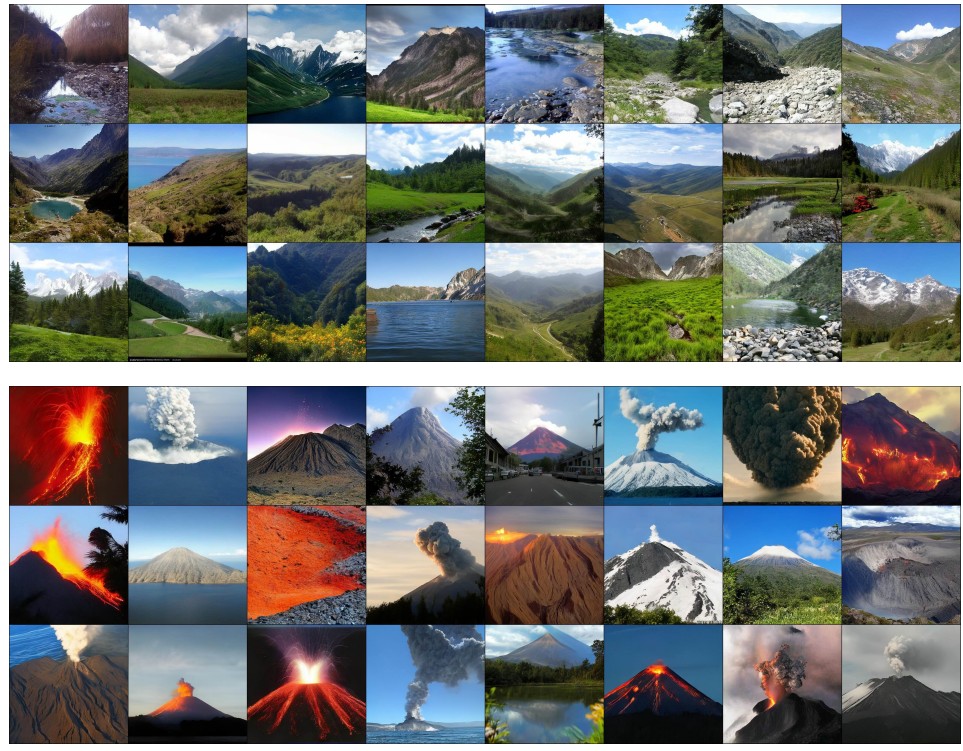

Figure 26: Selected two-step samples generated by our ImageNet512 AYF-S model, shown for classes 979 (valley) and 980 (volcano).

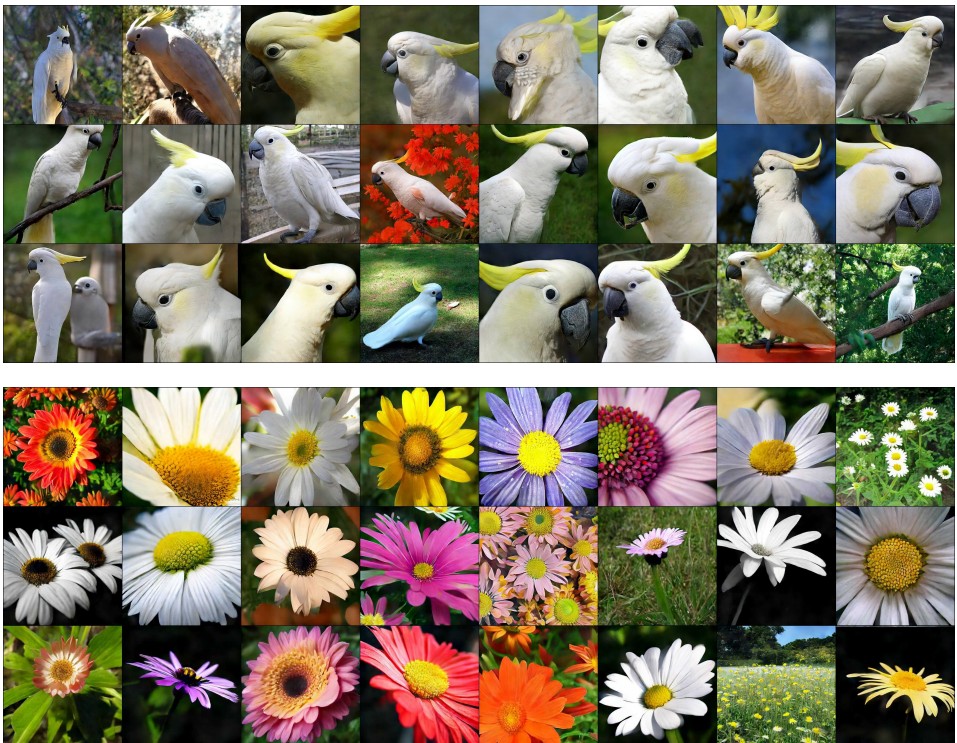

Figure 27: Selected one-step samples generated by our ImageNet512 AYF-S model, shown for classes 89 (sulphur-crested cockatoo) and 985 (daisy).

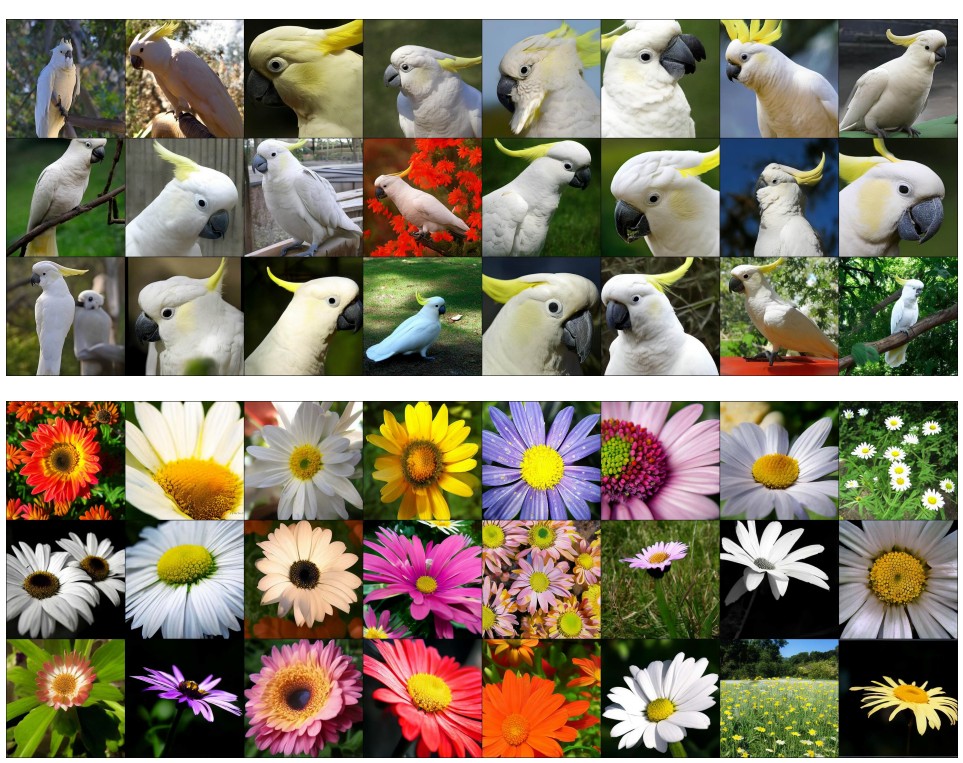

Figure 28: Selected two-step samples generated by our ImageNet512 AYF-S model, shown for classes 89 (sulphur-crested cockatoo) and 985 (daisy).

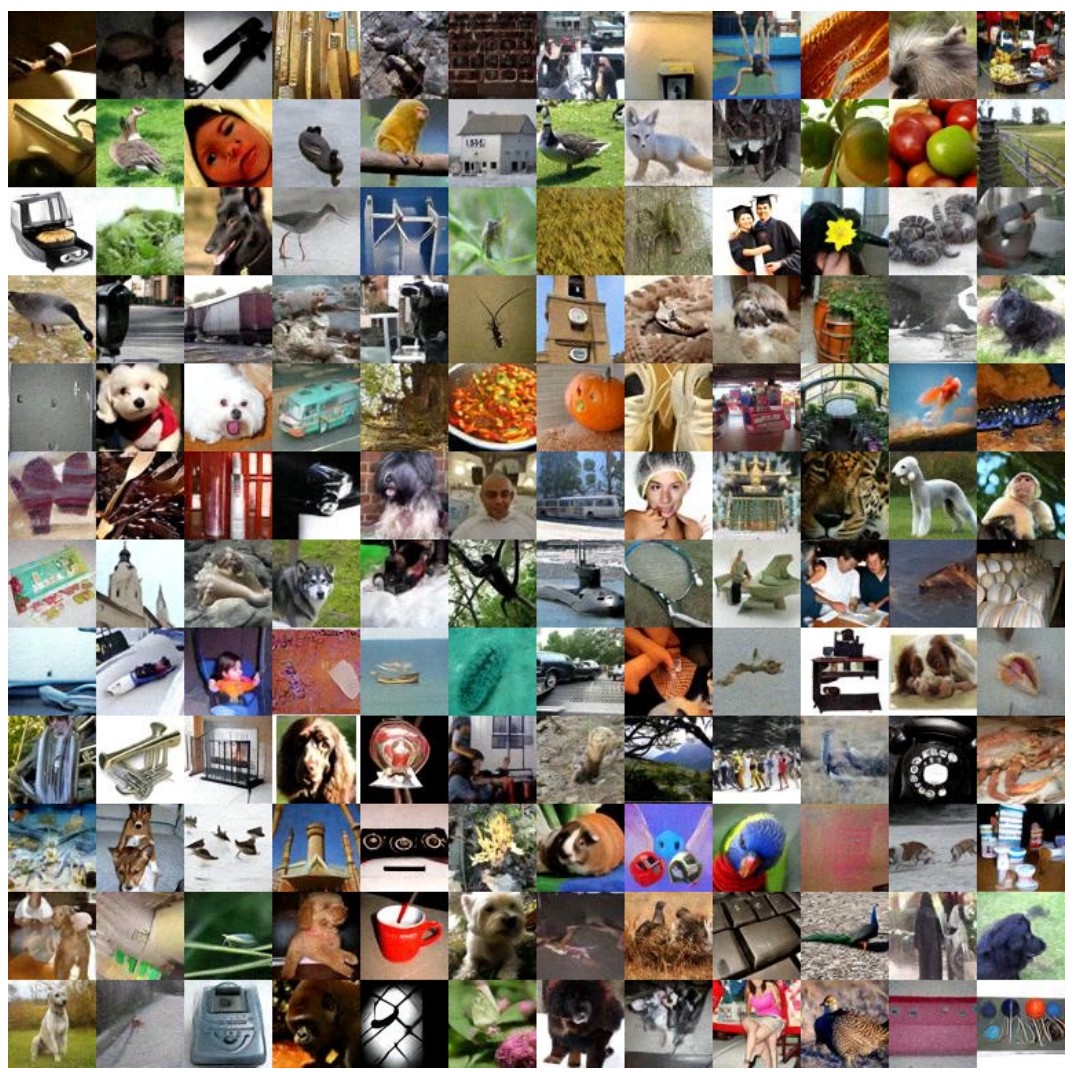

Figure 29: Uncurated one-step samples generated by our ImageNet64 AYF-S model, with randomly chosen class labels.

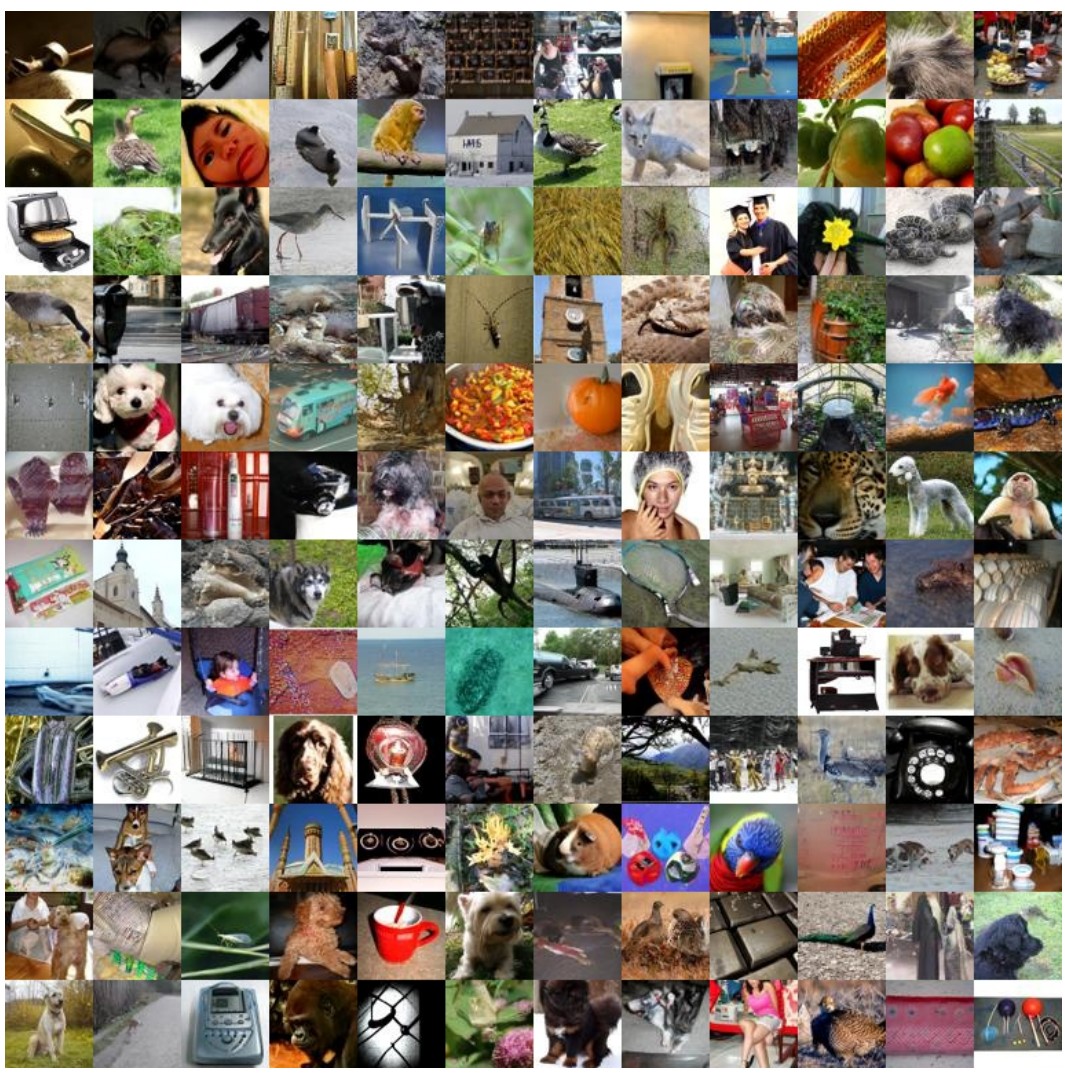

Figure 30: Uncurated two-step samples generated by our ImageNet64 AYF-S model, with randomly chosen class labels.

# I  Text Prompts for Generated Images

Here we will list all the text prompts used to generate the images in Figs. 1, 3 and 15.

- "A surreal and dreamlike scene featuring a small island with a lush green mountain encased in a giant, translucent sphere. The sphere reflects warm golden and green tones, blending harmoniously with the soft orange hues of a serene sunset. The scene is set on a calm, reflective body of water, with gentle ripples creating perfect reflections of the sphere and the sky. A lone wooden boat floats peacefully in the foreground, adding a sense of scale and solitude. Distant mountains frame the horizon, completing the ethereal and otherworldly atmosphere."

- "A dark, atmospheric forest shrouded in mist, with towering, shadowy trees. At the center stands a hooded warrior clad in black armor and a flowing cloak, holding a massive, glowing crimson sword. The blade emits an intense, fiery red light that contrasts sharply with the cool blue tones of the misty forest. Scattered red flowers grow on the forest floor, their vivid color echoing the sword's glow. Fiery red embers float through the air, adding to the ominous and mystical mood. The scene is cinematic and otherworldly, with a strong sense of power and mystery."

- "A mystical forest bathed in moonlight, with glowing blue and green bioluminescent plants. A gentle mist rolls through the trees, and faint magical runes glow on ancient stone pillars scattered throughout the scene. In the background, a shimmering waterfall cascades into a crystal-clear pool. The atmosphere is serene, with soft light beams piercing through the canopy above."

- "A hyper-realistic photograph of a delicate yet powerful creature, a rabbit with the striped fur of a tiger, combined with the soft, powdery wings and antennae of a moth. The rabbit's body is small and fluffy, with bold orange and black stripes covering its fur, and its back is adorned with large, soft moth wings that shimmer in muted tones. The scene is set in a moonlit garden, with the creature nestled among flowers, its wings gently fluttering as it sniffs at a bloom."

- "A luminous koi fish with translucent fins and shimmering galaxy-like patterns on its scales, gracefully swimming in a mystical pond under the soft light of a glowing full moon. The setting features delicate lotus flowers, glowing orbs, and vibrant foliage with intricate golden details weaving through the scene. The water reflects a celestial ambiance, with tiny stars and glowing accents creating a dreamlike atmosphere. The composition is highly detailed, with rich, deep colors of navy and gold contrasted by the vivid reds and oranges of the koi and flowers. The overall style combines fantasy realism with a touch of ornate elegance."

- "A serene bald monk in vibrant orange robes meditating and levitating above a pristine swimming pool, with a modern minimalist house and lush trees in the background. The atmosphere is calm and sunny, with bright daylight casting clean shadows. The scene captures a surreal and harmonious balance of spirituality and luxury, with vibrant colors, sharp details, and a reflective pool surface."

- "A cinematic and hyper-detailed and hyper-realistic portrait photograph captured in the rugged beauty of the scottish moor, a gorgeous young woman with wild, fiery red hair sits gracefully under a large, gnarled, ancient tree, leaning against the rough bark, her vibrant locks catching the golden hour light and flowing freely around her shoulders, her piercing green eyes sparkle with life and joy as she smiles warmly, her expression radiates youthful energy."

- "A weathered, yellow robotic cat kneels gently in a muddy alley, its mechanical paws tenderly holding a small, pure white rabbit. The robot's design is a mix of sleek and worn, with scratches and rust marks telling a story of resilience. Its glowing blue eyes exude warmth and curiosity as it carefully examines the rabbit, which has a vibrant yellow flower sprouting from its back. Around them, abandoned concrete buildings loom in the background, shrouded in a soft mist, while faint droplets of rain fall, creating tiny ripples in the puddles. The scene is quiet and melancholic, with a subtle touch of hope."

- "A lone samurai stands at the edge of a crimson maple forest, his silhouette dark against the golden light of dusk. Fallen leaves swirl around him as a gentle breeze carries the scent of

autumn through the air. A narrow wooden bridge stretches across a tranquil koi pond, its surface reflecting the fiery hues of the trees above. In the distance, the curved rooftop of a hidden temple peeks through the dense foliage, bathed in the last light of the setting sun."

- "A carnival floats above the clouds, its colorful tents and twisting roller coasters suspended in midair. The Ferris wheel glows with radiant neon lights, each cabin holding a different surreal sight—one with a floating goldfish, another with a miniature city inside. Hot air balloons drift lazily between the attractions, carrying visitors who gaze down at the soft, cotton-like clouds below. The air is filled with the sounds of distant laughter and the smell of caramel popcorn carried by the wind."

- "A warm and inviting cottage interior, lit by a crackling fireplace. The room features rustic wooden furniture, a patchwork quilt on a cozy armchair, and shelves lined with books and potted plants. Sunlight streams through a window, casting soft golden light across the space. A cat naps on a woven rug in front of the fire."

- "A charming, bright-eyed octopus sports a miniature, pointed witch's hat, its brim adorned with a delicate, sparkling broom. The octopus's eyes, shining like two bright, black jewels, sparkle with excitement as it gazes out from beneath its witchy disguise. One of its tentacles grasps a tiny, wooden broom, its bristles perfectly proportioned to the octopus's miniature size. Another tentacle holds a cauldron-shaped candy pail, its metal surface adorned with a miniature, glowing jack-o'-lantern face. The atmosphere is playful and lighthearted, with a hint of spooky, Halloween fun. The octopus's very presence seems to radiate a sense of joyful, mischievous energy, as if it's ready to cast a spell of delight on all who encounter it. highest-Quality, intricate details, visually stunning, Masterpiece"

- "A baby dragon with vibrant, golden-yellow scales sits on a forest floor, its enormous, glossy black eyes sparkling with curiosity. Its tiny horns curve gently upward, and a soft tuft of orange fur runs along its head, adding to its charm. The dragon's delicate claws rest on its chest as it tilts its head slightly, radiating an innocent, playful energy. The dappled sunlight filters through the trees, casting soft shadows around the creature while emphasizing the intricate textures of its scales and the gentle details of the forest floor."

- "A towering, moss-covered golem with glowing blue eyes trudges gently through an enchanted forest. Tiny birds nest in the cracks of its ancient stone body, and wildflowers bloom along its shoulders. Each step it takes leaves behind shimmering footprints, as if the earth itself is waking beneath it. A small, mischievous fairy perches on its shoulder, whispering secrets into its ear."

- "A celestial bard with flowing, star-speckled robes strums a crystalline harp that hums with the music of the cosmos. Their silver hair drifts as if caught in an eternal breeze, and their eyes shine like twin galaxies. As they play, glowing constellations dance around them, weaving stories of forgotten legends. The air vibrates with an ethereal melody, bending reality itself to their song."

- "A tiny mushroom spirit with a cap like a spotted toadstool scurries through a moonlit meadow, carrying a lantern made from a firefly trapped in a crystal jar. Their tiny hands clutch a satchel filled with enchanted spores, which they scatter as they run, causing luminescent fungi to sprout in their wake. The night air is filled with a soft, magical glow as they embark on their secret midnight errand."

- "A mischievous clockwork cat made of brass and polished wood prowls through an ancient library, its glowing emerald eyes scanning the towering bookshelves. Gears whir softly as it moves, its tail ticking like a pocket watch. Whenever it pounces, time seems to stutter for just a fraction of a second, as if reality itself is playing along with its game."

- "A wandering candy alchemist, dressed in a coat of shimmering, sugar-spun fabric, mixes glowing elixirs in crystal vials. Their hair is a cascade of molten caramel, and their eyes sparkle like rock candy. Each step they take leaves behind a trail of edible blossoms, and their belt is lined with tiny jars of potions that fizz, swirl, and pop with magical flavors."

- "A towering jellyfish queen glides gracefully through an underwater kingdom, her translucent tendrils trailing behind her like an elegant gown. Bioluminescent patterns ripple across her ethereal body, pulsing in sync with the deep ocean currents. Tiny fish swim in mesmerizing formations around her, drawn to the soft, hypnotic glow that follows her every movement."

- "A giant, four-armed baker made entirely of gingerbread hums a deep, rumbling tune as he kneads dough in a cozy, fire-lit kitchen. His icing-swirled eyebrows lift in delight as he pulls a tray of enchanted pastries from the oven—each one shaped like a tiny, dancing creature. The warm scent of cinnamon and sugar fills the air as his candy-button eyes twinkle with pride."

- "In the heart of an ancient cathedral, Excalibur rests upon an altar of marble, encased in shimmering, ethereal light. The stained-glass windows cast multicolored beams across the blade, illuminating the intricate runes carved into its steel. A quiet reverence fills the chamber—no one dares to approach, for legends say that only the true king may grasp its hilt without being turned to dust."

- "A mischievous minion transformed into a dark side warrior, inspired by Darth Vader, stands menacingly in a dimly lit chamber. Its yellow, cylindrical body is painted matte black, with glossy red accents glowing faintly. It wears a flowing black cape, a custom helmet with sharp edges and a single menacing goggle-eye glowing red. In its hand, a tiny yet powerful red lightsaber hums with energy. The minion's expression is a mix of determination and its usual playful mischief, as if ready to wreak havoc while still being adorably chaotic. The dark background is illuminated by faint red and blue lights, evoking the ominous atmosphere of a Sith lair."

- "A cunning anthropomorphic wolf wearing a coat made of sheep's wool stands amidst a flock of sheep. The wolf's face is sharp and expressive, with piercing golden eyes and a sly, toothy grin. The wool coat is textured and fluffy, blending seamlessly into the surrounding flock, while curved ram-like horns add an unusual twist to the disguise. The sheep in the background look curious but unaware of the predator among them. The scene is set in a misty, overcast countryside, with soft lighting emphasizing the eerie yet whimsical mood."

- "A bustling cyberpunk metropolis at night, filled with towering neon-lit skyscrapers, hovering vehicles, and busy streets lined with holographic advertisements. The city is alive with vibrant pink, purple, and blue lights reflecting off wet pavement. People in futuristic attire walk below, while drones fly overhead. A giant screen displays an AI figure speaking to the crowd."

- "A vibrant, cherry-red 1970s muscle car, gleaming under a warm afternoon sun. Chrome accents sparkle, reflecting the azure sky. The car is parked on a winding asphalt road, surrounded by lush green countryside. Show the car's powerful engine and classic design details. Capture a sense of nostalgia and freedom."

## J   Licenses

Our models and code are built upon the following codebases and datasets:

- **EDM2** (https://github.com/NVlabs/edm2): Used for our ImageNet experiments. Licensed under CC BY-NC-SA 4.0.

- **StyleGAN3** (https://github.com/NVlabs/stylegan3): We use its metric calculation logic to compute recall scores. Licensed under the NVIDIA Source Code License.

- **StyleGAN2** (https://github.com/NVlabs/stylegan2): We use its discriminator implementation for our adversarial finetuning experiments. Licensed under the NVIDIA Source Code License.

- **Diffusers** (https://github.com/huggingface/diffusers): Used for our text-to-image experiments. Licensed under the Apache License 2.0. We fine-tune a LoRA on top of FLUX.1-dev, which is under a proprietary non-commercial license (https://huggingface.co/black-forest-labs/FLUX.1-dev/blob/main/LICENSE.md).

- **Text-to-Image-2M Dataset** (https://huggingface.co/datasets/jackyhate/text-to-image-2M): Used to train our distilled text-to-image LoRAs. Licensed under the MIT License.

- **ImageNet Dataset**: Used for our main experiments. Distributed under a non-commercial research license.

