# OpenReview forum: "Align Your Flow: Scaling Continuous-Time Flow Map Distillation"
_NeurIPS.cc/2025/Conference — NeurIPS 2025 poster_

### Official Review · Reviewer_dC5b · 2025-06-23

**Clarity:** 3
**Significance:** 4
**Originality:** 4
**Rating:** 5
**Confidence:** 4

**Summary:**

Flow Matching and Diffusion are considered the current state-of-the-art (SOTA) method for media generation. These models learn a flow process from Gaussian noise to the data distribution by regressing the velocity field of the flow map which result in an iterative generative process. Inferencing the flow/diffusion models is compute expensive and generally require from 50 to 1000 function evaluations (NFE)  since it requires solving an ordinary differential equation. This paper suggest a new distillation method that generalize Consistency Models [1], and allows sampling with 2-4 NFE without hurting image quality. A critical contribution of the paper is that the Align Your Flow (AYF) method achieve this performance without adversarial loss. Given a trained velocity $v_t(x)$ of a flow/diffusion model,  the velocity $v_t(x)$ defines the trajectories $x_t$ (the flow process) such that $x_0\sim p_{\text{noise}}$ and $x_1\sim p_{\text{data}}$.  AYF learns the flow map between every $t<s\in [0,1]$, $f_{\theta}$(x,t,s) i.e., $f_{\theta}$(x_t,t,s) = x_s$. The paper suggest two training objective to learn the flow map. A crucial point is that these losses do not require to simulate the flow during training which make them tractable. Additionally, the paper provide a few "training tricks" that stabilize optimization and improve performance. Finally, the paper validates the proposed method on a number of experiments, including distillation of class-conditioned image generation models and a text-to-image model.

[1] Song, Yang, et al. "Consistency models." (2023).

**Questions:**

If you can provide any of the comparisons I mentioned on weakness 3. specially the third one, i.e., comparison of AYF with the non-distilled FLUX model.

**Ethical Concerns:**

["NO or VERY MINOR ethics concerns only"]

**Final Justification:**

1. The paper tackles the important challenge of fast sampling for flow/diffusion models.

2. AYF is able to distill flow/diffusion models without adversarial loss and achieve SOTA results for first time.

3. The contribution of the paper is novel and potentially have a large impact.

4. The paper with the additional provided results during the rebuttal provide an extensive comparison of the AYF method to baselines of class-conditioned image generation as well as on the text-to-image task. On the text-to-image class with the FLUX.1 model, they show that the AYF achieves competitive image quality to the original model (30 NFE)  with only 4 NFE. However, it appears that on the provided experiments the adversarial distillation method is still slightly better.

5. The paper is well written and provide proofs in the appendix for all of its claims.

**Limitations:**

yes

**Paper Formatting Concerns:**

.

**Quality:**

3

**Strengths And Weaknesses:**

> Strengths:
1. The paper tackles the important challenge of fast sampling for flow/diffusion models.
2. AYF is able to distill flow/diffusion models without adversarial loss and achieve SOTA results for first time.
3. The contribution of the paper is novel and potentially have a large impact.
4. The paper provide an extensive comparison of the AYF method to baselines, specially on the class-conditioned image generation.
5. The paper is well written and provide proofs in the appendix for all of its claims.

> Weaknesses:
1. The paper is not entirely self contained. For example, in line 221 it seems the authors assume the time dependency of $F_{\theta}(x,t,s)$ is inserted through a function $c_{noise}$ without mentioning it explicitly. Took me a while to understand it is assumed the reader is familiarise with the fine details of EDM [2].
2. Two training objectives are presented in the paper AYF-EMD, AYF-LMD in theorems 3.2, 3.3, however right after Theorem 3.3 the authors state that the AYF-LMD objective perform poorly on the image generation and subsequently all experiments on images in the main paper are done with the AYF-EMD objective. The AYF-LMD objective is not considered a baseline and does not show any advantage in the paper. Hence, it is my opinion that the AYF-LMD is not so interesting to the reader and adds an overhead for understanding the paper. I would recommend moving it to the appendix.
3. While on the text-to-image task the reported user study shows a great advantage to AYF, the evaluation still seems lacking and not completely fair:
  - The AYF on the FLUX [3] model is compared two baselines on the SDXL [4] and on FLUX model as well. It could certainly be that the user in general prefer images generated by the FLUX model.
  - No comparison of AYF and a  distillation baseline that uses adversarial loss. To the best of my knowledge, these type of distillation method are still considered the SOTA.
  - Most importantly, a comparison between the performance of a distilled AYF FLUX model and the non-distilled FLUX model is missing. Hence, the reader cannot know if to what degree of degradation the AYF distillation process causes (if any).

Finally, concurrent to your work, [5] seems to achieve the same method with a loss similar to the AYF-EMD, where your $F_{\theta}(x,t,s)$ is their "average velocity". They provide a different and interesting point of view on the AYF-EMD loss, and you can consider adding reference to their paper for completeness of discussion on this training objective. However, I want to stress that I acknowledge this is a concurrent work and do not count as a weaknesses or a strengths, I only provide here for the completeness of discussion.

[2] Karras, Tero, et al. "Elucidating the design space of diffusion-based generative models." Advances in neural information processing systems 35 (2022): 26565-26577.

[3] Black Forest Labs. Flux. https://github.com/black-forest-labs/flux, 2023.

[4] Podell, Dustin, et al. "Sdxl: Improving latent diffusion models for high-resolution image synthesis." arXiv preprint arXiv:2307.01952 (2023).

[5] Geng, Zhengyang, et al. "Mean flows for one-step generative modeling." arXiv preprint arXiv:2505.13447 (2025).

---

> ### Author Rebuttal · Authors · 2025-07-31
>
> **Dear reviewer dC5b,**
>
> We are glad you found our paper well written and our ideas novel. Thank you for highlighting our state-of-the-art empirical results, our extensive baseline comparisons as well as our proofs. Below we will address your questions.
>
> * **Insufficient explanation in Sec 3.4:** Thank you for pointing out the ambiguities in Section 3.4. We will make sure to clarify this point in the final version of the paper. Please do not hesitate to let us know if you see other areas of the paper where the presentation can be improved.
>
> * **AYF-LMD not being competitive in image generation**:  We appreciate the feedback. Although we did not find AYF-LMD to be competitive in image generation tasks, we did find that it was superior in our low-dimensional 2D settings in the Appendix. As such, the choice of which objective proves to be most beneficial for different applications may differ and depend on the dataset and data type (in the future, our framework may be applied in other domains, such as molecule or 3D point cloud generation, and it is not clear which objective will be better there). This is why we opted to include it in the original paper. This loss is also one of the things that differentiates flow maps from standard consistency models as it is only useful for the former. Nonetheless, we will consider your suggestion and may rearrange the content of the paper in the final version of the paper.
>
> * **Additional Text-to-Image Evaluations:** We appreciate your comments on our text-to-image experiments. The goal of our text-to-image experiments was mostly to demonstrate the scalability of our distillation approach to expressive and complex image generation tasks. An important reason why we decided on FLUX.1-dev is that AYF assumes a pretrained *continuous-time* diffusion or flow model, whereas SD1.5 and SDXL were trained with *discrete timesteps*. Therefore, we cannot directly use these models without additional complications. However, to the best of our knowledge, the only previous text-to-image consistency models are TCD and LCM, which were trained on SDXL/SD1.5, and it seemed appropriate to nonetheless compare to these works (and we outperform them significantly). We believe our AYF is the first *continuous-time* consistency or flowmap framework that has been successfully demonstrated at the text-to-image scale.
>
>   However, we agree that including additional comparisons between our few-step flow maps against an adversarially trained baseline (FLUX.1-schnell) as well as the original teacher model (FLUX.1-dev) is appropriate. As we are not able to share any images during this review process, we have conducted 3 additional pairwise user studies to compare the different approaches in terms of image fidelity and text alignment. We will include additional qualitative comparisons in the final version of the paper.
>
>   We generate 260 images from each model, using 30 steps for FLUX.1-dev and 4 steps for AYF and FLUX.1-Schnell (this follows the model's default sampling settings described on Huggingface). We then perform pairwise comparisons between each two methods. See the results below (numbers indicate preference votes on image pairs generated by the two methods with the same text prompt):
>
>   |    *FLUX.1-dev vs. AYF (ours)*        | FLUX.1-dev | AYF (ours) | Tie |
>   |------------|------------|-----|-----|
>   | Fidelity   | 266        | 229 | 25  |
>   | Alignment  | 262        | 238 | 20  |
>
>   |    *FLUX.1-dev vs. FLUX.1-schnell*        | FLUX.1-dev | FLUX.1-schnell | Tie |
>   |------------|------------|----------------|-----|
>   | Fidelity   | 261        | 238            | 21  |
>   | Alignment  | 253        | 242            | 25  |
>
>   |   *FLUX.1-schnell vs. AYF (ours)*         | FLUX.1-schnell | AYF (ours) | Tie |
>   |------------|----------------|-----|-----|
>   | Fidelity   | 246            | 249 | 25  |
>   | Alignment  | 273            | 223 | 20  |
>
>   As we can see from the results, both types of distillation cause some quality degradation compared to the base diffusion model. This is expected, but the base model's better quality comes at a much higher sampling cost (30 steps). However, when comparing our AYF model against FLUX.1-schnell, both methods achieve roughly the same number of votes in terms of image fidelity. However, the FLUX.1-schnell model appears to exhibit better text-alignment, which we hypothesize may be due to the limited number of rendered text examples in our finetuning dataset (this could be easily addressed in the future). We would like to emphasize that our method’s results are achieved by a brief LoRA-based fine-tuning of the base model, whereas FLUX.1-schnell relies on complex adversarial objectives and full fine-tuning of the model[1]. Therefore, we consider our results as highly successful and very promising, achieving such strong performance without any adversarial objectives at all here. However, future work could further boost performance with an additional adversarial fine-tuning phase, which we observed to significantly improve quality without loss of diversity in our ImageNet experiments.
>
> * **MeanFlow models:** Thank you for pointing out the concurrent MeanFlows work. We agree that this work is highly relevant to ours. The paper explores a loss similar to our AYF-EMD loss to train flow maps from scratch and achieves strong results. We will definitely include it in the discussion for the final version of the paper.
>
> We hope we were able to address your questions and concerns. We will include the above results and discussions in the final version of the paper. Thank you.
>
>
> **References:**
> 1. Sauer, A., Boesel, F., Dockhorn, T., Blattmann, A., Esser, P., & Rombach, R. (2024, December). Fast high-resolution image synthesis with latent adversarial diffusion distillation. In SIGGRAPH Asia 2024 Conference Papers (pp. 1-11).

---

> > ### Comment · Reviewer_dC5b · 2025-08-04
> >
> > I appreciate the authors' response, in particular the additional experiments. Can the authors please clarify two points:
> >
> > 1. You wrote that 260 images were generated from each model and used in the pairwise user studies. However, in your tables the numbers do not sum to 260. Can you clarify?
> >
> > 2. You choose 30 steps for the original teacher model (FLUX.1-dev), is this enough to reach the best performance of the model? For the adversarial distillation baseline (FLUX.1-schnell) you state that you used the default settings, not clear if this is the case for the for the original teacher model as well.

---

> > > ### Author Response · Authors · 2025-08-04
> > >
> > > Thank you for the reply. We are glad that the additional experiments are appreciated and we are happy to clarify:
> > > * For the pairwise experiments, each of the 260 image pairs was shown to two independent human raters to boost the sample size of the study. As a result, the total number of votes adds up to 260 x 2 = 520. There is also a small typo in the second row of the AYF vs. Flux-Schnell comparison table: the number of TIE votes should be 24 rather than 20. We'll correct this in the final version. On this occasion, we also double-checked all other numbers and can confirm that they are correct.
> > >
> > > * We are using the diffusers library for these experiments, and the default number of inference steps for FluxPipeline is set to 28, which we rounded up to 30. In our testing, we observed minimal improvements in image quality as the number of steps increased beyond 30.
> > >
> > > We will include these explanations in the final version of the paper. We hope that this answers your questions. Please do not hesitate to let us know if any further clarifications are necessary. Thank you!

---

> > > > ### Comment · Reviewer_dC5b · 2025-08-04
> > > >
> > > > Thanks for the clarifications. Overall, I think the additional results are still strong despite some advantage to the adversarial distillation method (FLUX.1-schnell), and I definitely recommend to accept this paper.

---

### Official Review · Reviewer_mGSb · 2025-07-02

**Clarity:** 2
**Significance:** 3
**Originality:** 3
**Rating:** 4
**Confidence:** 2

**Summary:**

This paper introduces Align Your Flow (AYF), a continuous-time flow map distillation framework that addresses the limitations of consistency models in multi-step sampling. The authors theoretically prove that consistency models suffer from error accumulation, motivating the use of flow maps to generalize between noise levels. They propose two novel training objectives (AYF-EMD and AYF-LMD), integrate autoguidance for improved quality, and demonstrate adversarial finetuning with minimal diversity loss. AYF achieves state-of-the-art few-step generation on ImageNet 64×64/512×512 and outperforms non-adversarial text-to-image samplers when distilling FLUX.1.

**Questions:**

Thanks for your work, I have the following questions:
- How effective is the method proposed in the article on ImageNet 256? Is there any comparison with other methods?
- For ImageNet 512×512, have AYF results been compared against recent works like DPM-Solver-3? Updated baselines would strengthen the significance claim.
- In the Text-to-Image task, LCM and TCD uses SDXL while AYF uses FLUX.1 [dev] as the base model. Is this comparison fair? Since FLUX.1 [dev] performs better than SDXL, is AYF’s superior performance due to using a better baseline rather than the AYF method itself? How is the performance if the same baseline is used?

**Ethical Concerns:**

["NO or VERY MINOR ethics concerns only"]

**Final Justification:**

Having reviewed the rebuttal as well as the feedback from the other reviewers, my concerns have been addressed. I recommend accepting this paper.

**Limitations:**

yes

**Paper Formatting Concerns:**

no.

**Quality:**

3

**Strengths And Weaknesses:**

### Strengths
- AYF generalizes consistency and flow matching objectives, introducing continuous-time losses that enable effective multi-step generation. Autoguidance integration and adversarial finetuning without severe diversity loss are notable advancements.
- AYF sets new benchmarks on ImageNet and text-to-image tasks, demonstrating superior FID metrics with fewer steps and smaller models.
- The paper provides detailed training algorithms, ablation studies, and comparisons, enhancing reproducibility.

### Weaknesses
- The paper is not easy to follow.
- The article does not provide experimental results on ImageNet 256, and the author does not explain the reason.
- In the Text-to-Image task, LCM and TCD uses SDXL while AYF uses FLUX.1 [dev] as the base model, the comparison may be unfair.

---

> ### Author Rebuttal · Authors · 2025-07-31
>
> **Dear reviewer mGSb,**
>
> We appreciate your feedback and thank you for highlighting our strong empirical results, our detailed training algorithms, as well as our ablation studies and comprehensive comparisons. Below, we will address your comments and concerns.
>
> * **Paper not easy to follow:** Please do not hesitate to kindly point to areas of the paper with ambiguities, and we will do our best to explain and to clarify in the final version of the paper. Overall, we will carefully analyze the paper and improve the writing where we believe the presentation can be improved.
>
> * **ImageNet 256:** As our paper is directly related to continuous-time consistency models [1], we followed their experiments and chose ImageNet64 and ImageNet512 to perform our evaluations on. Furthermore, we would also like to point out that many papers in the field report results primarily on ImageNet64 and very small datasets such as CIFAR10. In our case, we wanted to show the scalability of our approach, so we extended our evaluations to ImageNet512 and text-to-image models. Finally, considering that we have extensive evaluations on both ImageNet64 and ImageNet512, there are no additional insights to be gained from training on the same dataset at yet another resolution. And on ImageNet64 and ImageNet512 we already compare to many dozens of baselines (see Tables. 1, 2, as well as 4 and 5 in Appendix). Last but not least, training models on ImageNet to state-of-the-art performance requires substantial GPU resources and can in some cases take a substantial amount of time (hence, this seems beyond the scope of the rebuttal). Therefore, we opted to focus our resources on ImageNet64, ImageNet512 and text-to-image experiments.
>
> * **Comparing AYF with training-free diffusion acceleration methods:** Please note that our approach is about diffusion distillation which is not a training-free approach. Such distillation approaches are inherently different from training-free acceleration techniques of diffusion models, such as improved ODE solvers like DPM-Solver-3. Specifically, distillation techniques are able to dramatically accelerate the inference of these models by reducing their sampling steps all the way down to 1-4 steps. Training-free approaches such as DPM-Solver-3, on the other hand, are much easier to plug-and-play but still require at least around 10-15 steps for reasonable results. To illustrate this, below we compare our AYF models against DPM-Solver [2] and UniPC [3], which are two of the most widely used fast ODE solver approaches. Note that we chose these approaches because they are easy to add in a plug-and-play manner, unlike methods like DPM-Solver-3 (not 3M), which requires computing model-specific statistics and are harder to use in practice.
> | Sampling steps         | 1       | 2       | 4       | 8       | 16     | 32     | 64     |
> |------------------------|---------|---------|---------|---------|--------|--------|--------|
> | EDM2 + Heun (default)  | 566.137 | 453.135 | 388.007 | 142.896 | 9.324 | 1.675  | 1.402  |
> | EDM2 + DPM-Solver-2M   | 290.284 | 287.783 | 100.359 | 14.204  | 2.189  | 1.485  | 1.388  |
> | EDM2 + DPM-Solver-3M   | 290.284 | 287.783 | 100.359 | 21.476  | 2.735  | 1.414  | 1.379  |
> | EDM2 + UniPC-2M        | 290.284 | 287.783 | 101.163 | 12.862  | 2.072  | 1.427  | 1.376  |
> | EDM2 + UniPC-3M        | 290.284 | 287.783 | 101.163 | 30.016  | 4.399  | 1.396  | 1.369  |
> | AYF                    | 3.32    | 1.87    | 1.70    | 1.69    | 1.72   | 1.73   | 1.75   |
>
>   We see that while these training-free methods help the base model reach near-optimal performance with fewer steps, they still need around 32 steps or more for good results. In contrast, our AYF models significantly reduce the step count, reaching strong FID scores in just 1-4 steps. Note that at 1 or 2 sampling steps, these ODE multistep methods behave identically and they only start to differ when the number of steps exceeds 3. This explains why several entries in the table have the same values. We will add these results as well as a discussion on this, including references to solvers like DPM-Solver and UniPC, in the final version of the paper.
>
> * **FLUX vs SDXL:** We appreciate your insight on this matter and acknowledge that the base model from which we distill has a significant effect on the final performance after distillation. We would like to point out that the focus of our text-to-image experiments was mostly to demonstrate the scalability of our approach to expressive and complex image generation tasks. An important reason why we decided on FLUX.1-dev is that AYF assumes a pretrained continuous-time diffusion or flow model, whereas SD1.5 and SDXL were trained with discrete timesteps. Therefore, we cannot directly use these models without additional complications. However, the only previous text-to-image consistency models are TCD and LCM, which were trained on SDXL/SD1.5, and it seemed appropriate to nonetheless compare to these works (and we outperform them significantly). We believe our AYF is the first continuous-time consistency or flowmap framework that has been successfully demonstrated at the text-to-image scale. We would also like to point to our extensive experiments on ImageNet64 and ImageNet512, as well as our ablation studies in the Appendix, which unambiguously demonstrate the advantages of our framework over previous discrete-time techniques, which TCD and LCM rely on.
>
>   Finally, please note that we now performed additional comparisons between our 4-step AYF text-to-image flowmap and the teacher FLUX.1-dev model as well as the adversarially distilled FLUX.1-schnell model. Please see our response to **Reviewer dC5b** for details.
>
>
> We hope we were able to address your questions and concerns. We will include the above results and discussions in the final version of the paper. If you found our explanations convincing, we would like to kindly ask you to consider raising your score accordingly. Thank you.
>
>
> **References:**
> 1. Lu, C., & Song, Y. (2024). Simplifying, stabilizing and scaling continuous-time consistency models. arXiv preprint arXiv:2410.11081.
> 2. Lu, C., Zhou, Y., Bao, F., Chen, J., Li, C., & Zhu, J. (2025). Dpm-solver++: Fast solver for guided sampling of diffusion probabilistic models. Machine Intelligence Research, 1-22.
> 3. Zhao, W., Bai, L., Rao, Y., Zhou, J., & Lu, J. (2023). Unipc: A unified predictor-corrector framework for fast sampling of diffusion models. Advances in Neural Information Processing Systems, 36, 49842-49869.

---

> > ### Comment · Reviewer_mGSb · 2025-08-05
> >
> > I thank the author for their detailed reply. Having reviewed the rebuttal as well as the feedback from the other reviewers, my concerns have been addressed. I recommend accepting this paper.

---

### Official Review · Reviewer_jQYj · 2025-07-03

**Clarity:** 2
**Significance:** 3
**Originality:** 2
**Rating:** 4
**Confidence:** 4

**Summary:**

This paper presents "Align Your Flow" (AYF) for distilling diffusion or flow-based models into few-step samplers. This paper shows that standard consistency models (CMs) are "flawed" for multi-step generation. They introduce two continuous-time training objectives, termed Eulerian (AYF-EMD) and Lagrangian (AYF-LMD). To improve sample quality, the paper leverages distillation of an autoguided teacher model, incorporates several stability-enhancing training techniques, and employs a final adversarial finetuning step. The method is evaluated on ImageNet and text-to-image generation with state-of-the-art performance among non-adversarially trained few-step models.

**Questions:**

- Why not apply the proposed methods to the largest model, like EDM-XXL? Are there any stability concerns?
- AYF-LMD seems expensive in computation. Could you provide a cost comparison between AYF-EMD and AYF-LMD in terms of throughput or training FLOPs?
- Why does training become unstable as the tangent warmup coefficient r approaches 1, a problem not reported for the original sCM
- How does the lora rank change the model performance for the FLUX model?

**Ethical Concerns:**

["NO or VERY MINOR ethics concerns only"]

**Final Justification:**

Thanks for the response. I acknowledge the explanation regarding the absence of XXL experiments, and I recommend noting the estimated training cost or GPU hours. My opinion on AYF-LMD is similar to Reviewer dC5b’s. It seems not particularly interesting for the main task and adds overhead to understanding. The LoRA rank ablation and the text-to-image evaluations provided in response to Reviewer dC5b are helpful, and I encourage including them. Authors are encouraged to improve the clarity based on the comments.

**Limitations:**

yes

**Quality:**

3

**Strengths And Weaknesses:**

## Strengths

- Empirical results. FID scores on ImageNet 64x64 and 512x512 are highly competitive, particularly for the 2-4 steps. The method appears to achieve these results with smaller models and lower computational cost than strong baselines like sCD. This makes a compelling case for the practical utility of the proposed distillation pipeline.
- The visualization in Figure 2 and the corresponding mathematical framing are clear and provide good intuition.
- This paper discusses the flaws of Consistency Models in multi-step generation.

## Weaknesses

- The method's reliance on a collection of very specific, finely-tuned "tricks" (clamped warmup, specific time embeddings, etc.) may limit its generality and ease of application to new domains.
- Some ablation studies for some of these crucial components are missing, e.g., what happens if r is not clamped, or normalization is not applied?
- The paper introduces two objectives, AYF-EMD and AYF-LMD, but then casually dismisses the latter, stating it "leads to overly smoothened samples" on images. The paper offers zero analysis or intuition for this failure. Proposing two methods and admitting one doesn't work for the main application with a hand-wavy, one-sentence explanation is a significant scientific weakness.
- Furthermore, the Regularized Tangent Warmup section reveals that the training is unstable if the warmup coefficient r reaches 1. The proposed "fix" of clamping r at 0.99 feels like an ad-hoc hack. This suggests that the proposed AYF-EMD objective may be inherently less stable than the sCM objective it builds upon, a critical issue that is recorded rather than investigated.
- Line 353, "a fact we prove analytically for the first time," seems not to be a sound claim.
- It could be better to, instead of unifying every model under flow maps, which acts not informatively, introduce and discuss the formulation and training algorithms for each model definition.

### Typos

- "objective objective" in line 216.
- ?? in line 737.

---

> ### Author Rebuttal · Authors · 2025-07-31
>
> **Dear reviewer jQYj,**
>
> Thank you for your feedback. We are glad you found our empirical results compelling, highlighting the practical value of our proposed methods. We are also delighted that you appreciate our visualizations, the mathematical framing, and our analysis of the flaws of consistency models.
>
> Below, we will address your concerns.
>
> * **XXL runs:** We primarily wanted to show via our experiments the benefits of using flow maps with 4 steps compared to consistency models which excel in 1-2 steps. Simply put, one might wonder given great 1- and 2- step results, why even bother with 4-steps? As such, we aimed to show that smaller 4-step models can be shown to be competitive with, and even outperform, larger 1- or 2-step models. Specifically, we are demonstrating that through distilling small but autoguided models into efficient high-performance flow maps, these small but expressive AYF flow map models can outperform the large baselines. This is validated in our extensive evaluations. In particular, because we use small architectures, the wall clock sampling time of our 4-step models is faster than the sampling time of previous 1- or 2-step sCM [1] models using very large networks. This is why we focused only on the small EDM checkpoints. Moreover, training XXL models to state-of-the-art performance would be extremely costly and slow, potentially taking a week or more, and beyond the scope of the rebuttal. We thank the reviewer for their suggestion, though, and will consider adding EDM-XXL results in the final version.
>
> * **AYF-EMD vs. AYF-LMD computation requirements:** We would like to point out that our AYF-EMD and AYF-LMD objectives both require essentially the same computations (computing base velocity + computing a JVP operation), but these computations are performed in different orders.
>
>   Specifically, for AYF-EMD, we first compute the base velocity $v_t = v(x_t, t)$ (with guidance) at the current position. We must then run the flowmap $f(x_t, t, s)$ and compute its total time derivative with respect to $t$, i.e. $\frac{\mathrm{d} f(x_t, t, s)}{\mathrm{d}t} $. To do this, we use the JVP operation in pytorch using the base velocity. The output of this JVP operation is used to obtain our regression target (see Algorithm 1 in the appendix). Note that we do not backpropagate through the regression target, which means we do not have to worry about higher-order derivatives.
>
>   On the other hand, for AYF-LMD, we must first run the flowmap $f(x_t, t, s)$ and compute its time derivative with respect to $s$. Similar to before, we do this by using the JVP operation in pytorch. Next, we compute the base velocity (with guidance) at that new point $v_s = v(f(x_t, t, s), s)$, and use that velocity along with the time derivative of s to obtain our regression loss, which is also detached (i.e. also here no backpropagation through the JVP).
>
>   As such, both methods need to compute the base velocity once, and perform a JVP operation. Therefore, they have similar computations and exhibit identical training speed.
>
>
> * **Why destabilization when $r \to 1$?**
> The original sCM [1] paper introduces several training stabilization techniques, one of which is called *adaptive double normalization*. It applies a normalization to the scale and shift parameters in each AdaLN layer. In our experiments, using this normalization for both the flow map and the teacher flow-matching models results in a significant quality deterioration. As such, we opted to not include this technique in our architecture, and kept the network structure almost identical to the one from EDM2. However, this implied that we needed another means of regularization to stabilize training, leading to our proposal of clamping $r=0.99$. Importantly, unlike sCM, we make the (novel) connection between this clamping operation and a simple regularization added to the original objective, to avoid it seeming like an ad-hoc solution (see Eq. (5) in the paper and below). Without any regularization (i.e. setting $r=1$ and not using adaptive double normalization) training diverges. Note that it is standard practice in this field to leverage such regularizations.
> As we are unable to share the training curve as image during the rebuttal, we list the 2-step FID scores calculated from samples generated by the distilled model after different amounts of training:
>
>   | Training images seen         | 4M   | 8M   | 12M  | 16M  | 20M  | 24M  | 28M   |
>   |---------------------|------|------|------|------|------|------|-------|
>   | AYF without clamping| 4.91 | 2.75 | 2.53 | 2.46 | 2.72 | 4.03 | 51.04 |
>   | AYF with clamping r=0.99   | 4.92 | 2.91 | 2.65 | 2.70 | 2.69 | 2.44 | 2.25  |
>
>   In these experiments, both runs reach their maximum $r$ values at 20M images, after which the non-clamped version starts to exhibit unstable training dynamics, as seen by the diverging FIDs. In contrast, the clamped version continues to improve and exhibits smooth training curves.
>
> * **Text-to-Image LoRA rank ablations:**
> Thank you for pointing this out. In our original experiments, we opted to use rank=64 to follow prior works such as TCD [2], which used the same rank. To properly evaluate the impact of the LoRA rank, we now trained additional LoRA models with ranks 16, 32, and 128. Qualitatively, all variants produced nearly identical outputs when given the same noise input. As we are unable to share images during the rebuttal, we instead ran a user study. For each rank, we generated 260 images and asked human raters to evaluate the outputs based on fidelity and text alignment (vote for “best” images or say "Tie"). The preference vote counts are summarized below:
>
>   |             | Rank=16 | Rank=32 | Rank=64 | Rank=128 | Tie |
>   |-------------|---------|---------|---------|----------|-----|
>   | Fidelity    | 48      | 59      | 61      | 65       | 24  |
>   | Alignment   | 53      | 60      | 66      | 56       | 22  |
>
>   The results suggest that all LoRA ranks perform comparably, with rank 16 showing slightly lower fidelity. During training, we also observed that lower-rank LoRAs converge faster, while higher-rank ones require more iterations. Our ablation shows that the results are generally not very sensitive to the LoRA rank. We will add this to the final version of the paper.
>
> * **Typos:** Thank you for spotting and pointing out the typos. We will make sure to fix those in the final version of the paper.
>
> * **Better flowmap discussion:** We will include a more in-depth explanation and discussion around prior works that propose learning flowmaps in the final version of the paper. However, we would like to kindly point to both section 4 (related work) in the main text as well as to the discussions of related methods in Appendices C.3, C.5, D.1 and D.2.
>
> * **AYF-LMD being inferior:** As the reviewer points out, we introduce two objectives for training flowmaps, AYF-EMD and AYF-LMD, but in our image experiments we only end up using the former. However, we also show in the appendix that for smaller low-dimensional datasets, such as 2D points, it is the AYF-LMD loss that performs significantly better. As such, the choice of which objective to use in practice becomes entirely dependent on the dataset and even data type (keeping in mind that future work may apply our methods to data beyond images; e.g. molecules, 3D point clouds, etc.).
>
>   As for why exactly the AYF-LMD objective leads to overly smoothened outputs in image generation, our main point is our empirical evidence (see Table 5 in the appendix). Intuitively, we believe it to be due to compounding errors in the objective itself. This objective essentially regresses $f(x_t, t, s)$ towards a target that can be obtained by computing $f(x_t, t, s + \epsilon)$ and following the ODE flow from $s + \epsilon$ to $s$. However, if $f(x_t, t, s + \epsilon)$ has high error and ends up far from the true $x_{s + \epsilon}$, then following the flow from $s + \epsilon$ to $s$ may end up pushing the target away from $x_s$, leading to an error accumulation problem.
>   Note that this is not the case for AYF-EMD as we first follow the teacher ODE from $x_t$ to $x_{t - \epsilon}$ with this operation introducing no additional errors. We will add these intuitions to the final version of the paper.
>
> * **we prove for the first time:** We have not seen the problem of error accumulation in consistency models being mathematically proven anywhere else. Hence, to the best of our knowledge, this is a novel contribution to the field. Note that consistency models have become an important research direction and model class. Therefore, we believe that demonstrating and studying these flaws analytically represents an important, new insight.
>
> We hope we were able to address your questions and concerns. To summarize: we introduced two novel objectives for learning flow maps and showed that each excels in different settings (AYF-EMD for high-dimensional data, and AYF-LMD for low-dimensional tasks). Building on prior work, we also proposed training techniques that help stabilize gradients when using pretrained checkpoints and ensure stable flow map training. This allowed us to achieve state-of-the-art performance in extensive experiments and scale our AYF to the text-to-image generation level. Finally, we now extended these text-to-image experiments by ablating over different LoRA ranks, finding improvements up to rank 64, with diminishing returns beyond that.
>
> We will include the above results and discussions in the final version of the paper. If you found our explanations convincing, we would like to kindly ask you to consider raising your score, accordingly. Thank you.
>
>
> **References:**
> 1. Lu, C., & Song, Y. (2024). Simplifying, stabilizing and scaling continuous-time consistency models. arXiv preprint arXiv:2410.11081.
> 2.  Zheng, J., Hu, M., Fan, Z., Wang, C., Ding, C., Tao, D., & Cham, T. (2024). Trajectory Consistency Distillation. ArXiv, abs/2402.19159.

---

> > ### Author Response · Authors · 2025-08-06
> >
> > Dear Reviewer,
> >
> > Thank you again for your valuable review. Considering that the discussion period is approaching its end, please let us know if you have any remaining questions or concerns after reading our rebuttal. We would be happy to clarify anything further. Thank you.

---

### Official Review · Reviewer_Rb7o · 2025-07-06

**Clarity:** 3
**Significance:** 3
**Originality:** 2
**Rating:** 4
**Confidence:** 5

**Summary:**

This paper proposes two new objectives for continuous-time Flow Map distillation, which use a consistency condition at either the beginning or the end of a denoising interval. The authors first formally prove that the generation quality of a learned Consistency Models (CMs) does not always improve with the number of sampling steps, given a simple data distribution $p_{data}=\mathcal{N}(0, c^2I)$. They then derive the two objectives, based on 1-step Euler solver and 1-step Lagrangian solver. They further introduce a bunch of training techniques including autoguidance, stabilizing time embedding with modified time parametrization, and regularized tangent warmup from sCM paper, to improve the training stability and final performance. In their experiments, the proposed method demonstrates better FID than sCD and sCT, two SOTA methods in the consistency distillation family. They further show that the proposed method enables better usage of wallclock time than these baselines. The proposed method also performs well in distilling text-to-image models such as FLUX.1.

**Questions:**

I wonder if the authors would like to provide a dependency analysis, theoretical or analytical, for the replacement of the embedder.

**Ethical Concerns:**

["NO or VERY MINOR ethics concerns only"]

**Final Justification:**

Thanks the authors for their final response, please include those experiments in the revised version. Generally I think this is a paper with incremental contribution. I vote for Borderline Accept.

**Limitations:**

Yes.

**Quality:**

3

**Strengths And Weaknesses:**

\+ This paper is well written and easy to follow.

\+ The proof of the limitation of Consistency Models (CMs), though only on a very simple data distribution, is inspiring.

\+ The derivation of the the two new objectives is formal.

\+ The discussion of the use of existing techniques is helpful.

\+ The performance is good.

\- Directly modifying the time embedding module by supervised training a new embedder may introduce unbounded bias.

\- Most techniques covered in Sec. 3.4 are not original.

---

> ### Author Rebuttal · Authors · 2025-07-31
>
> **Dear reviewer Rb7o,**
>
> We are glad you found our paper well written, highlighted our model’s performance, and enjoyed our proof of the limitations of consistency models. We appreciate your feedback and address your comments below.
>
> * **EDM2 time embeddings are unstable, theoretical analysis:** Note that, as mentioned in the paper, EDM2-style models embed their timestep t by turning it into the logarithm of the inverse signal-to-noise ratio $\log \sigma_t = \log (\frac{t}{1-t})$ first, before passing the embedding on to the neural network. In such cases, when the time derivative with respect to $t$ is computed in these models, analytically the result ends up looking like:
> $$\partial_t v(x_t, t) = \partial_t NN(x_t, \log \sigma_t) = \frac{\partial}{\partial \log \sigma_t} NN(x_t, \log \sigma_t) \times (\frac{1}{\sigma_t}) \times \left(\frac{1}{(1 - t)^2}\right) = \frac{\partial}{\partial \log \sigma_t} NN(x_t, \log \sigma_t) \times \left(\frac{1}{t \times (1 - t)}\right)$$
> As we can see, when $t \to 0$ or $t \to 1$, the second term in this equation goes to $\infty$, which causes the time derivative to blow up, leading to instability. This is the reason we propose changing the time embedding from using $\log \sigma_t$ to $t$ instead.
>
> * **Using EDM2 checkpoints by stabilizing time embeddings:** To make this change, a short supervised finetuning stage is performed where we align the time embedding layers as described in Section 3.4. Optionally, we follow this with a brief full-model finetuning stage, aligning the student’s output with the teacher’s to minimize bias. In practice, we found that aligning only the time embedding yields an L2 loss of approximately 0.0011 between student and teacher outputs, which improves marginally to 0.0010 with full finetuning.
> To quantitatively measure the effect of this finetuning, we compare the original EDM2 checkpoints and our finetuned flow-matching checkpoints using FID scores at varying numbers of function evaluations (NFEs, corresponding to sampling steps). Results are shown below:
> | NFE             | 4       | 8       | 16      | 32      | 64      | 128     |
> |----------------|---------|---------|---------|---------|---------|---------|
> | EDM2           | 388.007 | 142.896 | 9.324  | 1.675 | 1.402 | 1.370   |
> | FM-Finetuned   | 192.377 | 46.82   | 7.892   | 2.370 | 1.453 | 1.381   |
>
>   As shown, our finetuned checkpoints achieve FID scores close to the original EDM2 models, with any differences falling within a reasonable margin likely due to differing timestep schedules used during sampling. In short, we observe no significant degradation in performance.
>
> * **Stable embeddings are important for smooth training:** We would also like to reiterate the importance of the timestep embedding finetuning. In addition to the analytical derivation above, here we demonstrate this empirically. To this end, we ran an additional ablation, doing AYF distillation while comparing two teacher models: one using EDM2-style embeddings and the other using our stabilized variant. The EDM2-style embedding led to frequent instabilities mid-training that ended with the run diverging entirely, whereas the stabilized variant was robust.
> As we are unable to share the training curve as an image during the rebuttal, we list the 2-step FID scores of model samples generated after different training iterations below:
> | Training images seen                        | 4M   | 8M   | 12M  | 16M  | 20M  | 24M  | 28M  | 32M   |
> |-----------------------------------|------|------|------|------|------|------|------|-------|
> | Teacher with EDM2-style embedding | 8.64 | 5.04 | 4.91 | 5.43 | 5.38 | 5.83 | 10.33| 20.54 |
> | Teacher with FM-style embeddings  | 4.92 | 2.91 | 2.65 | 2.70 | 2.69 | 2.44 | 2.25 | 2.11  |
>
>   We see that the distilled model from the original EDM2 teacher with the original embeddings does not reach low FID scores and eventually diverges (after around 28M to 32M seen training images). These results will be included in the final version of the paper.
>
> * **Novelty in Section 3.4:** As the reviewer mentioned, some techniques in Section 3.4 are adapted from sCM [1], which is the closest prior work to ours. Building on existing methods like this is standard practice, and in our case, we actually found several of sCM’s tricks to be unhelpful or even harmful, so we left them out. That said, there are also new contributions in this section. In particular, the connection between tangent warmup and regularization, as well as the idea of finetuning the time embedding layers to enable the use of pretrained checkpoints, are both novel and were not used in sCM. This is why we highlighted these new contributions in this section in the main text.
>
> We hope we were able to address your questions. In particular, we hope that we could address all concerns regarding the adaptation of the time embeddings. We will include the above results and discussions in the final version of the paper. If you found our explanations convincing, we would like to kindly ask you to consider raising your score accordingly. Thank you.
>
>
> **References**:
> 1. Lu, C., & Song, Y. (2024). Simplifying, stabilizing and scaling continuous-time consistency models. arXiv preprint arXiv:2410.11081.

---

> > ### Comment · Reviewer_Rb7o · 2025-08-05
> >
> > Thank the authors for their response. However, their focus on stability is orthogonal to my concern on the bias. I keep my initial rating.

---

### Note · Authors · 2025-08-11

Dear all,

We are excited that our paper is well received by the reviewers. In particular, our state-of-the-art performance, our extensive baseline comparisons and ablation studies, our mathematical analysis of the flaws of existing consistency models, our novel methodological approach and the associated proofs, and our paper’s writing and presentation have been highlighted.

We performed several additional experiments during the rebuttal phase:
* LoRA rank ablations on text-to-image flowmaps, with a new user study measuring relative quality across ranks. The results show that our method is not sensitive to the LoRA rank (see rebuttal to reviewer jQYj).
* New user studies comparing our 4-step flowmap, the original teacher model FLUX.1-dev, and the commercial adversarially distilled FLUX.1-Schnell. These show that our efficient LoRA-based flowmaps match the image quality of adversarial methods and fall only slightly short of the (slow, iterative) base model, consistent with our ImageNet results. This further validates our state-of-the-art performance among non-adversarial few-step generation methods (see rebuttal to reviewer dC5b).
* Ablations over the various proposed stabilization techniques. We showed that gradient warmup and stable time embeddings ensure smooth flowmap training (see rebuttals to reviewers Rb7o, jQYj).
* Comparisons against training-free acceleration approaches such as DPM-Solver and UniPC sampler. The results unambiguously show the benefit of our distillation approach in the few-step generation regime, where advanced solvers break down (see rebuttal to reviewer mGSb).

The results will be incorporated in the final paper and we believe that they further strengthen our submission. We are glad that our contributions are recognized by the reviewers, with the majority explicitly suggesting acceptance.

Final remarks on the “introduced bias when changing time embeddings” noted by reviewer Rb7o, and the points raised in the review by reviewer jQYj: (a) Regarding the bias concern, we provided additional experiments and clarifications in our discussion with reviewer Rb7o, which we believe addresses the issue. (b) We addressed reviewer jQYj’s concerns through new ablation studies on the stabilization techniques, through LoRA rank ablations, and further detailed explanations, and we hope that all questions could be answered. However, there was no engagement by the reviewer.

Thank you for your time and feedback,\
The authors of Align Your Flow

---

### Decision · Program_Chairs · 2025-09-17

**Decision:**

Accept (poster)

**Comment:**

The paper introduces a framework for creating  efficient, few-step generative models.  The main contributions are a proof on limitation of Consistency Models (CMs) and  they propose two novel training objectives for flow maps, which generalize CMs and diffusion models.
The method, called Align Your Flow (AYF), uses several techniques like autoguidance and adversarial finetuning
 from smaller and  efficient models.

Reviewers all agree that the paper presents strong contributions, including several tricks for enhancing stability and performances, backed by compelling empirical evidences. Therefore, they recommend acceptance.